# Molecular landscape and subtype-specific therapeutic response of nasopharyngeal carcinoma revealed by integrative pharmacogenomics

Ren-Bo Ding[1,2,6], Ping Chen[1,2,3,6], Barani Kumar Rajendran[1,2,6], Xueying Lyu[1,2,6], Haitao Wang[1,2], Jiaolin Bao[1,2], Jianming Zeng[1,2], Wenhui Hao[1,2], Heng Sun[1,2], Ada Hang-Heng Wong [1,2], Monica Vishnu Valecha[1,2], Eun Ju Yang[1,2], Sek Man Su[1,2], Tak Kan Choi[1,2], Shuiming Liu[4], Kin Iong Chan[4], Ling-Lin Yang[3], Jingbo Wu[3], Kai Miao [1,2,5], Qiang Chen[1,2,5], Joong Sup Shim [1,2,5], Xiaoling Xu [1,2,5] & Chu-Xia Deng [1,2,5✉]

Nasopharyngeal carcinoma (NPC) is a malignant head and neck cancer type with high morbidity in Southeast Asia, however the pathogenic mechanism of this disease is poorly understood. Using integrative pharmacogenomics, we find that NPC subtypes maintain distinct molecular features, drug responsiveness, and graded radiation sensitivity. The epithelial carcinoma (EC) subtype is characterized by activations of microtubule polymerization and defective mitotic spindle checkpoint related genes, whereas sarcomatoid carcinoma (SC) and mixed sarcomatoid-epithelial carcinoma (MSEC) subtypes exhibit enriched epithelial-mesenchymal transition (EMT) and invasion promoting genes, which are well correlated with their morphological features. Furthermore, patient-derived organoid (PDO)-based drug test identifies potential subtype-specific treatment regimens, in that SC and MSEC subtypes are sensitive to microtubule inhibitors, whereas EC subtype is more responsive to EGFR inhibitors, which is synergistically enhanced by combining with radiotherapy. Through combinational chemoradiotherapy (CRT) screening, effective CRT regimens are also suggested for patients showing less sensitivity to radiation. Altogether, our study provides an example of applying integrative pharmacogenomics to establish a personalized precision oncology for NPC subtype-guided therapies.

[1] Cancer Centre, Faculty of Health Sciences, University of Macau, Macau SAR, China. [2] Centre for Precision Medicine Research and Training, Faculty of Health Sciences, University of Macau, Macau SAR, China. [3] Department of Oncology, The Affiliated Hospital of Southwest Medical University, Luzhou, Sichuan, China. [4] Kiang Wu Hospital, Macau SAR, China. [5] MOE Frontier Science Centre for Precision Oncology, University of Macau, Macau SAR, China. [6] These authors contributed equally: Ren-Bo Ding, Ping Chen, Barani Kumar Rajendran, Xueying Lyu. ✉email: cxdeng@um.edu.mo

Nasopharyngeal carcinoma (NPC) is a type of malignant tumor that is commonly found in specific geographical locations including Southeast Asia, North Africa, southern provinces of China, Hong Kong, and Taiwan[1–3]. In 2015, 60,600 new NPC cases were identified, and 34,100 patients died in China[4]. Many risk factors are involved in NPC, including environmental factors, Epstein–Barr virus (EBV) infection, smoking, diet, and personal lifestyle of peoples, etc.[5], yet how these factors contribute to NPC formation is poorly understood. Genomic variations and familial risks are other important causes for NPC, with more young people being affected by NPC than other cancer types[6]. Unlike other head and neck cancers, the asymptomatic nature of NPC is a major challenge, which hinders the early diagnosis of this disease. Therefore, many patients are diagnosed at advanced stages, which reduces the patient's overall survival (OS)[6,7].

Earlier studies proposed an array of genetic factors and genetic aberrations leading to the development of NPC, including NF-κB pathway activating mutations, chromatin modification-related mutations, ERBB-PI3K signaling activating mutations, etc.[5,8–11]. However, many questions, such as those regarding the major causative factors, the key driving pathways, druggable genetic targets in NPC, especially in different histological subtypes, are still unanswered clearly.

According to the current World Health Organization (WHO) classification system, NPC is classified into three major histological subtypes: keratinizing squamous cell carcinoma (KSCC), non-keratinizing carcinoma, and basaloid squamous cell carcinoma. Non-keratinizing tumors are further subcategorized as non-keratinizing undifferentiated carcinoma (NKUC) and non-keratinizing differentiated carcinoma (NKDC)[12]. Although the WHO subtype system is the most common form of NPC clinical classification, an increasing number of clinicians realize that the current WHO classification is insufficient for predicting chemotherapy and radiotherapy (RT) outcomes[13,14]. The prognosis does not differ significantly between NKUC and NKDC, which are the two major subtypes of NPC and account for ~95% of all cases in China[13]. A new prognostic histopathologic classification system of NPC has emerged that classifies NPC into four subtypes based on morphologic characteristics: epithelial carcinoma (EC), sarcomatoid carcinoma (SC), mixed sarcomatoid-epithelial carcinoma (MSEC), and squamous cell carcinoma (SCC), and these NPC subtypes could be linked to prognosis[13]. However, further information about the genomic features and therapeutic response among different subtypes is worthy of investigation.

RT is established as the definitive treatment for nonmetastatic NPC at an early stage, which leads to favorable clinical and survival outcomes with a 5-year OS rate of 87.3–93%, for stage I NPC patients[15–18]. However, due to the intrinsic invasiveness and asymptomatic nature of the disease, the majority of NPC patients (60–70%) are diagnosed at an advanced stage, with local spread or regional lymph node metastasis[5]. For NPC patients with recurrent/metastatic tumor, the outcome is very poor, with a median OS of ~20 months[19], and chemoradiotherapy (CRT) is the standard-of-care treatment at this stage as recommended by the National Comprehensive Cancer Network guidelines (v2.2018). If the cancer cells have spread to distant organs, chemotherapy is the only option[19,20]. The commonly used chemotherapeutics for treating NPC includes cisplatin, fluorouracil, docetaxel, paclitaxel, gemcitabine, capecitabine, irinotecan, doxorubicin, vinorelbine, carboplatin and oxaliplatin. Several studies have indicated that CRT can significantly improve therapeutic outcome compared with RT alone[21,22]. To investigate new strategies identifying the personalized optimal CRT and chemotherapy regimens is promising to improve the prognosis of NPC patients. Furthermore, since only a few chemical drugs have been approved for NPC, to explore more therapeutic drugs that can be used for the treatment of NPC is necessary.

Recent studies have employed the patient-derived organoid (PDO) culture system for drug screening against various cancers, including breast[23], colorectal[24], gastric[25], gastrointestinal[26], prostate[27], esophageal[28], liver[29], pancreatic[30], and bladder cancers[31] etc. Vlachogiannis et al., had demonstrated the good potential of this system to accurately predict the clinical responses of cancer drugs using organoids derived from metastatic, heavily pretreated colorectal and gastroesophageal cancer patients. Their data indicated that the PDOs showed as high as 88% positive predictive value and 100% negative predictive value in cancer patients[26]. Similar high accuracy in predicting clinical outcomes were also revealed by other studies[32–34].

In this study, to explore the application of the PDO system in screening chemotherapy drugs that could be used alone or in CRT for NPC, we have conducted a pharmacogenomics-based precision medicine approach by integrating genomics with a drug sensitivity test on PDOs to comprehensively investigate effective regimens for individual patients. Our integrative approach uncovers potential subtype-related and patient-specific strategies for the use of drugs and CRT combinations against this deadly disease.

## Results

**Clinical information and histological subtyping of NPCs.** A total of 106 NPC tumors were collected from the hospital with routine records (Supplementary Table 1; Supplementary Data 1, 2), and we further studied their histological features. All NPC patients were of Chinese origin with a median diagnosed age of 48 years. EBV status was examined on 40 tumor samples, the results indicated that 36 samples showed strongly positive, and 4 samples were low positive (Supplementary Data 1). With characterization on WHO subtypes, 73.58% tumors were NKUC, 23.58% were NKDC, and 2.83% were KSCC (Supplementary Table 1; Supplementary Data 1). Based on the standard of the new NPC classification system[13], all tumors were classified into four subtypes: (1) epithelial carcinoma (EC) (57/106, 53.77%), which is mainly composed of morphologically round epithelial cells; (2) sarcomatoid carcinoma (SC) (20/106, 18.87%), which contains large proportions of spindle sarcomatoid cells; (3) mixed sarcomatoid-epithelial carcinoma (MSEC) (26/106, 24.53%), which shares both features of EC and SC; and (4) squamous cell carcinoma (SCC) (3/106, 2.83%), which is characterized by keratinizing phenotype that is rarely found in other subtypes (Fig. 1a). More percent of distant metastasis was diagnosed in SC subtype patients (35%) than in EC subtype patients (12.28%) (Supplementary Data 1).

Next, we conducted the immunofluorescence assay using several antibodies against tumor subtype markers. The data indicated that the EC subtype extensively expressed pan-epithelial markers, including AE1/3, CK5/6, and P63, and the SC subtype was largely positive for vimentin, while MSEC exhibited a mixed pattern of both epithelial and sarcomatoid cell markers (Fig. 1b, c; Supplementary Fig. 1a, b). These data uncovered distinct molecular features among EC, SC, and MSEC subtypes. Notably, highly proliferating Ki67-positive tumor cells and CD3e-positive lymphocytes were found across all subtypes, with no clear difference among subtypes (Supplementary Fig. 1c).

**Mutational landscapes of NPC.** To further investigate subtype-specific genomic features and therapeutic prognosis, we performed whole-exome sequencing (WES) to study the genomic landscape of NPC subtypes. A total of 2662 somatic mutations including 2306 missense mutations, 191 nonsense mutations, 82 deletions, 31 insertions, and 52 other types of mutations were

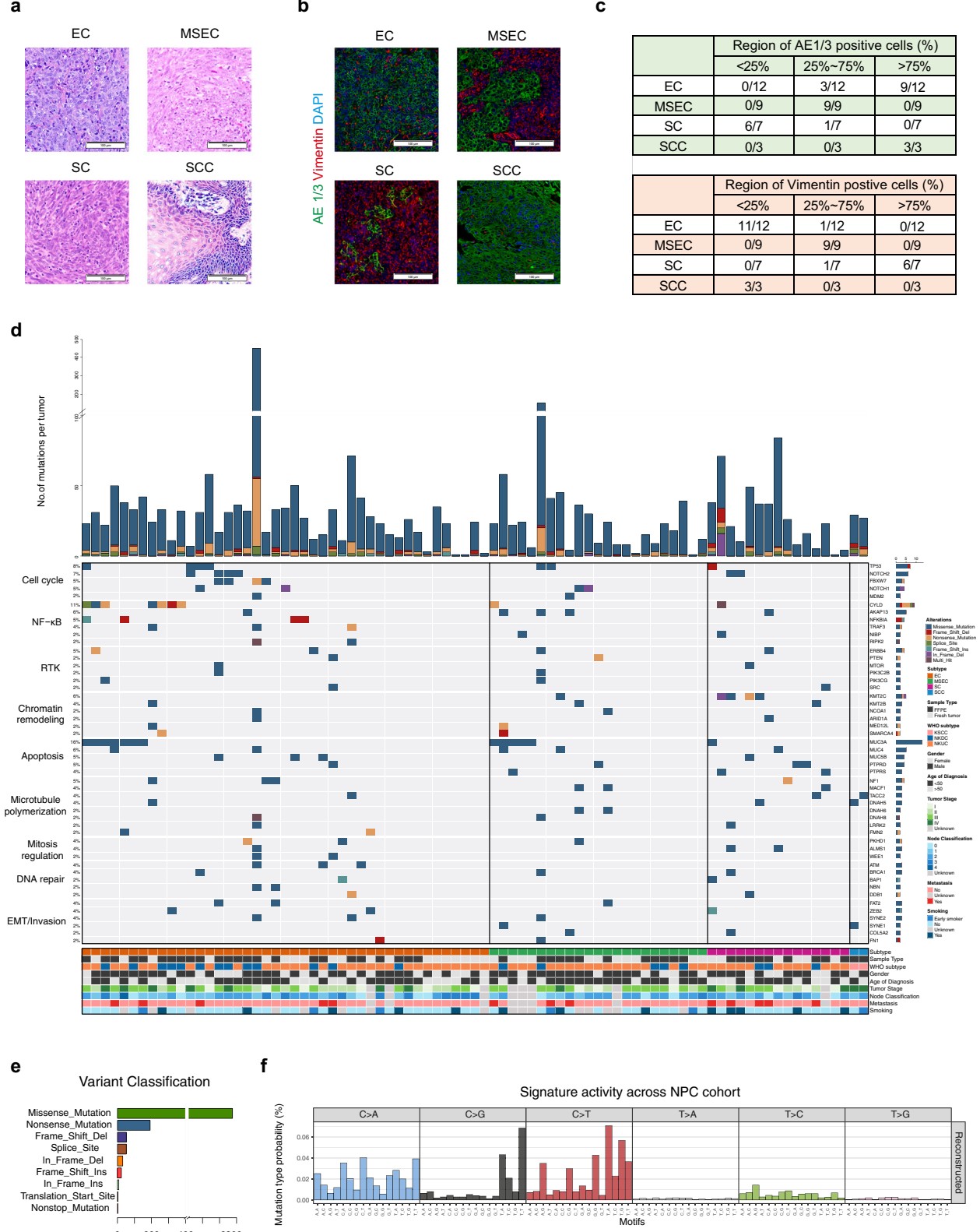

detected from 88 paired tumors by overlapping results from three different callers (MuTect2, Strelka2 and LANCET), and 2148 genes were affected (Fig. 1d, e; Supplementary Fig. 2a; Supplementary Data 3). Validation of candidate mutations with Sanger sequencing showed that a true positive rate of 100% was achieved (Supplementary Fig. 2b; Supplementary Data 4). The somatic mutation rate in NPC is relatively low compared to other types of

cancers, with the somatic mutation rate of less than one per megabase (Supplementary Fig. 2c), which is consistent with previous studies[35–37]. Averagely, we identified 30.3 somatic SNVs per sample. The recurred mutations presented in Fig. 1d revealed the oncogenic drivers of NPC, such as *TP53* (mutational frequency at 8.0%), *CYLD* (10.2%), *KMT2C* (5.7%), *NOTCH2* (6.8%), *NFKBIA* (4.5%), *FBXW7* (4.5%), *ARID1A* (2.3%), *PTEN*

**Fig. 1 Histological and molecular landscape of NPC subtypes. a** Representative histological view of four NPC subtypes. The EC subtype is characterized by round epithelial tumor cells, while the SC subtype is characterized by spindle sarcomatoid tumor cells. The MSEC subtype encompasses both round epithelial and spindle sarcomatoid tumor cells. SCC is a minor subtype of NPC characterized by a keratinizing phenotype. Scale bar, 100 μm. The H&E staining images are representatives of 106 tumors consist of 57 EC subtype, 20 SC subtype, 26 MSEC subtype, and 3 SCC subtype tumors. **b** Immunohistochemical staining of NPC markers. The EC subtype is positive for the epithelial cell marker AE1/3 and negative for the sarcomatoid cell marker vimentin, whereas the SC subtype is positive for the sarcomatoid cell marker vimentin and negative to epithelial cell marker AE1/3. The MSEC subtype exhibited mixed pattern of both epithelial and sarcomatoid cell markers. Scale bar, 100 μm. The immunohistochemical images are representatives of 31 tumors consist of 12 EC subtype, 9 MSEC subtype, 7 SC subtype, and 3 SCC subtype tumors. **c** Summary table of NPC marker expression levels among subtypes. The top panel shows levels of AE1/3 positive cells in EC ($n = 12$), MSEC ($n = 9$), SC ($n = 7$), and SCC ($n = 3$) tumors. The bottom panel presents levels of vimentin-positive cells in EC ($n = 12$), MSEC ($n = 9$), SC ($n = 7$), and SCC ($n = 3$) tumors. The corresponding immunohistochemical images are demonstrated in Fig. 1b and Supplementary Fig. 1a. **d** Top recurrent protein coding SNVs identified from this cohort and represented in pathway wise, including cell cycle, NF-κB signaling, receptor tyrosine kinase (RTK), chromatin remodeling, apoptosis, microtubule polymerization, mitosis regulation, DNA repair and EMT/invasion. SNV mutational rate for each tumor is shown at the top of the panel. Information of subtype, sample type, WHO subtype, gender, age of diagnosis, tumor stage, node classification, metastasis and smoking status are shown at the bottom. Only paired tumor samples ($n = 88$) were assigned for SNV identification. **e** Detailed somatic variation types and amounts discovered from this NPC cohort. A total of 2662 somatic mutations included 2306 missense mutations, 191 nonsense mutations, 47 splice site, 51 frame shift deletion, 31 in frame deletion, 23 frame shift insertion, 8 in frame insertion, and 5 others ($n = 88$ tumors). **f** Overall mutational signature of this NPC cohort revealed C > A and C > T base substitutions are predominant mutational signature in NPC ($n = 88$ tumors).

(2.3%), and *BAP1*(2.3%). The mutation frequencies identified here were comparable with those reported by several previous studies (Supplementary Table 2)[5,8–11], although they were, in general, lower than that of some other cancer types[35]. The recurrently mutated genes aggregated into signaling pathways involving in cell cycle, NF-κB signaling, receptor tyrosine kinase (RTK), chromatin remodeling, apoptosis, microtubule polymerization, mitosis regulation, DNA repair and EMT/invasion (Fig. 1d). Among these top mutated pathways, NF-κB signaling (*CYLD* (mutational frequency at 15.2% in EC), *NFKBIA* (8.7% in EC), *TRAF3* (6.5% in EC), and *RIPK2* (4.3% in EC)), mitosis regulation (*WEE1* (4.3% in EC), *NOTCH2* (8.7% in EC), and *FBXW7* (8.7% in EC)), DNA repair (*ATM* (6.5% in EC) and *NBN/NBS1* (4.3% in EC)), and microtubule polymerization (*FMN2* (4.3% in EC)) exhibited relatively high mutational frequencies in EC subtype, which indicated that they might be the contributive factors to subtype-specific oncogenic mechanisms.

Mutational signatures analysis revealed that C > T base substitutions was the predominant signatures in NPC without obvious subtype difference (Fig. 1f; Supplementary Fig. 2d). The second frequent signature in our NPC cohort was the C > A transition, which was associated with smoking exposure (Supplementary Fig. 2e)[38]. Consistent conclusion was also demonstrated by previous report[39]. Among top frequent COSMIC signatures in NPC, signatures 2 and 13 were related to APOBEC family, signatures 6, 15, 20, and 16 were related to DNA mismatch repair, signature 5 was of unknown aetiology, signatures 4 and 29 were due to tobacco, and signature 1 was associated with methylcytosine (Supplementary Fig. 2f). There was no significant difference on these NPC-related COSMIC mutational signatures among subtypes, but slightly higher APOBEC family and methylcytosine related signatures were observed in EC subtype than MSEC and SC subtypes, whereas tobacco associated signatures were higher in SC subtype than other subtypes (Supplementary Fig. 2g–k).

**Driver pathways and networks revealed by copy number variation (CNV) analysis.** Further investigation on somatic CNVs revealed that frequent chromosomal deletions of Chr. 3p, 9p, 14q, 16q, and amplifications of 3q, 8q, 12p, 12q, and 18q were the major features of NPC across all subtypes (Fig. 2a; Supplementary Fig. 3a), which suggested that such recurrent changes were critical genetic events leading to NPC tumorigenesis. The highest frequency of chromosome 3p deletion, locus of *MST1R* and *BAP1*, indicated that the inactivation of tumor suppressor genes in this

chromosome might be an early event contributing to transformation from nasopharyngeal epithelium to NPC. There were no significant differences on overall chromosomal gain and loss frequencies among EC, MSEC and SC subtypes, except EC vs. SC on chromosomal gain frequencies (Fig. 2b). Age also acted as an important contributive factor during NPC progression. Accumulation of chromosomal abnormality was observed with increasing age, and several unique chromosomal variations presented in patients over 50 years of age, such as chromosomal gain of Chr. 3q and losses of Chr. 16q and 19p (Supplementary Fig. 3b).

Somatic CNVs in gene level were called by Sequenza and CNVkit, and only consensus results shown in both callers were retained for further analysis (Fig. 2c; Supplementary Fig. 4a, b). The detected driver mutational events aggregated into several molecular mechanisms, including defective G1-S checkpoint surveillance (*CDKN2A/B*, *TP53*, and *CCND1*), activated NF-κB signaling (*CYLD*, *TRAF3*, *NFKBIA*, *NLRC5*, *LTBR*, *TNFRSF1A*, *RELA*, *NIBP*, *RELA*, *RIPK2*, *IKBKB*, and *BIRC2/3*), aberrant RTK (*PIK3CA*, *PTEN*, *ERBB3*, *KRAS*, *MET*, *BRAF*, and *MST1R*), and chromatin remodeling (*KMT2C/D*, *BAP1*, *ARID1A*, and *TET1*) (Fig. 2c). The mutation frequencies of these NPC drivers were in consistent with previous studies (Supplementary Tables 3, 4)[5]. They were the most frequently mutated pathways across all subtypes uncovered by CNV analysis.

One of the interesting findings on these top frequent mutations was that macrophage stimulating 1 receptor (*MST1R*) was detected with recurrent deletions in 55.6% of tumors (Fig. 2c). MST1R expression in wild-type and mutant samples was further confirmed by IHC staining (Supplementary Fig. 5a, b). *MST1R* was also known as c-Met-related tyrosine kinase, and normally harbored activation/gain mutations and/or overexpression in other cancer types[40–43]. In addition to playing an oncogenic role as tyrosine kinase to enhance activation of Ras/MAPK and other signaling cascades, *MST1R* also plays a vital function in host defense against viral infection, including Epstein–Barr virus (EBV) and human immunodeficiency virus (HIV)[6,44,45]. Considering the high frequent loss of *MST1R* had no obvious subtype and age preferences (Supplementary Fig. 5d, e), it was suggested that *MST1R* loss may act as an early event of NPC by increasing oncogenic susceptibility associated with EBV infection, although the action of *MST1R* in other types of cancers might be opposite. Consistently, when we applied CRISPR-Cas9 system to knockout *MST1R* in wild-type PDOs, loss of *MST1R* in NPC neither promoted nor decreased organoid growth (Supplementary Fig. 5e, f). Network analysis

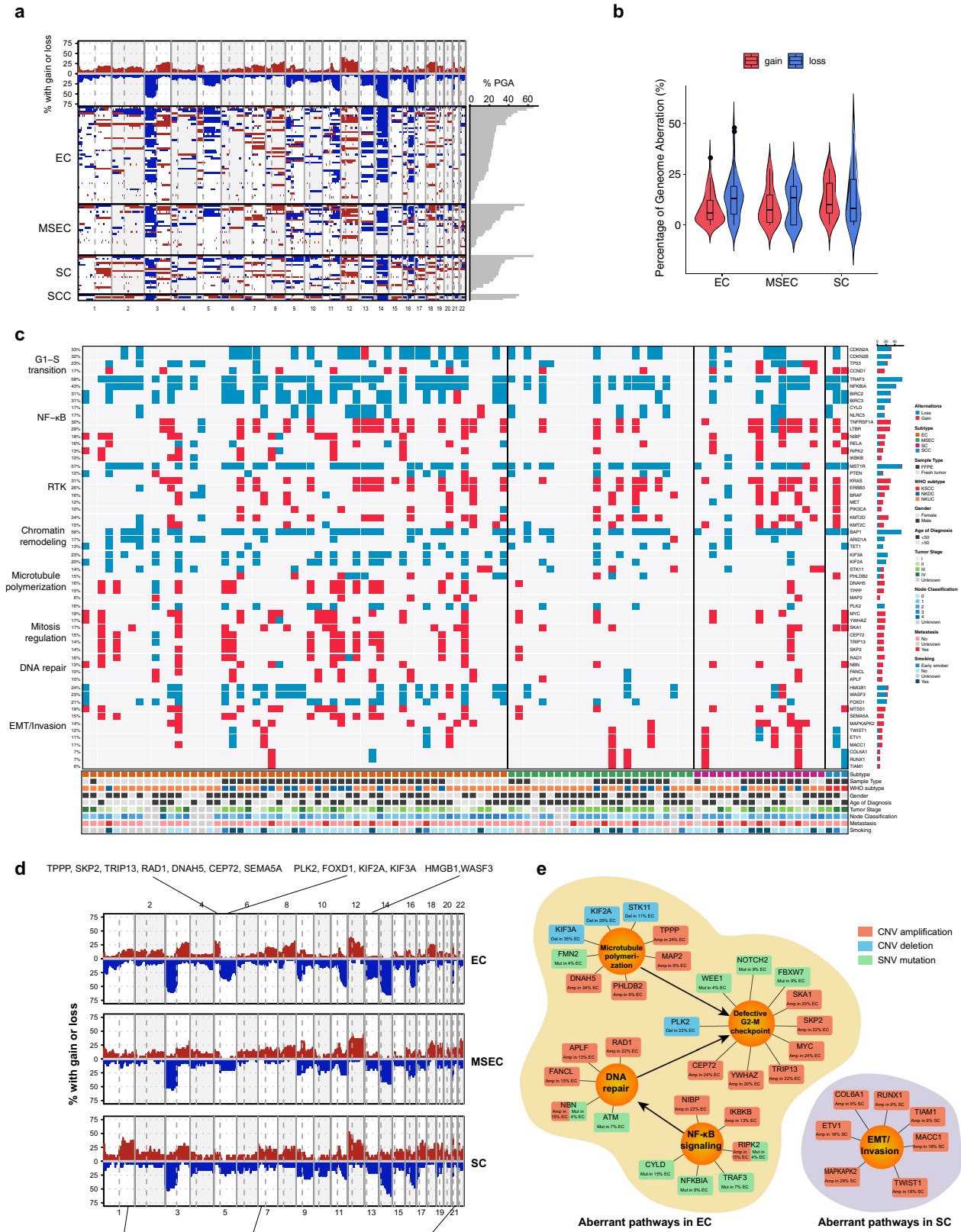

revealed that MST1R had close interaction with 14-3-3, and MST1R/14-3-3 complex was believed to contribute to the NPC susceptibility (Supplementary Fig. 5g)[6].

Since subtypes among NPC exhibited distinct morphological phenotypes (Fig. 1a, b), they should be driven by different oncogenic factors. To obtain further insights into subtype-specific

oncogenic drivers, we next sorted detected somatic CNVs by subtype-specific mutational frequencies and mapped the important subtype-specific mutations to known functional annotations (Fig. 2c, d), finally summarized as a pathway diagram to explain subtype-specific driver mechanisms (Fig. 2e). Chromosomal amplifications of Chr. 1q, 5p, 7p, 15q, 17q, 20q, 21q, 22q and

**Fig. 2 NPC cancer driver pathways and networks revealed by somatic CNVs. a** Global chromosomal gains (red) and deletions (blue) revealing frequent chromosomal deletions in Chr. 3p, 9p, 14q, 16q, and amplification of Chr. 3q, 8q, 12p, 12q, 18q as major feature of NPC ($n = 99$ tumors). **b** Average chromosomal gain (red) and loss (blue) frequencies among different NPC subtypes ($n = 99$ tumors). Violin plots show the kernel probability density of the data at different values. Inside boxplots represent median levels, first and third percentiles. Whiskers indicate 1.5× inter-quartile range (IQR) extending from the hinges. Points above the upper and lower whiskers represent the outliers (>1.5× IQR). Statistical significance was calculated using two-sided $t$-test. No significant subtype-specific difference on either gain or loss was observed ($p > 0.05$), except EC vs. SC on chromosomal gain frequencies ($p = 0.039$). 55 EC, 24 MESC and 17 SC tumors were analyzed. **c** Top frequent CNVs identified from this cohort revealing common and subtype-specific driver mutational pathways of NPC. CNVs involving in G1-S transition, NF-κB signaling, receptor tyrosine kinase (RTK), chromatin remodeling, microtubule polymerization, mitosis regulation, DNA repair, and EMT/invasion were suggested as NPC oncogenic drivers. Information of subtype, sample type, WHO subtype, gender, age of diagnosis, tumor stage, node classification, metastasis, and smoking status are shown at the bottom. 99 tumor samples were used for CNV identification. **d** Subtype-specific chromosomal gains (red) and deletions (blue). Chromosomal amplifications of Chr. 1q, 5p, 7p, 15q, 17q, 20q, 21q, 22q and deletions in Chr. 5q, 13q exhibited significant differences among NPC subtypes. Important subtype-specific CNVs are indicated at top and bottom region. **e** Subtype-specific aberrant cancer driver pathways and networks in EC and SC subtypes. Important signaling pathways, including microtubule polymerization, mitosis regulation, NF-κB signaling, and EMT/invasion, were suggested to contribute to NPC subtype specificity.

deletions in Chr. 5q, 13q exhibited significant differences among NPC subtypes (Fig. 2d; Supplementary Fig. 6). Further gene-level analysis revealed the EC subtype harbored more frequent mutations involved in microtubule polymerization (STK11/LKB1, TPPP, MAP2, PHLDB2/LL5ß, DNAH5, KIF2A, and KIF3A), defective mitotic spindle checkpoint regulation (PLK2, SKA1, SKP2, MYC, TRIP13, CEP72, and YWHAZ), DNA repair (APLF, RAD1, FANCL, and NBN/NBS1) (Fig. 2c–e). Although several mutations related to NF-κB signaling activation were found across all subtypes as listed above, more activators (NIBP, IKBKB, and RIPK2) were shown at higher frequencies in the EC subtype than in MSEC and SC (Fig. 2c). Moreover, several important EMT/invasion promoting CNVs (ETV1, MACC1, MAPKAPK2, COL6A1, RUNX1, TIAM1, and TWIST1) were also detected in SC with relatively high frequencies (Fig. 2c–e), which might account for the invasive phenotype of SC subtype (Fig. 1a–c).

Thus, as revealed by genomics analysis, some somatic CNVs and SNVs with distinct subtype-specific mutational frequencies were exhibited in different NPC subtypes, and they aggregated into several important signaling pathways, including microtubule polymerization, mitosis regulation, NF-κB signaling, EMT/invasion, etc., which might contribute to the distinct histological and molecular features among NPC subtypes.

**NPC organoid model recapitulated morphological signature of NPC subtypes.** To develop a patient-specific therapeutic approach, we performed 3-dimensional (3D) organotypic culture for fresh NPCs by adapting well-developed methods established for other cancer types. Distinguished from earlier organoid culture studies, we incorporated drug sensitivity with genomic features to establish a precision oncology pipeline for integrative pharmacogenomics studies. We successfully generated patient-derived organoid (PDO) models for 40 out of 43 (93%) patients under our optimized conditions, including 23 EC, 10 MSEC, and 7 SC (Supplementary Table 5). Among 40 tumors with PDOs, four organoid lines were derived from two MSEC tumors. In total, we have 42 PDO lines available for further drug treatment study. All established PDOs could readily expanded at least five passages, and later were stored in our living biobank.

Organoids derived from EC subtype NPC commonly showed solid sphere-like structures with smooth surfaces, exhibiting few invading cells found at the surface of organoids (Fig. 3a). In contrast, tumor cells originating from the SC subtype were unable to grow dense spheres under 3D culture conditions, instead, they exhibited a loose discohesive phenotype with spreading spindle cells. To better mimic cell–cell contacts of the tumor microenvironment, we adjusted our culture method slightly, specifically for SC-type organoids. Initially, we allowed cells to aggregate into

dense spheres in low-attachment plates, then the formed spheres were embedded into Matrigel as regular organoid culture. Following this improvement, we clearly observed the matrix-invading capability of spindle tumor cells derived from the SC subtype with extensive spike-like protrusions displayed on the organoids surface (Fig. 3b). Organoids from MSEC subtype usually showed a spindle organoid phenotype similar to SC after stable culture, but smooth solid spheres could still be observed at an early stage (1–2 passages) and were progressively overtaken by spreading of spindle cells during passaging. In general, PDOs well recapitulated the subtype-specific morphological features of the corresponding original tumors. Except for those of the MSEC subtype, PDOs could only preserved the mixed sarcomatoid-epithelial pattern for up to 2 weeks under unified culture conditions. We made further efforts to establish separated cultures for two distinct populations of cells within individual MSEC tumor. After manually separating smooth solid spheres and spindle cells at the first passage of organoid culture from one MSEC tumor, we subsequently established several pairs of organoid lines with EC-type organoids and SC-type organoids (Fig. 3c). EC-type organoids and SC-type organoids derived from MSEC tumors shared similar morphological phenotypes with organoids derived from EC and SC tumors, respectively (Fig. 3a–c) .

Our earlier data revealed that EMT probably serves as an important molecular feature to distinguish subtypes among NPC (Fig. 1b, c), therefore, we stained PDOs with the pan-epithelial marker AE1/3 and the mesenchymal marker vimentin. In consistent with the parental subtype-specific features, EC subtype organoids showed extensive AE1/3 expression (Fig. 3d), while organoids derived from MSEC and SC subtypes were positive for vimentin expression (Fig. 3e). Moreover, PDOs also retained the EBV latency status as evidenced by LMP1 staining on paired tumors and PDOs (Fig. 3f). Further, we conducted transcriptome sequencing for 14 PDOs, including 8 EC, 4 MSEC, and 2 SC PDOs. Unsurprisingly, EMT promoting genes were extensively enriched with high expression in SC-type organoids, including VIM, ZEB1/2, S100A4, FN1, MMP2, and TWIST1/2 (Fig. 6b, d). Distinct expression patterns of EMT genes were observed in PDO63E and PDO63S organoids derived from the same MSEC tumor, suggesting the mixed sarcomatoid-epithelial feature of MSEC (Fig. 6d). These data supported that PDO model could faithfully recapitulate subtype-specific morphological and molecular signatures of NPC and further confirmed the EMT signature as a discriminative molecular feature for NPC subtypes.

To examine whether the PDOs preserved genomic features of parental tumors, WES was performed on 15 pairs of tumors and PDOs. Genome-wide CNV analysis demonstrated that chromosomal gains and losses of parental tumors were well retained in

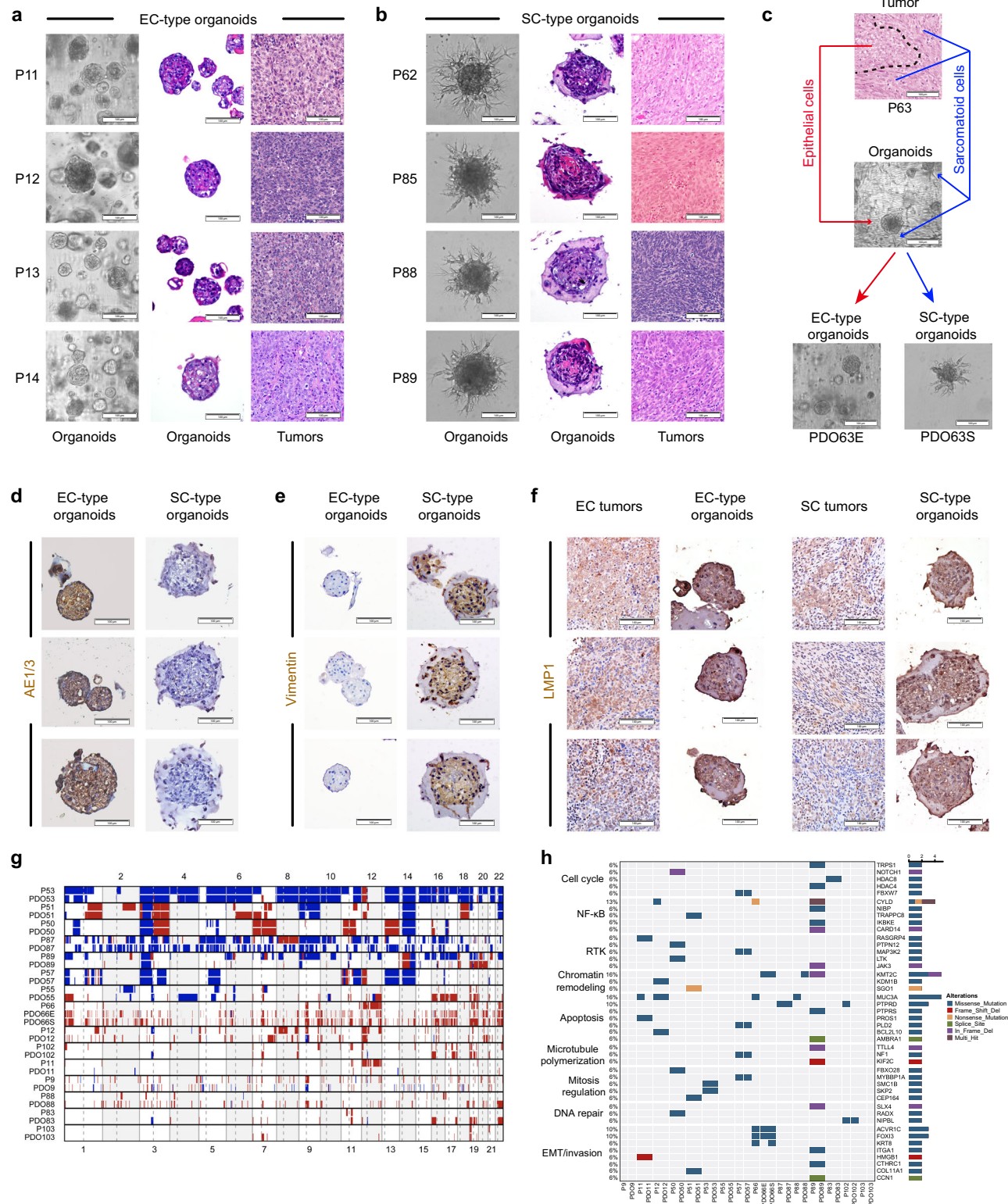

PDOs for the majority of pairs (Fig. 3g). Similarly, paired tumors and PDOs maintained consistent mutational signature profiles (Supplementary Fig. 7a,b). Furthermore, the cancer related gene mutations in parental tumors, thought to be tumor-specific drivers, were preserved in most paired PDOs (Fig. 3h). Since there were two morphologically distinct cell populations observed in same MSEC tumor, it was of interest to explore whether they originated from same clone or polyclone. WES analysis on paired

P66, PDO66E, and PDO66S indicated that EC-type PDO66E share a number of common mutations with SC-type PDO66S. This suggested that the heterogeneity of two distinct populations within MSEC might originate from the same cell during tumor evolution (Supplementary Fig. 7c).

Altogether, our data demonstrated that NPC organoid model faithfully recapitulated the signatures of parental tumors, including morphological, molecular and genomic features. PDOs

**Fig. 3 Patient-derived 3D organoids recapitulate features of parental tumors. a, b** Representative bright field microscopy and H&E staining images of EC-type patient-derived organoids (PDOs) and matched primary tumors from the EC subtype (**a**); SC-type PDOs and matched primary tumors from the SC and MSEC subtypes (**b**); **c** Organoids derived from MSEC subtype tumor (P63) recapitulated the mixed sarcomatoid-epithelial pattern at early passages, later developed two distinct EC-type (PDO63E) and SC-type organoids (PDO63S). **d** Immunohistochemical staining of AE1/3 on PDOs. EC-type organoids were positive to AE1/3 expression, while organoids derived from MSEC and SC subtypes were AE1/3 negative. **e** Immunohistochemical staining of vimentin on PDOs. Organoids derived from MSEC and SC subtypes were positive to vimentin expression, while EC-type organoids were vimentin negative. **f** Immunohistochemical staining of LMP1 on PDOs and parental tumors. PDOs retained EBV infection status. Scale bars (**a–f**) are 100 μm. The organoid images (**a–f**) are representatives of 6 EC-type and 6 SC-type organoid lines. **g** Comparison of the CNV landscape in paired tumors and PDOs. PDOs faithfully preserved chromosomal gains and losses of parental tumors ($n = 15$ pairs). **h** Oncoplot comparing the somatic SNVs between PDOs and parental tumors. PDOs maintained important mutations of parental tumors ($n = 15$ pairs).

may serve as good proxies to evaluate treatment responses of potential therapeutics for individual patients.

**Subtype-specific drug response revealed by PDO-based drug screening.** Next, we conducted drug screening to identify effective therapeutic candidates using PDOs. 42 PDOs were successfully screened with a drug library containing 48 drugs, including several first-line chemotherapeutic drugs for NPC, such as docetaxel, paclitaxel and bleomycin, in real-time at passages 1–3 within 2–3 weeks after biopsy collection (Supplementary Fig. 8; Supplementary Data 5). Each drug was screened at six concentrations from 0.08 to 20 μM at 3-fold dilutions, and the drug sensitivity was represented by an inhibitory concentration of 50% (IC50). In general, none of the PDOs exhibited identical drug responses, reflecting individual differences, although subtype-specific responses were noticed to some drugs (Fig. 4a; Supplementary Fig. 8). For example, organoids derived from SC and MSEC, except EC-type PDO63E and PDO66E, were sensitive to microtubule inhibitors (MTi), while organoids derived from EC exhibited resistance to MTi at various levels (Fig. 4a). Some representative drug responses were repeated and shown in Fig. 4b–e. As mentioned earlier, PDO63E and PDO63S were derived from a single MSEC tumor and exhibited distinct EC-type and SC-type morphology, respectively. EC-type PDO63E organoids showed the same resistance response to MTi as EC PDOs, while SC-type PDO63S organoids were sensitive to MTi, similar to SC PDOs (Fig. 4b, c). On the other hand, for EGFR inhibitors (EGFRi), 22 of 23 PDOs derived from EC and EC-type PDO63E were highly sensitive, while all organoids derived from SC and 9 out of 12 PDOs derived from MSEC showed resistance to EGFRi. Of note, MSEC PDOs displayed diverse responses to EGFRi (Fig. 4a), indicating that MSEC contains both SC and EC features. In general, EC subtype PDOs were sensitive to EGFRi and resistant to MTi, while opposite phenotypes were shown in PDOs derived from SC and over half of MSEC.

To understand the regulating key molecular signaling responsible for this distinct responsiveness, we firstly treated several resistant and sensitive cell lines with gefitinib, a known EGFRi, and examined key EGFR downstream targets by Western blot analysis. Our data revealed that gefitinib significantly reduced both p-AKT and p-ERK in sensitive PDOs PDO63E and PDO4, p-STAT3 also showed a reduction in PDO4 but not in PDO63E (Fig. 4f), and in the other four sensitive PDOs, three detected downstream signaling pathways of EGFR exhibited blocked phosphorylation to varying degrees, which were seldom found in resistant PDOs (Fig. 4f; Supplementary Fig. 9a). In general, ERK and AKT phosphorylation were blocked in all tested sensitive PDOs under gefitinib treatment, except PDO14, which did not show a decrease in AKT phosphorylation. Combinational treatment with AKT inhibitor MK2206, ERK inhibitor GDC0994, STAT3 inhibitor C188-9 and gefitinib could overcome the EGFRi resistance in PDO63S (Supplementary Fig. 9, c–f). These data

indicate that the MAPK and PI3K-AKT signaling pathways might play crucial roles in the treatment efficiency of EGFRi in NPC.

Furthermore, MTi (docetaxel) treatment increased the expression of mitotic checkpoint proteins (cyclin A and cyclin B), degraded the antiapoptotic protein Bcl2 through phosphorylation, and activated proapoptotic proteins such as Bax and cleaved caspase-3 (CASP3) in sensitive PDOs, while no significant changes in these proteins were observed in resistant PDOs (Fig. 4g; Supplementary Fig. 9b). Cyclin B is expressed during the late G2 and early M phases of the cell cycle and drives cells into the M phase. Docetaxel treatment blocked the formation of the spindle, leading to accumulation of cells in M phase, which was consistent with the increased level of cyclin B. Suppression of the apoptosis-blocking function through phosphorylation and degradation of Bcl2 was one of the important established MTi-affecting mechanisms of tumor cells[46,47]. In all six sensitive PDOs, docetaxel strongly induced Bcl2 degradation and activated the apoptotic death marker Bax, which indicated the importance of Bcl2-mediated antiapoptosis in NPC. In the resistant PDOs, none of these events were occurred (Fig. 4g; Supplementary Fig. 9b), which was consistent with their resistance to MTi. As representative MSEC-derived PDOs, EC-type PDO63E, and SC-type PDO63S exhibited opposite responses to MTi. SC-type PDO63S responded well to MTi treatment with mitotic checkpoint protein blockage and Bcl2 degradation, followed by induction of apoptosis activation, whereas EC-type PDO63E showed no significant changes in these signaling pathways. Together with their distinct responses to EGFRi mentioned earlier, these results further demonstrated the heterogeneity of MSEC subtype tumor, at least two distinct tumor populations were occurred in one tumor: one population shared a similar drug response pattern similar to that of EC, and the other shared similarity to SC.

**Individualized synergistic therapeutic combinations identified from PDO-based chemoradiotherapy (CRT) screening.** In addition to chemotherapy, which is the widely accepted regimen for advanced NPC with metastasis, chemoradiotherapy (CRT) is recognized as the standard therapy for stage II and more advanced NPC, yet only limited chemical drugs were approved for NPC treatment. To investigate whether PDOs could be used for identifying more drugs for CRT with enhanced therapeutic effect towards individual patients, we conducted sensitivity tests for radiotherapy (RT) and CRT. Through treatment on eight PDO lines encompassing three major NPC subtypes with increased ionizing radiation (IR) dose from 2, 4, 6, 8, and 10 Gy, we found that the tested EC subtype PDOs were more sensitive to RT than the SC and MSEC subtype PDOs (Fig. 5a, b). Our RT results reflected the previously reported clinical outcome that SC subtype patients had an ~20% lower 5-year OS rate than EC subtype patients[13].

For recurrent and/or metastatic NPC, CRT is the standard-of-care treatment. We next examined the efficacy of commonly using CRT combinations on 20 PDOs derived from individual

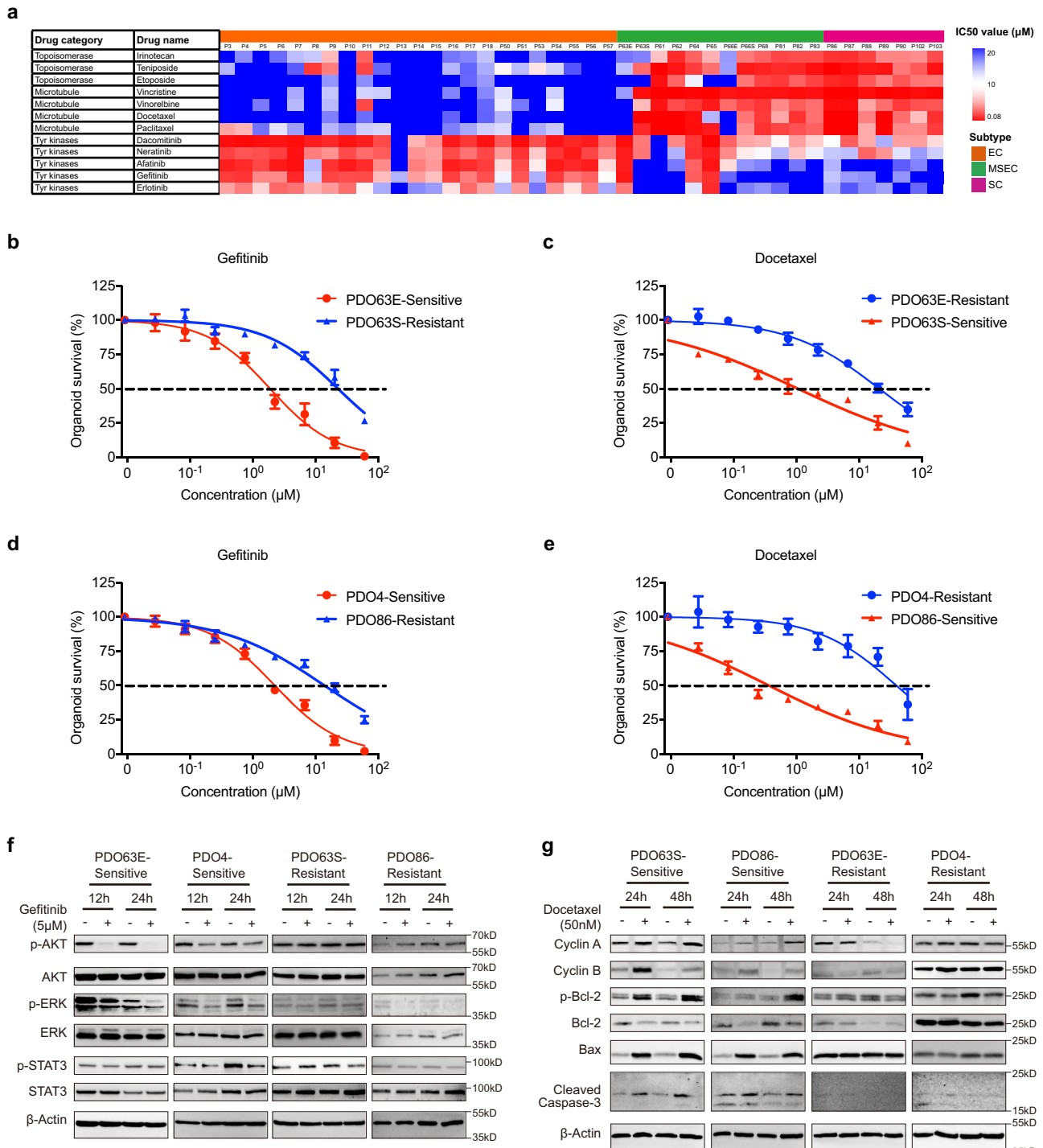

**Fig. 4 Drug treatment on patient-derived organoids (PDOs). a** Heatmap showing subtype-specific drug responses (IC50 values) of topoisomerase (irinotecan, teniposide and etoposide), microtubule (vincristine, vinorelbine, docetaxel and paclitaxel) and EGFR (dacomitinib, neratinib, afatinib, gefitinib, and erlotinib) inhibitors on 42 NPC PDOs. **b** Representative gefitinib response curves on sensitive MSEC-derived organoids PDO63E and resistant MSEC-derived organoids PDO63S. **c** Representative docetaxel response curves on sensitive MSEC-derived organoids PDO63S and resistant MSEC-derived organoids PDO63E. **d** Representative gefitinib response curves on sensitive EC-derived organoids PDO4 and resistant SC-derived organoids PDO86. **e** Representative docetaxel response curves on sensitive SC-derived organoids PDO86 and resistant EC-derived organoids PDO4. Sensitive drug response curves (**b**–**e**) are presented with red color, and resistant curves (**b**–**e**) are labelled with blue color. Data points (**b**–**e**) represent mean % viability relative to control ± SEM, for $n = 3$ biological replicates. Graphs (**b**–**e**) are representatives of 3 independent experiments. **f**, **g** Western blot analysis on EGFR downstream proteins (**f**) of gefitinib sensitive organoids PDO63E and PDO4, and resistant organoids PDO63S and PDO86; Mitotic checkpoint and apoptotic proteins (**g**) of docetaxel-sensitive organoids PDO63S and PDO86, and resistant organoids PDO63E and PDO4. Blots are representatives of two independent repeats. Western blot analysis demonstrated by more PDO samples are shown in supplementary Fig. 9a,b.

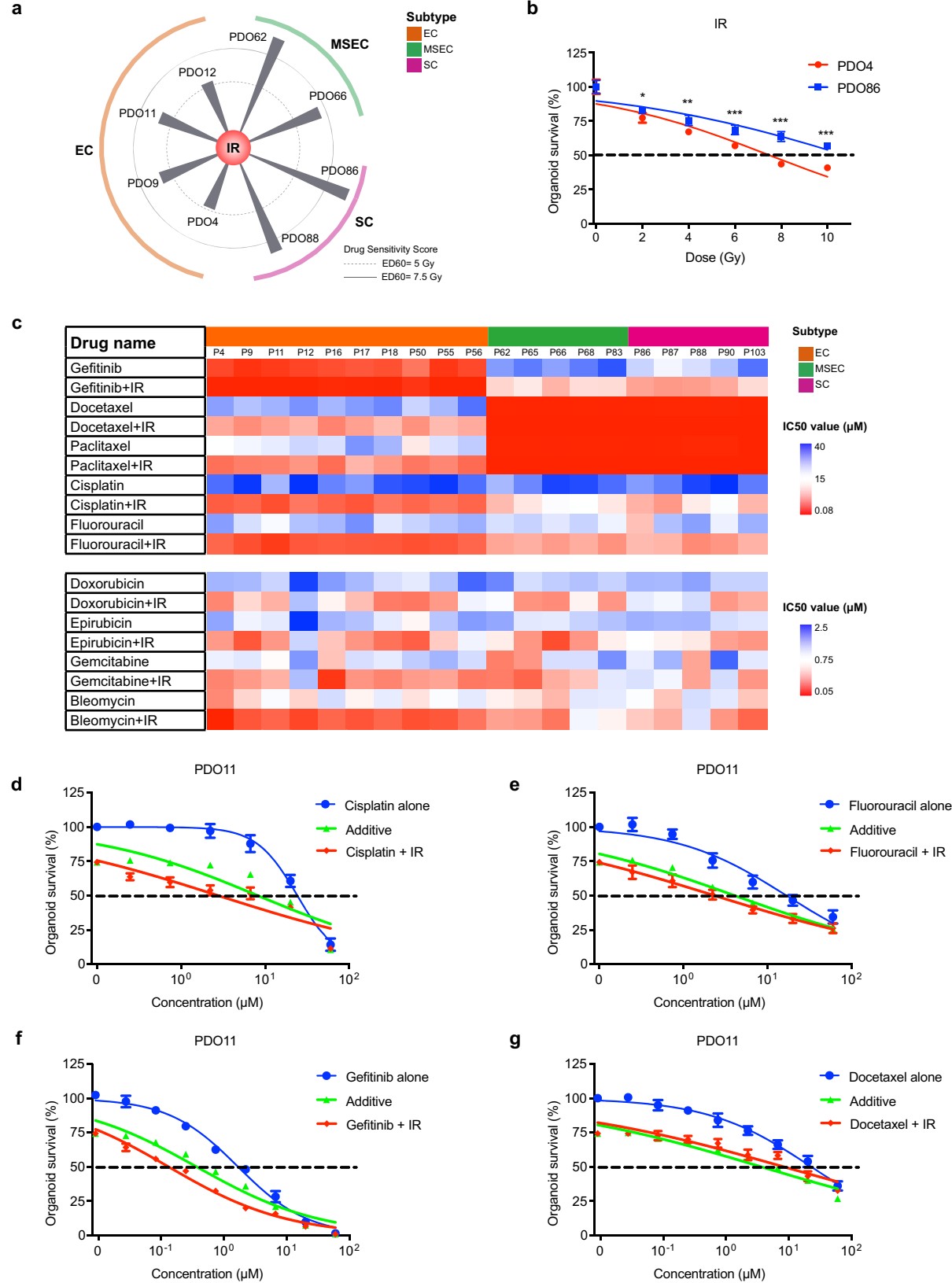

NPCs (Fig. 5c; Supplementary Data 6). In general, CRT exhibited better sensitivities on NPC PDOs than chemotherapeutics alone, especially the most commonly used CRT regimens, RT + cisplatin and RT + fluorouracil (Fig. 5d,e), which generally reflected the clinical treatment outcomes[13,21,22]. When comparing efficacy among NPC subtypes, CRT demonstrated better

sensitivities on EC-type organoids than SC-type organoids for the majority of cases, including CRT regimens combined with cisplatin and fluorouracil (Fig. 5c). These results were consistent with the previously reported clinical outcome that CRT contributed to longer survival time for EC subtype patients than that for SC subtype patients[13].

**Fig. 5 Chemoradiotherapy (CRT) screening identified combinations with enhanced therapeutic effect. a** Radiotherapy responses among PDOs derived from EC, MSEC, and SC subtypes. EC subtype PDOs were more sensitive to RT than the SC and MSEC subtype PDOs. **b** Representative IR response curves on EC-derived organoids PDO4 and SC-derived organoids PDO86. Sensitive drug response curve is presented with red color, and resistant curve is labelled with blue color. Data points represent mean % viability relative to control ± SEM, for $n = 6$ biological replicates. Statistical significance was calculated using two-sided $t$-test, with $*p \leq 0.05$, $**p \leq 0.01$, $***p \leq 0.001$. Graphs are representatives of three independent experiments. **c** Heatmap showing responses (IC50 values) of commonly used CRT regimens on 20 NPC PDOs. CRT generally exhibited better sensitivities on NPC PDOs than chemotherapeutics alone. **d** Representative dose response curves of the PDO11 treated with cisplatin alone and cisplatin combined with IR at 4 Gy. Good synergy was exhibited by combinational treatment of cisplatin and IR on PDO11. **e** Representative dose response curves of the PDO11 treated with fluorouracil alone and fluorouracil combined with IR at 4 Gy. Fluorouracil and IR combination exhibited drug synergy on PDO11. **f** Representative dose response curves of the PDO11 treated with gefitinib alone and gefitinib combined with IR at 4 Gy. Great synergistic effect was also demonstrated by combinational treatment of gefitinib and IR on PDO11. **g** Representative dose response curves of the PDO11 treated with docetaxel alone and docetaxel combined with IR at 4 Gy. No synergy was observed on PDO11 with combinational treatment of docetaxel and IR. The single chemotherapy curves (**d–g**) are presented with blue color, the combinational treatment curves (**d–g**) are labelled with red color, the additive curves (**d–g**) are marked with green color. The additive curves (**d–g**) represent the estimated combined effect of chemotherapy and IR being equal to the sum of their separate effects. If the combinational treatment curve (red) was lower than the additive curve (green), the synergistic effect is demonstrated. Data points (**d–g**) represent mean % viability relative to control ± SEM, for $n = 3$ biological replicates. Graphs (**d–g**) are representatives of three independent experiments.

To identify more drugs that could sensitize PDOs to RT, we further screened 48 anticancer drugs on eight PDOs combined with IR at a dose of approximately IC20-30 (4 Gy) (Supplementary Data 7). One of the most striking difference was demonstrated by CRT combinations with EGFRi on organoids derived from EC subtype (Fig. 5b; Supplementary Fig. 10a). For 4 tested EC subtype PDO lines, CRT exhibited synergistically better inhibitory effect than EGFRi treatment alone, with the average IC50 value fold increases at 7.09, 5.87, 6.01, and 8.17 in each of these PDOs, respectively (Fig. 5f; Supplementary Fig. 10a). Since EGFRi was identified as EC subtype-sensitizing drug, as demonstrated by our earlier data (Fig. 4a), our CRT screening results further suggested that the EGFRi and RT combination could be a good clinical CRT choice to achieve synergistic killing effect for EC subtype patients. In addition, an improvement on MTi efficacy was also found when combining with RT, but there was no observation on drug synergy (Fig. 5c, g).

Furthermore, a few other CRT combinations with enhanced sensitivity were also revealed by our screening data, such as combinations with arsenic trioxide (Supplementary Fig. 10b), olaparib (Supplementary Fig. 10c), vorinostat (Supplementary Fig. 10d), sunitinib (Supplementary Fig. 10e), and zoledronate (Supplementary Fig. 10f). Thus, our analysis uncovered the subtype-related and patient-specific responses to RT and CRT and potentially provided more choice of drugs for CRT to treat patients suffering from advanced or metastatic stage cancer.

**Transcriptome analysis revealed genomic driver mechanisms of NPC subtypes.** Transcriptome sequencing was performed for 14 PDOs and subsequently assigned for unsupervised clustering by principal component analysis (PCA) to evaluate the overall transcriptional pattern of the samples. The results clearly demonstrated that organoids derived from the same subtypes were well clustered into the corresponding subtypes (Fig. 6a), indicating that PDOs maintained their subtype differences at the transcriptional level. As revealed by our earlier genomic sequencing analysis, EC exhibited distinct driver mechanisms with SC and MSEC, including microtubule polymerization, G2-M checkpoint regulation, NF-κB signaling, and EMT. We wondered whether these subtype-specific genomic mutational pathways could be converted to functional transcriptional signaling. The unbiased geneset enrichment analysis (GSEA) was performed to investigate top differentiated pathways between EC- and SC-type PDOs (absolute NES value > 1, $p$-value < 0.05, and FDR value < 0.05) (Fig. 6b). Significantly differentiated expressed genes were filtered by DESeq2 and limma ($\log_2$ fold change > 1, $p$-value < 0.05, FDR (padj) value < 0.05) (Fig. 6c; Supplementary Fig. 11a–d;

Supplementary Data 8), then were presented in heatmap (Fig. 6d). Microtubule-targeting drugs associated mechanisms, including microtubule signaling, mitotic cell cycle, apoptosis, NF-κB signaling, and androgen response, were among the top enriched pathways in EC-type organoids (Fig. 6b, Supplementary Fig. 12a-l)[48–50]. Meanwhile, multiple EGFR signaling related pathways and cellular response to radiation were also among top differentiated pathways (Fig. 6b; Supplementary Figs. 12a-l, 13a-i), which might contribute to the subtype-differentiated responses to EGFR inhibitors and RT. EMT and invasion related pathways, including EMT, extracellular matrix and response to TGF-β, were enriched in SC-type organoids (Fig. 6b, Supplementary Fig. 13a-c). Of note, differentially expressed pathway genes were also present between PDO63E and PDO63S (Fig. 6d; Supplementary Fig. 13j), which was consistent with earlier results on their drug response and protein expression differences, and further demonstrated the heterogeneity feature of MSEC. In addition, the stemness score was calculated for each NPC subtype PDOs, and the results indicated that EC subtype PDOs maintained high stemness properties, whereas the opposite phenotype was shown in SC subtype PDOs (Supplementary Fig. 13k). Distinctly to EC and SC, a diverse stemness distribution was observed in MSEC PDOs (Supplementary Fig. 13k). Moreover, the immune score analysis was also performed to determine whether there were subtype differences among immune cell types, and the results indicated no obvious differences in CD4 + T cells, CD8 + T cells, B cells, or macrophages among the three subtypes (Supplementary Fig. 13l)

**Pharmacogenomics-based precision medicine.** During past decade, next-generation sequencing has been extensively applied for precision oncology and has made great progress in matching targeted therapy with predictive gene signatures. However, several essential limitations have also come to surface, including the lack of identified druggable mutations for the majority of patient in current clinical practice and the reliability of drug prediction value, which was controversial[51]. By following conventional practice focusing on targetable mutations identification, we performed genomics sequencing for tumors of individual patients, but unfortunately, well-known drug sensitizing hotspot mutations were seldom found in the majority of NPC tumors as expected[51], for example, the frequency of EGFR SNVs was only 0.94% (1/106) in our NPC cohort (Supplementary Data 3), which kept us from directly identifying corresponding targeted drugs.

To better deliver precision medicine to NPC patients, we next proposed to develop a pharmacogenomics-based precision medicine (PBPM) approach that integrated genomics/transcriptomics and drug tests to better provide faithful precision

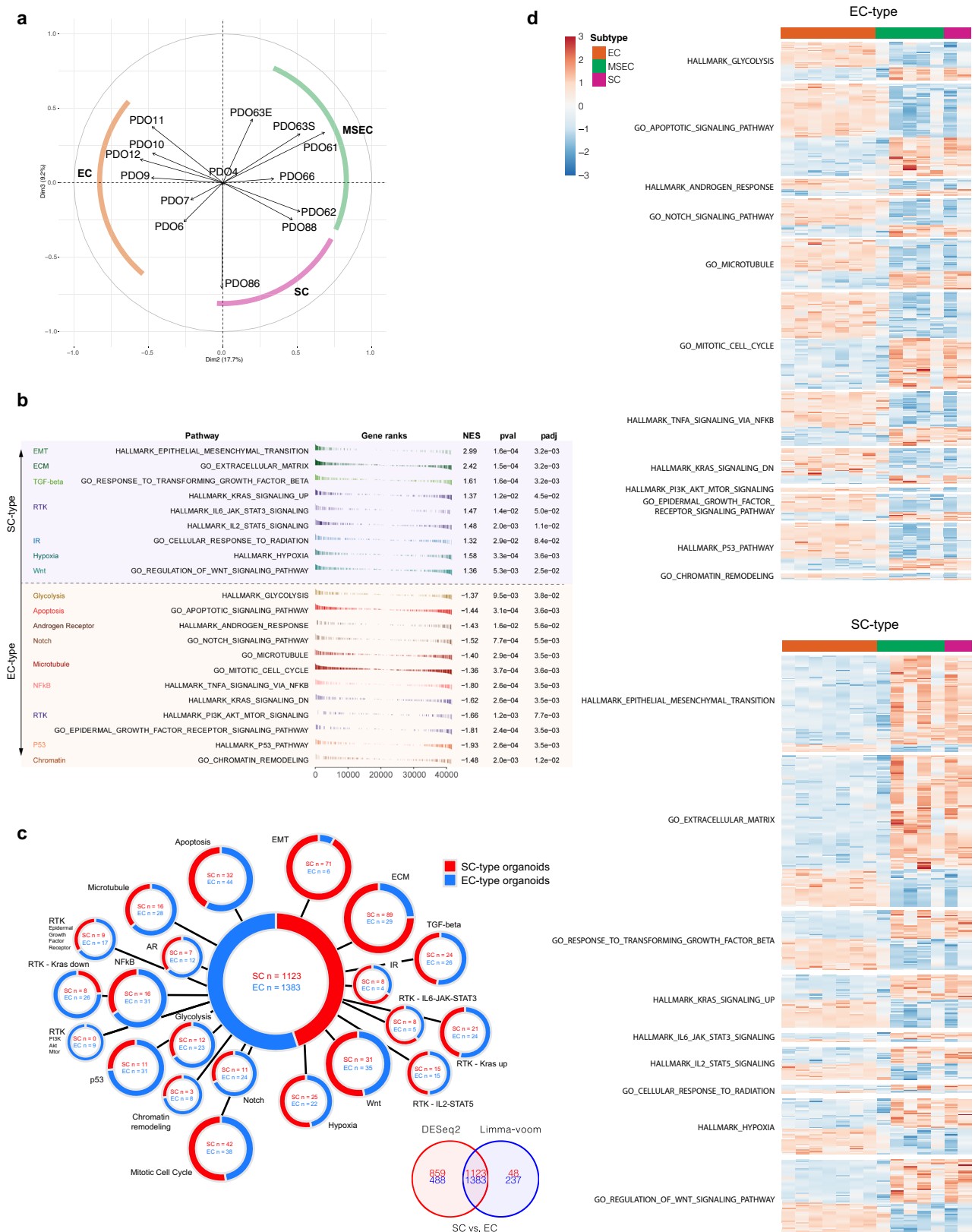

**Fig. 6 Subtype-specific driver pathways revealed by transcriptional analysis. a** Unsupervised clustering by principal component analysis (PCA) on sample-wise PDO transcriptome. Organoids derived from the same subtypes were well clustered into the corresponding subtypes. **b** Top differentiated pathways between EC-type and SC-type PDOs according to GSEA enrichment analysis. **c** Upregulated signature genes in EC-type organoids (blue) and in SC-type organoids (red) overlapped by DESeq2 and Limma-voom. **d** Heatmap showing expression of differentiated signature genes between EC-type and SC-type organoids.

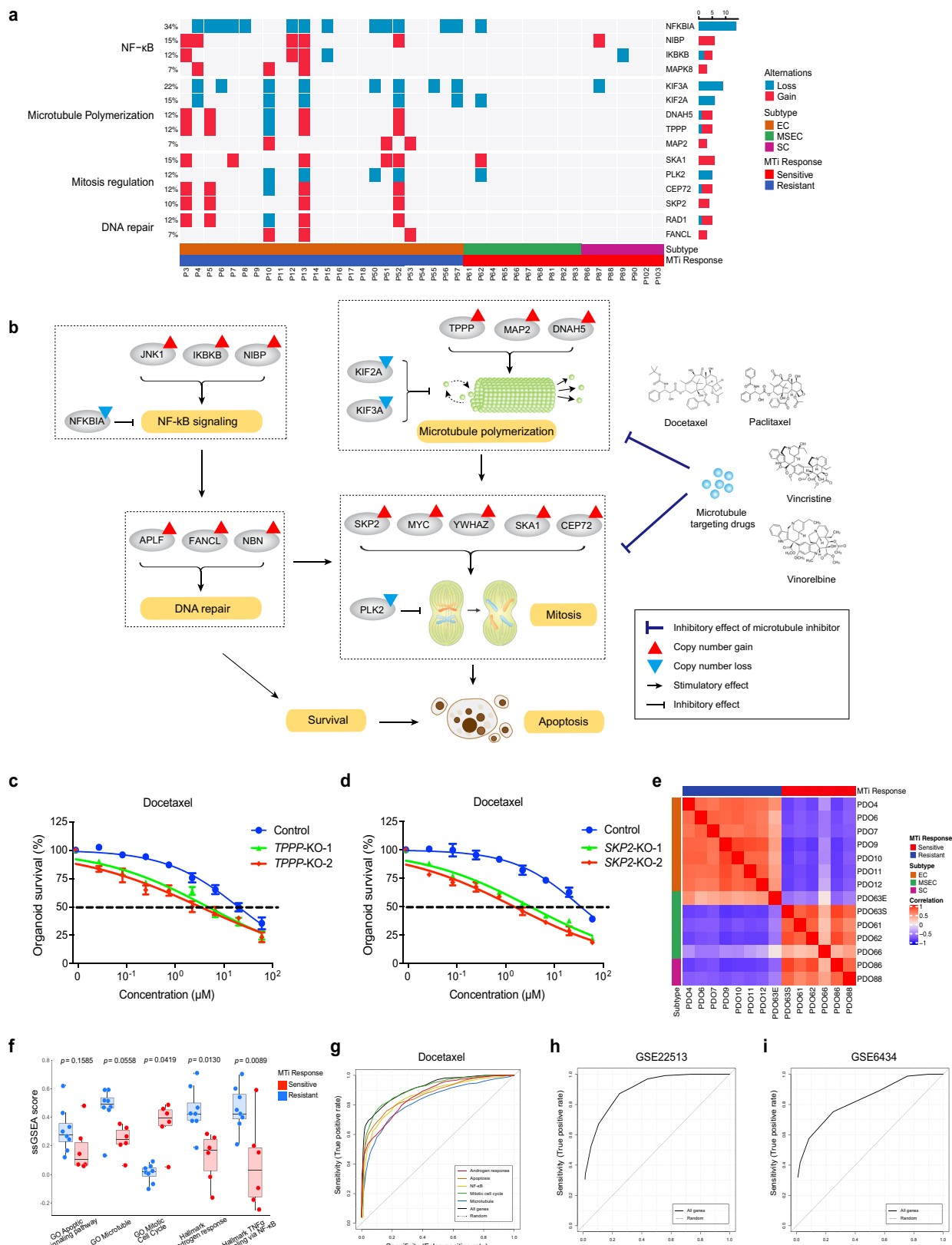

treatment guidance. On the drug test side, we applied PDOs as the proxies to examine drug responsiveness for individual patients and found that many EC subtype PDOs were resistant to MTi, while opposite response of MTi was observed in SC and MSEC subtype derived organoids (Figs. 4a, 7a). The results indicated that some subtype-associated gene signatures might

contribute to the subtype-specific MTi responses, thus uncovering these gene signatures might be valuable for predicting MTi sensitivity. Next, we incorporated genomic variations of all protein coding genes into our analysis, in addition to only druggable mutations. By grouping samples with MTi sensitivity on PDOs, we revealed that consistent subtype-specific mutations

**Fig. 7 Pharmacogenomics-based precision medicine. a** Subtype-specific CNVs illustrating microtubule inhibitor response mechanism involving in NF-κB signaling, microtubule polymerization, mitosis regulation and DNA repair. **b** Microtubule inhibitor resistance in P13 associated with CNV involving in mitosis regulation, microtubule polymerization, NF-κB signaling and DNA repair. **c** TPPP knockout in PDO9 overcome docetaxel resistance. **d** SKP2 knockout in PDO4 overcome docetaxel resistance. Drug response curves (**c**, **d**) are presented with blue color for nontargeting control, green and red color for gene knockout groups. Data points (**c**, **d**) represent mean % viability relative to control ± SEM, for $n = 3$ biological replicates. Graphs (**c**, **d**) are representatives of three independent experiments. **e** Correlation heatmap of NPC subtypes versus docetaxel response based on 105 signature genes from five pathways essential to drug mechanism, including GO apoptotic signaling pathway, GO Microtubule, GO Mitotic cell cycle, Hallmark Androgen response, and Hallmark TNFα signaling via NF-κB. **f** Boxplots representing ssGSEA scores of docetaxel-sensitive and resistant PDOs on five key drug mechanism-related pathways. Boxplots represent median levels, first and third percentiles. Whiskers indicate 1.5× inter-quartile range (IQR) extending from the hinges. Statistical significance was calculated using two-sided $t$-test, the $p$ values regarding comparison between sensitive and resistant groups for each pathway are presented. $n = 14$, including six sensitive samples and eight resistant samples. **g** Receiver operating characteristic (ROC) curves illustrating estimated docetaxel prediction accuracy using signature genesets of five key correlated pathways, including all genes (black, area under curve (AUC) = 0.918), GO apoptotic signaling pathway (orange, AUC = 0.876), GO Microtubule (blue, AUC = 0.83), GO Mitotic cell cycle (green, AUC = 0.914), Hallmark Androgen response (red, AUC = 0.866), Hallmark TNFα signaling via NF-kB (yellow, AUC = 0.864). **h** ROC curve illustrating estimated paclitaxel prediction accuracy using all signature genes from five key pathways (black, AUC = 0.914) on GSE22513 dataset of human breast cancer patients. **i** ROC curve illustrating estimated docetaxel prediction accuracy using all signature genes from five key pathways (black, AUC = 0.841) on GSE6434 dataset of human breast cancer patients.

pattern was potentially involved in microtubule-targeting drug responses (Fig. 7a). Microtubule-targeting drugs were known to disrupt microtubule dynamics, invoke the mitotic checkpoint, lead to cell cycle arrest, and subsequently induce apoptosis[48–50]. Thus, activation of key mutations involved in antagonizing these processes may help tumor gain resistance to MTi. Of note, we observed an obvious positive correlation between EC-specific MTi resistance and EC-specific genomic activations of microtubule polymerization, mitosis regulation, NF-κB and DNA repair (Fig. 7a). A representative personalized case was demonstrated with P13, which help to illustrate the intrinsic correlation mechanism (Fig. 7b). To further validate these mutations as predictive signatures for MTi response, we applied CRISPR-Cas9 to knockout TPPP (tubulin polymerization-promoting protein), MAP2 (microtubule-associated protein 2), and SKP2 (S-phase kinase-associated protein 2) in PDO9 and PDO4, respectively (Supplementary Fig. 14a–c; Supplementary Data 9). TPPP, MAP2, and SKP2 knockout organoids showed sensitive responses to docetaxel treatment when comparing to parental wild-type organoids (Fig. 7c, d; Supplementary Fig. 14d), indicating mutations involving in microtubule polymerization and mitosis regulation could serve as potential predictive signatures for MTi response in NPC.

Another personalized case was demonstrated with P86, which acquired EGFRi resistance. As shown in Supplementary Fig. 15a, copy number gains of MET and KRAS were observed in P86 tumors. Activation of oncogenic MET and KRAS potentially induced EGFRi resistance through bypassing conventional EGFR signaling. To further confirm their contributive effect on EGFRi resistance, we applied crizotinib and vorinostat to perform combinational inhibition together with gefitinib in PDO86 (Supplementary Fig. 15c-e). Crizotinib is a specific inhibitor of c-MET, and HDAC inhibitor vorinostat was previously reported to overcome EGFRi resistance by blocking KRAS[52–54]. The result showed that after dual-blocking of c-MET and KRAS, EGFRi resistance in PDO86 was significantly overcome (Supplementary Fig. 15e), indicating that copy number gains of MET and KRAS might contributed to EGFRi resistance in PDO86.

Based on our previous practice on genomics driven PBPM strategy, although as many as all protein coding mutations were involved into analysis, only a portion of patients could be identified with the discriminative genomic signatures associated to drug response and got the benefit. Integrating transcriptomics with drug response seems to be another good option to develop PBPM. As shown in Fig. 6b–d, microtubule signaling, mitotic cell cycle, apoptosis, NF-κB signaling, and androgen response, were

among top subtype-differentiated pathways. They were reported to contribute to MTi response mechanisms[48–50]. We assembled gene cohorts containing signature genes involved in these pathways with differential expression patterns between EC and SC/MSEC subtypes, and these genesets was further applied to analyze the correlation between subtypes and MTi response (Supplementary Data 8). EC subtype PDOs showed a positive correlation with the docetaxel resistance, while SC and MSEC subtype PDOs exhibited a positive correlation with the docetaxel sensitivity (Fig. 7e). We next examined the impact of individual subtype-differentiated pathways on discriminating the MTi responses. The results demonstrated that individual pathways could well distinguish docetaxel-sensitive and docetaxel-resistant PDOs (Fig. 7f). This data led us to believe that these five subtype-differentiating pathway signatures were the candidate contributive factors for the NPC subtype-specific MTi response. To further validate our hypotheses as well as test the drug prediction value of these pathway signatures, we download public data of both transcriptome sequencing and MTi treatment results from the Genomics of Drug Sensitivity in Cancer (GDSC) database[55]. After sorting MTi-sensitive and MTi-resistant cell lines with the bottom 10% and top 10% IC50 values of docetaxel, paclitaxel, vinorelbine, and vincristine, we applied our pathway signatures to GDSC data to perform receiver operating characteristic (ROC) analysis. As shown in Fig. 7g, the individual pathway signature obtained area under curve (AUC) values ranging from 0.83 to 0.914 for predicting docetaxel sensitivity, indicating significant contributive impacts of these pathway signatures on MTi sensitivity, while the predictive AUC value of the random geneset was 0.5. Of note, when combining all 5 pathway signatures together, we achieved the best drug predictive value with AUC values of 0.918 (docetaxel) (Fig. 7g), 0.949 (paclitaxel) (Supplementary Fig. 16a), 0.906 (vincristine) (Supplementary Fig. 16b), and 0.915 (vinorelbine) (Supplementary Fig. 16c). To further confirm the efficiency of prediction power in clinical practice, we applied datasets of human cancer patients treated with paclitaxel and docetaxel to evaluate the predictive value. Both GSE22513 and GSE6434 human dataset achieved good prediction accuracy for MTi response with AUC values of 0.914 and 0.841 using the 105 combined gene signatures (Fig. 7h,i). Similar result was also demonstrated with patient-derived xenografts (PDXs) (AUC = 0.872) (Supplementary Fig. 16d).

Through our proposed PBPM approach by integrating genomics/transcriptomics and PDO-based drug tests, the multi-dimensionally confirmed results could serve as promising indications to guide individual treatment, even to those subjects

without any identified druggable mutations from genomic analysis. More importantly, guided by the integrated PBPM analysis results, we further demonstrated the histological subtype-related drug responsiveness of MTi on PDOs, which could significantly facilitate precision oncology for NPC to be subtype-guided therapy in the future.

## Discussion

In the present study, we have conducted a comprehensive integrative analysis of genomic and transcriptomic data, and drug sensitivity on PDOs to study the biological features of different subtypes of NPC, their responses to therapeutic drugs, and different combinations of CRT. Our data uncovered several notable findings, including (1) EC, MSEC, SC, and SCC subtypes of NPC have some distinct characteristics in terms of genomics, gene expression, organoid growth pattern, and drug responses, which have not been recognized before; (2) under our optimized culture conditions, PDOs of NPC were established with a high successful rate (93%); (3) anticancer drug screening using PDOs identifies remarkable differential responses of different subtypes to the treatment; and (4) our integrative analysis revealed correlations between subtype-specific molecular characterizations and subtype-specific drug and CRT responses of NPC, which demonstrates a promising pipeline for subtype-guided precision medicine.

Heterogeneity is observed in all cancer types, including NPC, which often leads to treatment failure and tumor recurrence whenever fractions of tumor cells survived from initial therapy[56,57]. Recurrent NPC usually has a poor prognosis with a median OS of ~20 months[58]. Using precision medicine in the initial treatment would significantly improve prognosis and implementing subtype-guided therapeutic regimens is a good attempt. By analyzing genomic mutations and transcriptional expressions of three main subtypes of NPC, including EC, MSEC, and SC, which account for approximately 97% of NPC, we identified aberrant driver pathways of each subtype: EC is driven by microtubule polymerization activations, defective mitotic spindle checkpoint regulation and DNA repair; SC is associated with EMT/invasion promoting signaling; and MSEC is a more heterogeneous subtype with molecular features of both EC and SC. Of note, both EC-type epithelial and SC-type sarcomatoid tumor cells, such as PDO63E and PDO63S in MSEC subtype, although displaying distinct transcriptional expression patterns and diverse responses to certain type of cancer drugs, they share some key common genomic features. This suggests that the heterogeneity of two distinct populations within MSEC might originate from the same cell during tumor evolution.

However, different subtypes of NPC also share some common features, such as defective G1-S checkpoint surveillance, activated NF-κB signaling, aberrant RTKs and chromatin remodeling. Of note, one of the most notable common features is that they all lose *MST1R* at an extreme high frequency (55.6%), indicating that *MST1R* may act as a tumor suppressor in NPC, contrary to the carcinogenic effects of *MST1R* in other types of cancer[40–43]. It has been reported that *MST1R* is an NPC susceptibility gene associated with MST1R/14-3-3 interaction networks in the response to EBV infection, rather than functions as a tyrosine kinase that promotes proliferation and migration[6]. Loss of *MST1R* function may be a cancer susceptible marker for NPC.

Previous studies revealed significant prognostic variation between EC and SC subtypes.[13]. Generally, the SC subtype has an ~20% lower OS rate than the EC subtype, and the OS rate of MSEC is between EC and SC[13]. Other independent studies also demonstrated that NPC with a high proportion of vimentin-positive spindle tumor cells showed poor prognosis, including

survival, clinical stage, lymphatic invasion, and recurrence, rather than nonspindle subtypes[14,59,60]. However, the underlying molecular basis contributing to this subtype difference remains a gap in the field until our present study.

Early-stage NPC is sensitive to RT, and ~90% of patients at stage I benefit from RT. However, 15–58% of NPC patients experience recurrence, and most of them develop metastases and therefore must undergo retreatment[19,61]. The outcome for patients with recurrent or metastatic NPC is very poor, with a median OS of only approximately 20 months[58]. Although CRT and chemotherapy usually provide better treatment outcomes for advanced NPC patients than RT alone[21,22], and have been widely applied as standard care for patients with advanced stage NPC, it remains a huge challenge to improve prognosis by deciding a suitable regimen for the corresponding patient. By classifying NPC into subtypes and identifying effective drugs for each subtype, this strategy offers a good option to facilitate precision medicine. In this study, by utilizing PDOs, we identified several drugs that were efficacious in specific subtypes, including MTi for SC and EGFRi for EC. Among these drugs, docetaxel and paclitaxel are FDA-approved microtubule-targeting drugs for NPC and have been widely applied in clinical practice. EGFR inhibition has also emerged as a new effective strategy against advanced NPC, with options including approved anti-EGFR monoclonal antibodies, cetuximab and nimotuzumab, and chemical EGFR inhibitors, gefitinib and neratinib, in phase III clinical trials for NPC[62–69]. Importantly, our study identified a dozen of CRT combinations with good synergistic effects against NPC PDOs with individual- or subtype-specific response patterns. One of the most effective CRTs was the combination of RT with EGFRi, which exhibited a synergistic and potent killing effect on EC subtype PDOs.

Precision medicine is a promising strategy for treating the right patient with right drugs, which is urgently needed for cancer patients. It was estimated that only a 25% response rate, on the average, is achieved with each round of conventional cancer chemotherapy across cancer types[70], and the combination with druggable mutation detection may significantly increase the therapy response rate to ~50%[71]. Nevertheless, for tumors without known druggable targets or cancer types with less genomic characterization, such as NPC, it remains difficult to pursue a suitable regimen for individuals. Organotypic culture opened a door to fill this need, as this system faithfully recapitulates the original tumor and works as a proxy for testing drugs for individual patients[23–26]. Researchers have demonstrated a series of perfect examples for personalized medicine applying organotypic culture in various cancer types[26,32–34]. We are one of the pioneering groups to establish this platform for treating NPC. Of note, in contrast to the majority of previous reports studying precision medicine approaches either from the genomics/transcriptomics perspective or from the PDX/PDO-based drug screening aspect, we integrated advantages from both sides and proposed the PBPM approach. Genomics/transcriptomics analysis and drug response assay on PDOs mutually validated each other to provide higher confidence in the therapeutic suggestions for physicians and patients than conventional methods used previously. We proposed that the PBPM approach could be a feasible strategy for precision medicine in NPC. Nevertheless, some limitations on the proposed PBPM approach still requires necessary attentions for the field at current stage. First, more cases of clinical validation are essentially needed to evaluate the efficacy and safety of PBPM. Second, there is a potential risk that the heterogenous tumor linages might gradually evolve or lose during a long-term organotypic culture, subsequently affect the accuracy of PBPM. So, to screen for effective drugs using early-passages PDOs is highly recommended. Third, for some tumors of mixed

subtypes, the possibility of selective killing effect of PBPM-identified drugs might occur. Thus, further studies on developing the approach to find drug combinations against various tumor heterogenous clones are necessary.

In summary, we analyzed EC, MSEC, SC, and SCC subtypes and identified subtype-specific driver pathways for NPC. We have established PDOs with high efficiency and uncovered subtype-specific drug and CRT responses by screening a library of anticancer drugs. We elucidated that drug responses are well correlated with corresponding genomic mutations, transcriptional expression and protein changes (Supplementary Fig. 17). Thus, our study provides an example of applying integrative pharmacogenomics to establish a personalized treatment strategy for NPC subtype-guided therapies.

## Methods

**Sample collection**. Fresh tumor biopsies and paired blood samples pathologically diagnosed with NPC were collected from the Kiang Wu Hospital, Macau from August 2016 to December 2019. Formalin fixed, paraffin-embedded (FFPE) NPC tumors and adjacent tissue sections samples were collected at the Affiliated Hospital, Southwest Medical University, Luzhou, Sichuan. Prior patient written consents were obtained from donors with informing the use for genomics sequencing, organoid culture, drug test, publication, and associated scientific studies. All collected tumors were further confirmed, classified, and stage assessed through TNM staging. The protocol of this study was assessed and approved by the ethics committees of University of Macau, Kiang Wu Hospital and the Affiliated Hospital of Southwest Medical University. Detailed clinicopathological characteristics of all NPC patients included in this study are summarized in Supplementary Table 1 and Supplementary Data 1, the prior consents to publish information of gender, age, ethnicity and associated clinical characteristics were obtained from subjects.

**Tissue dissociation**. Upon arrival, tumor tissues were firstly minced into 1–3 mm³ pieces. Two random pieces were picked up for DNA isolation and formalin fixing. The remaining portions were digested with collagenase buffer (Supplementary Data 10) at 37 °C for about 1 h with gentle shaking. The material was further digested with dispase II (5 mg/ml)/deoxyribonuclease (0.1 mg/ml) solution for 5 min at 37 °C and later dissociated with 0.25% trypsin for 1–2 min at 37 °C. After treating with RBC lysis buffer (eBiosciences) for 3 min, the remaining tumor cells were collected for organoid culture and cryopreservation.

**Organoid culture**. Dissociated tumor cells were resuspended in Matrigel solution, then seeded in prewarmed 24-well culture plates at 30 μL per drop. Once cell-Matrigel drops were solidified at 37 °C, 400 μL/well organoid culture medium (Supplementary Data 10) was added in to initiate continuous culture. Medium were refreshed every 2–3 days and passage developed every 5–10 days depending on organoid density and size.

Upon passaging organoids, organoid-Matrigel drops were firstly mechanically disrupted with 0.25% trypsin by pipetting, later were transferred to 37 °C allowing further dissociation to smaller cell aggregates. During continuous culture, tumor organoids originated from different subtypes eventually developed to two distinct phenotypes: EC-type and SC-type. For EC-type organoids, dissociated organoids were directly resuspended in Matrigel solution and seeded for amplifying culture. For SC-type organoid passage, passaging cells were firstly cultured in low-attachment plates to allow aggregation to dense spheres, then formed spheres were setup culture as same as EC-type organoids.

**Histology and immunostaining**. Tumor tissues and organoids were fixed in 10% neutral buffered formalin followed by gradient dehydration, wax immersion, paraffin embedding, and sectioning. For processing organoids, prestaining of eosin was performed during dehydration in order to locate organoids in paraffin blocks and sections. Haematoxylin–eosin (HE) staining and immunostaining was performed using standard protocols on 5 μm sections[72,73]. The antibodies used were listed as following: AE1/3 (1:50, Abcam, ab27988), Vimentin (1:100, CST, 5741 S; 1:100, Santa Cruz, sc-6260), CK5/6 (1:50, Abcam, ab17133), p63 (1:50, Abcam, ab124762), MST1R (1:100, ATLAS, HPA008180), LMP1 (1:100, Abcam, ab78113), Ki67 (1:400, CST, 9449 S), and CD3e (1:100, Dako, A0452).

**Tumor subtyping**. NPC are histologically classified into the following four subtypes based on morphologic characteristics: epithelial carcinoma (EC), sarcomatoid carcinoma (SC), mixed sarcomatoid-epithelial carcinoma (MSEC), and squamous cell carcinoma (SCC)[13]. Most of tumor regions in EC subtype are identified with round epithelial cells and vesicular nuclei phenotype appearance. SC subtype has a large proportion of spindle-shaped and fusiform cells. MSEC subtype encompasses with both round epithelial tumor cells and spindle sarcomatoid tumor cells, with

scattered or nest infiltration of spindle cells in round epithelial cell region. SCC is characterized by keratinizing squamous tumor cells.

**DNA and RNA extraction**. Of 106 samples assigned for WES, DNA of 43 samples were isolated from bulk fresh primary tumors, 4 from bulk FFPE samples, 59 from microdissected FFPE samples (Supplementary Data 1). The tumor purities for bulk samples were assessed by H&E stanning and calculated by Sequenza and Absolute methods (Supplementary Data 1,2)[74,75]. Genomic DNA and paired blood were extracted using DNeasy Blood & Tissue kit (Qiagen) according to the manufacturer's instructions. LASER-capture microdissection of FFPE samples was conducted under Leica LMD 7000 Laser MicroDissection system so that the clusters of tumor cells and adjacent normal tissues could be separately dissected for genomic DNA preparation by QIAamp DNA Micro kit (Qiagen). RNA from organoids was extracted using RNeasy Micro kit (Qiagen).

**Whole exome and transcriptome libraries preparation and sequencing**. DNA library was prepared for Illumina platform using NEBNext Ultra DNA library Prep kit for Illumina (NEB) and subjected to whole-exome enrichment using Nextera Rapid Capture Exome kit (Illumina) following the standard protocol. RNA libraries were prepared using NEBNext Ultra Directional RNA Library Prep kit for Illumina (NEB) following the manufacturer's protocol. Barcoded DNA and RNA libraries were mixed to pool libraries, then run sequencing performed by Novogene Co. Ltd.

**Whole-exome sequencing (WES) data analysis**. Whole-exome sequencing was performed on tumor, paired normal and paired organoid samples with average sequencing coverage ~100× (Supplementary Data 11), including 43 fresh tumor, 63 FFPE tumor and 15 organoid samples. The FASTQ raw files are available at Sequence Read Archive (SRA) database of NCBI under accession number PRJNA716262 (https://www.ncbi.nlm.nih.gov/bioproject/PRJNA716262/) and the National Omics Data Encyclopedia (NODE) database under accession number OEP001733 (https://www.biosino.org/node/project/detail/OEP001733/). The data deposition was complied with the Regulations on Management of Human Genetic Resources in China. The pipeline used for data processing was summarized in Supplementary Fig. 18. In brief, sequence reads were aligned to human genome build 38 (Hg38) using Burrows-Wheeler Alignment tool (BWA-version 0.7.12) (20080505) with maximal exact matches (v0.7.17)[76] followed by reducing duplicates with Sambamba (v0.6.8)[77], realignment of indels and base recalibration with Genome Analysis ToolKit (v4.1.0.0)[78] according to the best practice guidelines.

Somatic SNVs and INDELs were called by three independent programs with default parameters, including MuTect2 (v4.1.0.0), Strelka2 (v2.9.10)[79], and LANCET (v1.1.0)[80] (Supplementary Fig. 2a). Paired blood control or adjacent control samples were applied as the reference to call somatic mutations of each tumor independently. The results of each caller were stored in VCF format and further filtered for PASS variants. The vcf2maf tools (https://github.com/mskcc/vcf2maf) was used to convert VCF into MAF with performing annotation by ENSEMBL Variant Effect Predictor (v100.0)[81] according to reference release 93. Only mutations detected by at least two callers were kept as true positive ones and assigned for further analysis. Somatic SNVs and INDELs were further filtered by ngs-filters (https://github.com/mskcc/ngs-filters) to determine high-confident variants. The following criteria were applied on candidate mutations identification: (i) minor allele frequency in ExAC less than 0.0004. (ii) tumor sample total depth more than 20, and support reads for the reference less than 1 while for alteration more than 3. (iii) variants located in low-mappability regions were filtered. (iv) mutations with allele frequency more than 5% were retained.

SNVs and INDELs called by three tools were merged into a single nonredundant MAF file which contained 2662 somatic mutations including 2148 genes for downstream analysis. Mutation signature analysis was performed by MutationalPatterns (v2.0.0)[82] R package and further compared with COSMIC mutational signatures version 2 using deconstructSigs (v1.8.0)[83] R package applying 'exome2genome' normalized method.

Copy number analysis was performed in both Sequenza (v3.0.0)[75], and CNVkit (v0.9.0)[84] with default parameter. Further chromosome arm-level and gene-level variations were detected by GISTIC 2.0[85]. Consensus CNVs detected by both CNVkit and Sequenza were considered as tune positive ones. Tumor purity was assessed using Sequenza and Absolute method[74,75].

**Transcriptome sequencing and data analysis**. RNA sequencing was performed for 14 PDOs and the quality check was carried out using MultiQC (v1.5)[86]. The FASTQ raw files are available at SRA database of NCBI under accession number PRJNA682500 (https://www.ncbi.nlm.nih.gov/bioproject/PRJNA682500/). The data deposition was complied with the Regulations on Management of Human Genetic Resources in China. RNA reads were aligned to human reference genome GRCh38 using HISAT2 (v2.1.0)[87] followed by counting RNA reads by featureCounts[88], one of programs in Subread package (v1.6.4). Genes with zero read counts for all samples were removed. Then differentially expressed genes between EC- and SC-type PDOs were identified using two tools, DESeq2 (v1.26.0)[89] and limma (v3.42.2)[90], respectively. Significantly differential expressed genes were defined based on overlapped genes between two tools. The cutoff criteria for both tools were the absolute value of $\log_2$ fold change > 1, $p$-value < 0.05,

FDR (padj) value < 0.05. The multiple hypothesis testing was performed using Benjamin–Hochberg correction implemented in the DESeq2 and limma package. DESeq2 found 3853 significantly differential expressed genes while limma detected 2791. There were 2506 common genes between two callers. Pathway enrichment analysis and geneset enrichment analysis were performed using R package clusterProfiler (v3.14.3), results were filtered by absolute NES value > 1, $p$-value < 0.05, and FDR value < 0.05. The multiple hypothesis testing was performed using Benjamin–Hochberg correction implemented in clusterProfiler package.

**Evaluation on drug prediction accuracy with signature geneset.** Public data source of transcriptional expression and drug treatment responses on cancer cell lines, PDXs and cancer patients were download from Genomics of Drug Sensitivity in Cancer (GDSC) database (https://www.cancerrxgene.org/) and Gene Expression Omnibus (GEO) database (https://www.ncbi.nlm.nih.gov/geo/). Cancer cell line data treated with docetaxel (880 lines), paclitaxel (758 lines), vinorelbine (751 lines), and vincristine (727 lines) were download, and samples with IC50 values of bottom 10% and top 10% were defined as drug sensitive and resistant ones. PDX data treated with docetaxel were download from GSE110153 dataset. Data of human breast cancer patients treated with paclitaxel and docetaxel were download from GSE22513 and GSE6434 dataset, respectively. The receiver operating characteristic (ROC) analysis was used for evaluating drug prediction accuracy of in-house signature geneset on downloaded public data. ROC analysis was performed by online tool MetaboAnalyst (v4.0) (https://www.metaboanalyst.ca).

**Validation of gene expression by real-time PCR.** Important differentially expressed genes among NPC subtypes including *BIRC3, AR, FN1*, and *MMP2* were further confirmed using real-time PCR. The details of primers used were listed in Supplementary Table 6.

**Drug screening.** After preliminarily testing 146 chemical drugs on PDOs, we organized a drug library containing 48 therapeutic drugs with good efficacy on NPC for large-scale screening (Supplementary Data 12). The drug library was assembled with 40 cancer-approved drugs in clinical use, four repurposed non-cancer drugs and four nonapproved compounds, but now in phase II/III clinical trials. Compounds were stored in DMSO at the concentration of 10 mM, then diluted in sterile PBS and arrayed in 384-well plates at 6-point serial dilutions (0.82–200 μM, as 10× working dilution). Organoids at early passages were assigned for drug screening. Initially, organoids were enzymatically dissociated into single cells, then diluted with the medium for drug screening (organoid culture medium minus Y-27632, SB202190, and A83-01; supplemented with 2.5% Matrigel). Cells were seeded in type-I collagen gel precoated 384-well plate at a density of about 1200-1500 cells per well. On the other day, drugs were added into each well with 6-point dilutions and cells were incubated with drugs for 4 days. To examine organoid viability, quantifying ATP content in each well as the proxy for metabolically viable cells using the CellTiter-Glo 2.0 assay (Promega) or CellTiter-Lumi Plus assay (Beyotime) was applied. Luminescence readout from drug treated wells were normalized against control wells and expressed as percentage cell viability, IC50 values were calculated using GraphPad Prism software.

**Chemoradiotherapy (CRT) screening and synergistic effect determination.** Organoid viability was firstly examined with ionizing radiation (IR) treatment alone at dose (Gy) of 0, 2, 4, 6, 8, 10 by following same protocol with previous drug screening. For CRT screening, organoids were treated by either chemical drug alone or chemical drug combining with 4 Gy IR, and IR treatment was prior to adding chemical drug in treated wells. When evaluating the combinational effect of each drug with IR, the CRT combination with fold change of IC50 over 3 was recognized as the candidate having good combinational effect. Fold change was calculated by dividing IC50 value of chemical drug treatment alone by IC50 value of combinational treatment. If IC50 value (μM) is >20 or <0.08, the maximum treatment dose (μM) 20 or minimum treatment dose (μM) 0.08 would be served as proxy respectively for calculating fold change. To demonstrate whether CRT combination has synergistic effect or not, we compared three dose response curves, including curves of chemical drug alone treatment, combinational treatment and theoretical additive effect. The predictive additive killing effect is calculated as $E_{total} = E_1 + E_2 - E_1 \times E_2$ (where $E_1$ is inhibitory effect of IR at dose of 4 Gy and $E_2$ is inhibitory effect of drug X at defined dose). This classifies a synergistic CRT combination if dose response curve of combinational treatment is lower than additive effect curve.

**Western blot analysis.** Organoids were passaged and cultured in 6-well plate for one day before adding drugs (5 μM gefitinib or 50 nM docetaxel or vehicle). For gefitinib inhibitor experiment, organoids were collected for total protein extraction at 12 and 24 h after treatment, and for docetaxel, at 24 h and 48 h. The protein bands were visualized by using Immobilon Western Chemiluminescent HRP Substrate (Millipore) under ChemiDoc Touch Imaging System (Bio-Rad). The antibodies used were listed as following: Phospho-AKT (1:1000, CST, 4060 S), AKT (1:1000, CST, 9272 S), Phospho-ERK (1:1000, CST, 4370 S), ERK (1:1000, CST, 4695 S), Phospho-STAT3 (1:1000, CST, 9145 S), STAT3 (1:1000, CST, 9139 S), Cyclin A (1:200, Santa Cruz, sc-751), Cyclin B (1:200, Santa Cruz, sc-752),

Phospho-Bcl-2 (1:1000, CST, 2827 S), Bcl-2 (1:1000, ProteinTech, 12789-1-ap), Bax (1:1000, ProteinTech, 50599-2-lg), Cleaved Caspase-3 (1:1000, CST, 9661 S), MST1R (1:1000, ATLAS, HPA008180), and β-Actin (1:1000, Sigma, A5316).

**Generation of Cas9-sgRNA plasmid and knockout (KO) PDO lines.** Sequences of sgRNA targeting human *TPPP, MAP2*, and *SKP2* were obtained Human CRISPR Knockout Pooled Library (GeCKO v2)[91] (Supplementary Data 13), and were cloned into lentiCRISPR v2 vector (Addgene plasmid # 52961). To generate the KO PDO lines, lentivirus system was employed to transduce organoids. Briefly, lentiviruses were produced by co-transfection of a lentivirus vector plasmid, pCMV delta R8.2, and pCMV-VSV-G into HEK 293 T cells. Fresh culture medium was changed 12 h later and culture supernatant was collected at 72 h post-transfection and concentrated by ultracentrifugation. For organoid transfection, the concentrated lentiviruses were firstly mixed with Matrigel at 1:1 volume ratio, then organoids were embedded into the Matrigel containing lentiviruses and cultured with regular organoid culture medium. To select for mutated cells, the culture medium was supplemented with 2 μg/ml puromycin at 72 h post-transfection and continued for 1 week. The remaining organoids after selection were assigned for growth curve and drug treatment evaluation.

**Statistics and reproductivity.** All representative experiments were performed in triplicates or duplicates independently. The results were reported as the mean ± SEM or mean ± SD. Statistical significance was calculated using the two-sided $t$-test, unless otherwise indicated. The $p$ values were considered statistically significant if the $p$-value ≤ 0.05.

**Reporting summary.** Further information on research design is available in the Nature Research Reporting Summary linked to this article.

## Data availability
The sequencing raw data have been deposited in Sequence Read Archive (SRA) database of the National Center for Biotechnology Information (NCBI) under accession number PRJNA716262 (WES sequencing data) and PRJNA682500 (RNA sequencing data). The WES sequencing data are also available at the National Omics Data Encyclopedia (NODE) database under accession number OEP001733. Public data analyzed in this paper were download from Genomics of Drug Sensitivity in Cancer (GDSC) database (https://www.cancerrxgene.org/) and Gene Expression Omnibus (GEO) database (https://www.ncbi.nlm.nih.gov/geo/). Drug response (docetaxel, paclitaxel, vincristine and vinorelbine) and transcriptional data of cancer cell lines are available from GDSC (https://www.cancerrxgene.org/downloads/bulk_download). Drug sensitivity (docetaxel) and transcriptional data of PDX were download from GEO under accession number GSE110153. Treatment response (paclitaxel, docetaxel) and transcriptional data of cancer patients were download from GEO under accession number GSE22513 and GSE6434. All the other relevant data supporting the key findings of this study can be found within the supplementary files. Source data are provided with this paper.

## Code availability
The code used in the study is available at: https://github.com/xueyinglyu/DengLab-NPC-genomic-analysis.

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

## Acknowledgements

We thank members of the Deng laboratory for helpful advice and discussion, and Information and Communication Technology Office (ICTO) for providing High-Performance Computing Cluster (HPCC) for computational analysis. We also appreciate Dr. Jian-Yong Shao and Dr. Hai-Yun Wang from Sun Yat-sen University Cancer Center for providing valuable suggestions to this study. This work was supported by the Chair Professor Grant (CPG 2015-00016-FHS), Multi-Year Research Grant (MYRG)2016-00139-FHS, MYRG2016-00132-FHS, and MYRG2017-00113-FHS; by The Science and Technology Development Fund, Macau SAR (065/2015/A2, 094/2015/A3, 0048/2019/A1 and 0011/2019/AKP).

## Author contributions

C.X.D. and R.B.D. conceived the study and designed experiments. R.B.D. developed organoid culture protocol, generated and cultured organoids, conducted drug screening, analyzed drug response results, subtype-specific genomic mutations and transcriptional pathways, interpreted gene-drug correlation, performed drug prediction analysis. X.L., B.K.R., and H.W. contributed to data analyses of cancer genome and transcriptome. P.C. performed LASER-capture microdissection on FFPE samples. R.B.D. and J.B. performed H&E staining, immunostaining, qPCR, and Western blot experiment. P.C., W.H., and R.B.D. prepared DNA and RNA libraries for sequencing. H.W., J.Z., and M.V. contributed to bioinformatic analysis protocol development. X.L. performed GSEA analysis on transcriptional sequencing data and correlation analysis among subtype, signature genes and drug response. H.S. and T.K.C. processed tumor dissociation and primary cell isolation. A.H.H.W. coordinated clinical records and specimens. K.I.C. contributed expertise in pathology and tumor subtyping for this study. S.L., L.Y., and J.W. provided patient tumor specimens. E.J.Y. and J.S.S. contributed to drug screening and sensitivity analysis. S.M.S. and X.X. aided on histology experiment. H.S., K.M., Q.C., J.S.S., and X.X. reviewed the manuscript and provided comments. C.X.D., R.B.D., and B.K.R. wrote the manuscript.

## Competing interests

The authors declare no competing interests.
