## [Peer Review File · Nature Communications]

Reviewers' comments:

Reviewer #1 (Remarks to the Author): Expertise in NPC genomics and transcriptomics

In this manuscript, the authors have explored the genomic landscape of 93 NPC samples by WES and then identified the potential somatic changes associated with four newly defined NPC histological subtypes, EC, SC, MESC and SCC. Importantly, they have successfully established a panel of 28 patient derived organoids from the EC, SC and MESC subtypes of NPC. The new valuable resource allows the authors to investigate the signalling pathways and cellular mechanisms contributed to the unique phenotype/features of these NPC subtype by transcriptome sequencing. The authors also demonstrated the usefulness of these PDOs in drug screening and precision oncology. The PDO study is novel for NPC and the establishment of a large panel of PDOs for drug screening is a great achievement in the field.

However, several issues about the study design, hypothesis, methodology, data analysis, accuracy of the results and conclusion are still needed to improve and clarify. Although the PDO establishment and drug screening part are very impressive and convincing, the findings did not sufficient to support their conclusion about the molecular landscape, transcriptome profile and drug response of the NPC subtypes. Furthermore, NPC is an EBV-associated cancer. The status of EBV infection and EBV gene expression in these NPC subtypes have not been examined.

My major concerns are as following:

(1) In addition to the new subtypes, the histological type according to current WHO classification should also be indicated in the Figures and Tables. The current classification should also be shown in the organoids transcriptome and genome assay.

(2) In the introduction section, the reference of "current WHO classification of NPC" should be provided. As I known, the cited "three major histological subtypes: Type-I squamous cell carcinoma; Type-IIA keratinizing undifferentiated carcinoma; and Type-IIB non-keratinizing undifferentiated carcinoma" should not be the current WHO classification! The term "Keratinizing undifferentiated carcinoma" should be wrong!

(3) In Page 3, the statement "for the stage II NPC, especially for patients with metastasis or recurrence, RT alone is less effective" should be revised! Has the subtype of all NPC cases been reviewed by pathologists?

(4) Figure S1A and 1B did not provide evidences for the specificity of markers for EC and SC subtypes. The expression of CK5/6, P63, AE1/3 and Vimentin should be shown in all 4 subtypes. Multiple cases for each subtypes should be shown and a summary table for the expression level and staining pattern of the samples cases should be presented.

(5) For WES, the patient information is incomplete and needed to clarify. In Table S1, a number of cases are with unknown tumor site and tumor stage. The diagnosis of the cases with unknown tumor site is in doubt. The meaning of these catalogues is unclear. In the table, only 14 patients are with stage IV diseases, but 17 patients with metastasis! What is the definition of tumor classification? What is the different between the cases classified as keratinizing (2 cases) and that as SCC (3 cases). Why 9 cases with no

information for the “tumor classification”, but all cases with the information about the new “histological subtype”? Whether all of the cases are diagnosed and reviewed by one or more pathologists ?

(6) In the table, the nature of NPC tumors involved has not been clearly described. Whether all of them from the primary tumors, recurrent tumors, lymph node or distinct metastasis. A comprehensive table including all clinical pathological information and other details (site collected, frozen or FFPE, with and without microdissected?) of each tumor should be included.

(7) A critical point is that the EBV status in all of these tumors have not been mentioned in the articles. Since EBV infection is a critical feature of NPC in endemic regions, the author should confirm the EBV status in the tumors by standard EBER ISH assay.

(8) The major issue of WES study in NPC is the low cellularity in the tumor samples. Since the NPC is heavily infiltrated by lymphocytes and other stromal cells. The tumor content in most NPC specimens is lower than 30%. Biopsies with 0-10% tumor cells were also commonly found. The quality of NPC tumor samples (cellularity and sample source, frozen tissue or microdissected FFPE section?) should be determined and listed in the report. According to Figure S2A, no mutation (or >5?) is noted in 19/93 (20%) of NPC cases (e.g. T5, T6, T19, T24, T29, T32, T33, T39, T40, T41, T50, T64, T68, T70, T71, T72, T73, T79 and T88). It is possibly due to the SNV and indel calling problems in the tumor samples with very low cellularity. Unfortunately, no information about the percentage of tumor cells in each sample were provided. The mutant allele frequency and case number of the tumors were not shown in Table S2a. The informatics analysis seems to have problems in a number of the samples. In Table S2a, most than 40 cases are with indels only while no SNV is detected (e.g. S6879, S8197, S9864, S6743, S27208, S26982, S2624, S2616, S24472, s23662, S21300, S15536, s14565, s9881, s9598, s2942, s24301, s228, s20573, s20471, s18133, s17888, s140, s13088, s11026, s25239, s19960, s21125, s17568, s10760, s8292, s6630, s3935, s22498, s20721, s13799, s11078, km1856, KM1618, ...). The quality of informatics analysis is questionable! The authors should further review the quality of their raw data, informatics pipelines and comprehensively validate the mutations in multiple cases (such that listed above). The low frequency of TP53 mutations identified also evidences the problem in SNV and indel calling in current study.

(9) The C>A base situation has not been shown as predominant signatures in previous WGS studies, the authors are needed to confirm whether it is due to the low cellularity of the tumors and poor SNV calling in these samples. The reported mutational signatures as well as subtype-specific mutational frequency may also be affected by these technical problems.

(10) For the CNV analysis, it is interesting that the MST1R homozygous deletion was found in 48.4% of tumors. Nevertheless, this important finding should be carefully validated. The authors should perform FISH analysis to confirm the MST1R homozygous deletion in all these tumors. It is also noted that the BAP1 gene on the same chromosomal region also shows high frequency of homozygous deletion. It is possible that MST1R is not the driver cancer genes, only a bystander in this region. The BAP1 tumor suppressor function has been demonstrated in previous NPC WES study.

(11) Furthermore, Supplementary Figure 3A, C and D indicates a high background noise signals. It is also noted that the frequencies of amplification/homozygous deletions identified in this study is much higher that reported previously. The CNV calling should be carefully reviewed and selected amplification and deletion regions in the tumors are needed to be validated by FISH analysis. The potential subtype-associated cancer driver pathways and networks should be reanalyzed while the SNV/indel/CNV data is confirmed.

(12) The establishment of multiple patient-derived organoids from NPC tumor is a breakthrough in the field. Importantly, the EBV latency in these PDOs should be determined. Furthermore, STR profiling should be performed in the established PDOs and corresponding tumor samples for authentication. The authors should be clearly mentioned the PDOs are derived from the tumors involved in the WES study. WES should also be performed in these PDOs. The somatic alterations identified in PDOs and corresponding tumor samples should be compared. A number of SNPs/Indels/CNV missing in WES study may be identified from these homogeneous tumor samples. It may allow us to identify the potential link between somatic changes and this EC and SC subtypes.

(13) It is possible that the EC and SC features may be influenced by the EBV latent gene expression, e.g. the heterogeneous LMP2A or LMP1 expression. The authors should examine the EBV transcriptome from the RNA-seq data of these two types of PDO, as well as T55E and T55S. It would be important to check whether there are different acquired mutations in these two PDO sublines. The correlation of somatic genetic alterations and the transcriptome profiles of the two/three subtypes PDOS should be investigated to check whether the transcriptome and phenotypes of these PDOs are driven by somatic alterations or EBV latent gene expression. In addition to the signaling pathways and cellular mechanisms described in Figure 5, the stem cell-related expression profile in these organoids should be explored. The unique transcriptome profile and phenotype of SC or MSEC PDOs may be due to the different passage method by which the stem-cell population was enriched.

(14) As the MSEC type organoids showed morphology changes to SC type organoids, it suggests that the EC type NPC cells were loss or undergone ENT transition to SC subtype during continuous passage. Thus, it is reasonable the transcriptome profile of MSEC type is similar to that of SC type. The author should discuss this point in their manuscript. It is likely that the epithelial- and spindle-types of NPC cells occur in all EC, MSEC and SC cells, but with different proportions. For the MSEC type NPC , the treatment of MTi or EGFRi may only result in the selection/enrichment of either EC or SC subclones of the tumors in the patients. It may also be happened in the EC and SC subtype of NPC while the tumors contain a small subpopulation of EC and SC cells. A poor response of the NPC patients was reported in a previous clinical trial of EGFRi . The PDO drug screening provides a novel approach for NPC precision treatment. However, it is important to identify a drug that is able to efficiently inhibit the growth of both epithelial- and spindle-type tumor cells. It is noted that the antitumor effect of Cisplatin, a common first line chemo-agent for NPC treatment has not been included in the drug screening panel. It is important to compare the new potential drugs to the

standard chemotherapy for NPC. In Figure 6, the treatment of RT+ Cisplatin should be conducted on the PDOs for comparison.

(15) In Figure 7, the somatic changes in PDOs should be further validated. Since there is no hotspot mutations or functional activations of TRKs and other pathways by the CNVs in Figure 7A and 7B, there are no sufficient evidences to support authors' conclusion in the last section. In supplementary Figure 11, the data shows that the copy ratio of MET and KRAS is less than 1.5 and 1. It means that the two genes are with copy number gains, but not amplification. The low level copy number gain may not be able to activate these oncogenes.

Reviewer #2 (Remarks to the Author): Expertise in organoids

NPC is a leading cause of cancer-related deaths in Southeast Asia and North Africa. NPC is currently classified into 4 morphological subtypes that are linked to prognosis. However, the major driving events (genomic and transcriptomic features) for the 4 histological subtypes and the druggable targets among the 4 subtypes are currently unknown. Only few drugs are approved for NPC and many NPCs of the SC type show relapse to current therapies. Better understanding of the driving events of NPC subtypes is needed that can be used to find new subtype-related strategies.

Ding et al. use transcriptomic and genomics on a cohort of 93 nasopharyngeal cancer (NPC) samples to explore subtype-specific molecular and genomic features.

With respect to point mutations, authors classify significantly mutated genes in 9 signaling pathways. Predominant mutational signature across all NPCs was Signature4, a result of smoking.

Next, copy-number alteration analysis revealed that specific genomic deletions and amplifications are consistent features for all NPCs.

Furthermore, they made a patient-derived organoid (PDOs) biobank of 28 NPC samples containing 3 different subtypes (out of 4). PDOs were used to integrate transcriptomics/genomics information underlying subtype classification with functional assays like drug response. Interestingly, NPC PDOs from EC subtype showed sensitivity towards treatment with EGFRi and resistance to treatment with microtubule targeting agents. Opposing sensitivities were found for SC and MSECs.

Overall, the strength of the manuscript lies in its extensive transcriptomic and genomic analyses of NPCs. In addition, new NPC PDOs methods were developed, including the establishment of a PDO biobank. Interestingly, drug screens on the PDOs revealed differentiating drug responses that correlate with morphology-based subtype classification. Despite the impressive amount of resource-like data, the manuscript remains rather descriptive concerning the drug phenotypes and might benefit from supporting underlying mechanistic insights.

Comments:

1. The results regarding the mutational landscape of NPCs is largely descriptive and overinterpreted by clustering mutations in 9 signaling pathways. What I found striking when studying the data is that not a single driver gene seems to be mutated in more than few cases. Even if you subclassify per tumor type, few mutations make it barely over 10%. Is it possible that no clear-cut driver mutations underly NPC development?

In contrast, based on CNA profiles, few genes were consistently deleted (up to 50% of the cases) in NPCs across subtypes. Most of them seem to co-occur in same patients. Are these genes positioned at similar chromosome locations (MST1R, BAP1, TRAF3, NFKB1A etc)? Are these tumor suppressor genes? Rescue experiments by ectopic expression of some of these genes might provide interesting answers.

In light of the overinterpretation of the genomic data, I don't think that the systems biology interpretation of the genomic data (Fig. 2D and E) adds much value to the manuscript.

2. The authors demonstrate convincing differences in tumor phenotypes that are recapitulated during organoid growths. In fig 5 gene expression is shown that correlates to MT inhibitor response. Transcriptional profiling of tumors is of interest, but this data should be presented from an unbiased angle.

3. Using all the genomic/ transcriptomic information, can the authors demonstrate or speculate about the possibility that the tumor subtypes originate from different cell types as origin, rather than accumulation of different driver mutations/ CNA?

4. The authors show that ECs show higher mutational frequencies in pathways affecting MT polymerization, mitotic spindle checkpoint, DNA repair and ABC transporters, whereas RTK genes show more CNV amplifications in SCs. However, from figure 2C it seems that RTK signaling is deregulated in both subtypes.

5. The authors show that MSECs contain organoids with EC phenotypes as well as organoids with SC phenotypes that show distinct expression of markers for EMT and epithelium.

First, are these MSECs a mixed population of subtypes that originate from the same original tumor cell or are MSECs a polyclonal tumor? Overlap in mutational landscapes between EC and SC organoids from 'same' MSEC tumor might be informative.

6. Drug screening in organoids reveals that ECs are sensitive to EGFRi treatment, whereas SCs and MSECs are sensitive to MTi treatment. The differences in drug responses per subtype is very striking. However, not only MTi show differential effect, same holds true for the topoisomerase inhibitors.

a) Can it be that SCs and MSECs are sensitive to all classic chemotherapies? The authors should expand the list with chemo's with different mechanism of action (cisplatin/ carboplatin/ 5FU) for a number of organoids.

b) Drug response was tested with ATP assay, which cannot make the distinction

between cell death or cell cycle arrest. Can the authors confirm by microscopy or FACS that organoids are dying or arrested in cell cycle?

c) Drug screening and western blotting was done on 1-day old organoids, which could affect therapy outcomes due to stress. Do authors observe same results on 5-day old organoids?

d) The relative IC₅₀ concentrations of EGFRi are very high and are likely not be reached in patients. Can similar results (pERK inhibition) be obtained with concentration ~0,5μM?

e) EGFRi-resistant SCs show very low ERK activity during unperturbed conditions compared to ECs, suggesting that this pathway is not essential (hence the resistance upon inhibiting something that is not active). Can the authors confirm that limited baseline ERK levels correlate with no response to EGFRi in SC PDOs?

f) In relation to point 4: low basal pERK activity as detected in SC PDOs (Fig 4F) is at odds with the statement of deregulated RTK signaling in this subtype.

7. The authors attribute the MTi resistant phenotype of ECs to CNVs in various signaling pathways. However, most significantly different CNVs between ECs and SCs affect ABC transporters. Are ABC transporters involved in drug resistance? Can authors block ABC transporters with inhibitors to see if enhanced sensitivity to MTi are established in ECs? Furthermore, in relation to point 6a, do ECs show a general resistant phenotype to most classical chemo's or only MTi? Perhaps ABC transporters underly most drug phenotypes?

8. The authors showed that ECs were more sensitive to irradiation than SC and MSECs. In addition, they show that EGFRi + irradiation show synergistic effects in ECs. Is MTi + irradiation relevant in SCs?

9. The authors show that treatment of SCs with EGFRi, HDACi and METi induces a synergistic effect. However, effect of individual drug responses should be shown, as well dual combinations in order to evaluate synergism of three combined.

Reviewer #3 (Remarks to the Author): Expertise in organoids drug screens

In this manuscript, Ding Reng-bo et al acquired tumors from 93 NSC patients and classified them into four different subtypes EC, SC, MSEC and SCC based on histology and molecular markers. In addition, they used genomic analysis to begin to investigate the molecular basis for the different subtypes.

The authors also report generation of a library of ~28 organoids and identify feature that relate them to some of the NPC subtypes. They used the organoids test for sensitivities to drugs and found subtype differences in drug response, where EC-subtype PDOs are sensitive to EGFRi, while SC-subtype are sensitive towards MTs. Interestingly the EC subtype organoids were more sensitive when treated with EGFRi

in combination with radiotherapy (RT). Overall this is an interesting study that requires further development before it can be considered for publication in Nature Communication.

Major concerns:

1) This study expands previous genomic analysis of NPC. Although the authors attempted to understand subtype specific differences in genetic alternations, the analysis did not provide an compelling argument for a relationship between genomic signatures and histopathological subtype classification perhaps due to low sample number representing some of the subtypes and hence statistical rigor. Most of the analysis presented in Figures 1 and 2 are weak and may need to be revisited and reduced in scope.

2) The development and use organoid models is interesting and an important new contribution to the field. However, it is not clear how well the expression profile performed in the 14 organoid models relate to the gene expression profile of matched primary patient tumors. Furthermore, it would be important for the authors to investigate how the drug response observed in organoid models respond the treatment received by the patient to understand how well the organoid models replicate the patient's clinical response.

3) The comparison between the PDO sensitivity and GDSC database is of unknown significance because there are many confounding factors that differentiate the two sets of observations, which include culture conditions, tumor type, cell lines used etc.,

4) The relationship between EC subtype and sensitivity to CRT is interesting and can be significant clinical benefit. However, a pharmacogenomic analysis to tailor such a treatment may not be needed as a combination of histopathological and IHC based identification of EC subtype may be sufficient. It would be informative and important for field, if the authors can demonstrate a correlation between EC subtype and clinical response to CRT in a retrospective analysis using a classification system that uses IHC and histopathology.

5) A mechanistic insight into how do MT and EGFR inhibitors affect organoids growth will be necessary. Perhaps this relates to inhibition of one or more pathways identified by genomic analysis.

6) The manuscript needs significant editing for English usage as there are major grammatical errors which makes it hard for the reader to comprehend and follow.

Minor

1. Fig1A, B. Images of better quality and resolution should be provided. The

Histological features claimed by the authors are not clear.

2. Fig 3 – how were the 28 organoids chosen for PDO analysis? Provide better phase images for Fig 3A.
3. Fig 3D – Provide images showing that the spindle type organoids generated using the aggregate method and stained for pan-epithelial markers.
4. Do the tumor cells restart growth after drug withdrawal?
5. Can the authors provide insight into the genomics of cells remaining in PDO cultures that resist treatments?

Responses to Reviewers' Comments

Reviewers' comments:

Reviewer #1 (Remarks to the Author): Expertise in NPC genomics and transcriptomics

In this manuscript, the authors have explored the genomic landscape of 93 NPC samples by WES and then identified the potential somatic changes associated with four newly defined NPC histological subtypes, EC, SC, MESC and SCC. Importantly, they have successfully established a panel of 28 patient derived organoids from the EC, SC and MESC subtypes of NPC. The new valuable resource allows the authors to investigate the signalling pathways and cellular mechanisms contributed to the unique phenotype/features of these NPC subtype by transcriptome sequencing. The authors also demonstrated the usefulness of these PDOs in drug screening and precision oncology. The PDO study is novel for NPC and the establishment of a large panel of PDOs for drug screening is a great achievement in the field.

However, several issues about the study design, hypothesis, methodology, data analysis, accuracy of the results and conclusion are still needed to improve and clarify. Although the PDO establishment and drug screening part are very impressive and convincing, the findings did not sufficient to support their conclusion about the molecular landscape, transcriptome profile and drug response of the NPC subtypes. Furthermore, NPC is an EBV-associated cancer. The status of EBV infection and EBV gene expression in these NPC subtypes have not been examined.

My major concerns are as following:

(1) In addition to the new subtypes, the histological type according to current WHO classification should also be indicated in the Figures and Tables. The current classification should also be shown in the organoids transcriptome and genome assay.

Thank you very much for the suggestions. We have annotated NPC tumors with WHO classification in Supplementary Data 1 (clinicopathological characteristics), Fig. 1D (SNV oncoplot), and Fig. 2C (CNV oncoplot). Because all samples used for transcriptome analysis are derived from non-keratinizing undifferentiated carcinoma (NKUC) subtype tumors, we didn't indicate the WHO subtype, but annotated samples with EC, MSEC, SC subtypes in Fig. 6.

(2) In the introduction section, the reference of "current WHO classification of NPC" should be provided. As I known, the cited "three major histological subtypes: Type-I squamous cell carcinoma; Type-IIA keratinizing undifferentiated carcinoma; and Type-IIB non-keratinizing undifferentiated carcinoma" should not be the current WHO classification! The term "Keratinizing undifferentiated carcinoma" should be wrong!

Thank you very much for the suggestions. We have revised the description on WHO classification of NPC according to latest edition (2017) as following:

"According to the current World Health Organization (WHO) classification system,

NPC is classified into three major histological subtypes: keratinizing squamous cell carcinoma (KSCC), non-keratinizing carcinoma, and basaloid squamous cell carcinoma. Non-keratinizing tumors are further sub-categorized as non-keratinizing undifferentiated carcinoma (NKUC) or non-keratinizing differentiated carcinoma (NKDC) (Stelow and Wenig, 2017).” (Page 3 in the manuscript)

Our previous description was from a document jointly issued by WHO and Union for International Cancer Control (UICC), (at page 234, in the book of “The Selection and Use of Essential Medicines: Report of 20th WHO Expert Committee”, regarding NPC section can be download at WHO website from the following link: https://www.who.int/selection_medicines/committees/expert/20/applications/NasopharyngealCarcinoma.pdf?ua=1). We referred their description “NPC has historically been classified into different histological subtypes: Type 1 (I) squamous cell carcinoma; Type 2a (II) keratinizing undifferentiated carcinoma; and Type 2b (III) non-keratinizing undifferentiated carcinoma.” We now used the latest edition (v. 2017).

(3) In Page 3, the statement “for the stage II NPC, especially for patients with metastasis or recurrence, RT alone is less effective” should be revised! Has the subtype of all NPC cases been reviewed by pathologists?

Thank you very much for the suggestions. We have revised the description by giving more details (Page 4), as following:

“RT is established as the definitive treatment for non-metastatic NPC at an early stage, which leads to favorable clinical and survival outcomes with a 5-year OS rate of 87.3% to 93% for stage I NPC patients (Chen et al., 2014; Chua et al., 2003; Lee et al., 2005; Yi et al., 2006). However, due to the intrinsic invasiveness and asymptomatic nature of the disease, the majority of NPC patients (60–70%) are diagnosed at an advanced stage, with local spread or regional lymph node metastasis (Li et al., 2017a). For NPC patients with recurrent/metastatic tumor, the outcome is very poor, with a median OS of approximately 20 months (Perri et al., 2019), and chemoradiotherapy (CRT) is the standard-of-care treatment at this stage as recommended by the National Comprehensive Cancer Network guidelines (v. 2.2018). If the cancer cells have spread to distant organs, chemotherapy is the only option (Ma et al., 2018; Perri et al., 2019).” The subtype of all NPC cases has been reviewed by pathologists from the collaborating hospitals.

(4) Figure S1A and 1B did not provide evidences for the specificity of markers for EC and SC subtypes. The expression of CK5/6, P63, AE1/3 and Vimentin should be shown in all 4 subtypes. Multiple cases for each subtype should be shown and a summary table for the expression level and staining pattern of the samples cases should be presented.

Thank you very much for the comments. We proposed AE1/3 and Vimentin as the molecular markers to distinguish NPC subtypes in our manuscript. The immunofluorescence staining of AE1/3 and Vimentin were shown in 12 EC-subtype

tumors, 9 MSEC-subtype tumors, 7 SC-subtypes tumors and 3 SCC-subtype tumors (Fig. 1B; Supplementary Fig. 1A). We summarized their expression levels among subtypes in Fig. 1C. Among 31 stained cases, 9/12 EC-subtype tumors are strongly positive for AE1/3 expression, whereas 6/7 SC-subtype tumors extensively express Vimentin (Fig. 1C). Our data indicated AE1/3 and Vimentin as molecular marker candidates to distinguish NPC subtypes. Because CK5/6 and P63 showed very similar expression pattern with AE1/3, so we just demonstrated their representative images in Supplementary Fig. 1B.

(5) For WES, the patient information is incomplete and needed to clarify. In Table S1, a number of cases are with unknown tumor site and tumor stage. The diagnosis of the cases with unknown tumor site is in doubt. The meaning of these catalogues is unclear. In the table, only 14 patients are with stage IV diseases, but 17 patients with metastasis! What is the definition of tumor classification? What is the different between the cases classified as keratinizing (2 cases) and that as SCC (3 cases). Why 9 cases with no information for the “tumor classification”, but all cases with the information about the new “histological subtype”? Whether all of the cases are diagnosed and reviewed by one or more pathologists?

Thank you very much for the comments. We carefully reviewed the clinicopathological records of all 106 NPC patients (93 previous cases and 13 new cases) together with pathologists of collaborating hospitals again and list the detailed clinicopathological annotations for individual patients in Supplementary Data 1. All the 106 samples were from the tumor site of nasopharynx and they all have information for the “tumor classification”. All the clinicopathological information were diagnosed and reviewed by multiple pathologists from the collaborating hospitals.

To be noted, the clinical status of regional lymph node metastasis was documented in Macau sample cohort but wasn't provided in Luzhou cohort (because the Hospital does not have this information). In the previous description, we classified samples with regional lymph node metastasis as metastasized cases, which accounts for the reason why some lower grade tumors were listed as metastasis tumors. In present version, we defined metastasis case as patient who was recorded with distant metastasis. According to the revised statistics on all enrolled NPC patients (Supplementary Table 1), 18 patients were diagnosed with distant metastasis; 15 patients were diagnosed with stage IV tumor, 43 were stage III, 25 were stage II, 13 were stage I; all 106 patients' samples were assigned for subtype classification, including 78 NKUC, 25 NKDC and 3 KSCC under WHO classification system, and 57 EC, 26 MSEC, 20 SC and 3 SCC under new histological classification system.

(6) In the table, the nature of NPC tumors involved has not been clearly described. Whether all of them from the primary tumors, recurrent tumors, lymph node or distinct metastasis. A comprehensive table including all clinical pathological information and other details (site collected, frozen or FFPE, with and without microdissected?) of each

tumor should be included.

Thank you very much for the suggestions. We have provided the detailed clinicopathological annotations for individual patients in Supplementary Data 1. All 106 samples assigned for WES and organoid culture all were from the primary site of nasopharynx. Among samples assigned for WES, 43 were isolated DNA from bulk fresh tumor samples, 4 from bulk FFPE samples, 59 from microdissected FFPE samples. We have indicated this information in Supplementary Data 1.

(7) A critical point is that the EBV status in all of these tumors have not been mentioned in the articles. Since EBV infection is a critical feature of NPC in endemic regions, the author should confirm the EBV status in the tumors by standard EBER ISH assay.

We have examined the EBV status by PCR assay. Briefly, we applied PCR to amplify region of Epstein–Barr nuclear antigen 1 (EBNA1), which is specific to EBV genome. This PCR method for detecting EBV infection status is well established for decades (Lo et al., 1999). Tumor DNA samples from 40 NPC patients were assigned for this examination. The results showed that 36 samples were strongly positive, and 4 samples were low positive (Supplementary Data 1). Our results was close to the previous report that 96.5% NPC patients were positive for plasma EBV DNA detection (Lo et al., 1999).

(8) The major issue of WES study in NPC is the low cellularity in the tumor samples. Since the NPC is heavy infiltrated by lymphocytes and other stromal cells. The tumor content in most NPC specimens is lower than 30%. Biopsies with 0-10% tumor cells were also commonly found. The quality of NPC tumor samples (cellularity and sample source, frozen tissue or microdissected FFPE section?) should be determined and listed in the report. According to Figure S2A, no mutation (or >5?) is noted in 19/93 (20%) of NPC cases (e.g. T5, T6, T19, T24, T29, T32, T33, T39, T40, T41, T50, T64, T68, T70, T71, T72, T73, T79 and T88). It is possibly due to the SNV and indel calling problems in the tumor samples with very low cellularity. Unfortunately, no information about the percentage of tumor cells in each sample were provided. The mutant allele frequency and case number of the tumors were not shown in Table S2a. The informatics analysis seems to have problems in a number of the samples. In Table S2a, most than 40 cases are with indels only while no SNV is detected (e.g. S6879, S8197, S9864, S6743, S27208, S26982, S2624, S2616, S24472, s23662, S21300, S15536, s14565, s9881, s9598, s2942, s24301, s228, s20573, s20471, s18133, s17888, s140, s13088, s11026, s25239, s19960, s21125, s17568, s10760, s8292, s6630, s3935, s22498, s20721, s13799, s11078, km1856, KM1618,). The quality of informatics analysis is questionable! The authors should further review the quality of their raw data, informatics pipelines and comprehensive validate the mutations in multiple cases (such that listed above). The low frequency of TP53 mutations identified also evidences the problem in SNV and indel calling in current study.

Thank you very much for the comments and suggestions, which we completely agree.

We have improved our bioinformatic analysis pipelines by using multiple callers for mutations identification. Briefly, to identify somatic SNVs, instead of using only MuTect2 (v4.1.0.0), we now also use Strelka2 (v2.9.10) (Kim et al., 2018), and LANCET (v1.1.0) (Narzisi et al., 2018). Mutations detected by at least 2 callers were kept as true positive ones and were assigned for further analysis (Supplementary Fig. 2A).

Supplementary Figure 2A. Mutation amounts called by MuTect2, Strelka2 and Strelka2 with two-two overlap.

Somatic SNVs were further filtered by ngs-filters (<https://github.com/mskcc/ngs-filters>) to determine high-confident variants. The detailed mutation identification protocol and filtering criteria could be found in the materials and methods section (Page 23). A total of 2721 somatic mutations including 2358 missense mutations, 192 nonsense mutations, 83 deletions, 31 insertions and 57 other types of mutations were detected (Fig. 1D) (Supplementary Data 2).

To identify CNVs, both Sequenza (v3.0.0) (Favero et al., 2015) and CNVkit (v0.9.0) (Talevich et al., 2016) were applied with default parameters. Further chromosome arm-level and gene-level variations were detected by GISTIC 2.0 (Mermel et al., 2011). Consensus CNVs detected by both CNVkit and sequenza were considered as true positive ones.

After comparing our results with previous studies, we believe the current results in the revised manuscript were of good reliability. For example, among somatic SNVs, frequencies of TP53 were 9.0% (Li et al., 2017b), 10.6% (Lin et al., 2014), 7.3% (Zheng et al., 2016) and 5.9% (Zhang et al., 2017) in previous studies, which is consistent to our result (7.4%). More comparisons on NPC driver SNVs and CNVs frequencies between ours with several previous studies are summarized in Supplementary Table 2, 3, 4, which indicated our current results were of good confidence.

Supplementary Table 2. Comparison of driver SNV frequencies in NPC.

Gene (SNV)	Our study (n=95)	(Li et al., 2017a) (n=111)	(Lin et al., 2014) (n=113)	(Zheng et al., 2016) (n=124)	(Zhang et al., 2017) (n=51)
TP53	7.4%	9.0%	10.6%	7.3%	5.9%
CYLD	9.5%	9.9%	-	3.2%	5.9%
KMT2C	5.3%	5.4%	-	3.2%	2.0%
FBXW7	4.2%	3.6%	-	-	3.9%
NOTCH2	6.3%	2.7%	0.9%	-	-
NFKBIA	4.2%	6.3%	-	4.8%	-
TRAF3	3.2%	8.1%	-	-	5.9%
PTEN	2.1%	3.6%	-	2.4%	-
MTOR	2.1%	1.8%	1.8%	-	-
PIK3C2B	2.1%	-	0.9%	-	-
PIK3CG	2.1%	-	1.8%	-	-
KMT2B	3.2%	1.8%	-	2.4%	-
ARID1A	2.1%	-	4.4%	1.6%	-
NF1	4.2%	1.8%	-	-	-
ATM	3.2%	-	-	1.6%	-
BAP1	2.1%	-	3.5%	3.2%	-
SYNE1	2.1%	-	8.9%	-	2.0%
FAT2	3.2%	-	2.7%	-	-

“-” refers to unmentioned.

We believe the relatively low mutational rate mainly comes from the intrinsic nature of NPC. As demonstrated by multiple previous studies, the number of somatic SNVs per NPC sample was ranging from 22 to 32 in these cohorts (Chow et al., 2017; Lin et al., 2014; Zhang et al., 2017; Zheng et al., 2016), which is very consistent to our results (29 SNVs per sample).

It is true that low tumor purity could be another cause contributing to low mutational rate. But in our samples, we used microdissection to isolate cancer cells and the tumor purity is relatively high. As shown in Supplementary Data 1, only 7 samples were with tumor purity lower than 20%, which indicated that tumor purity issue was not the essential cause for the low mutational rate of NPC in our cohort. In addition, because we sequenced our samples at averagely 100X coverage (Supplementary Data 10) to avoid the potential effect of low purity and consequently our data indicated that, despite a few tumors showed low SNVs, they were not related to the cancer purity.

SNVs of unpaired samples could not be reliably called and very few mutations (0-6 mutations per sample) were detected if applying strategy of filtering by known NPC mutations. In new version, we excluded 11 unpaired samples from SNV analysis, and

also add 13 new paired samples. We totally included 95 paired tumors in current SNV analysis. This also improve the reliability of mutation calling of current version.

In sum, our old analysis is indeed problematic. We should not include 11 unpaired samples in SNV analysis and also more mutation calling programs should be used. Thank you very much for pointing out this, which helps us to improve this study.

(9) The C>A base situation has not been shown as predominant signatures in previous WGS studies, the authors are needed to confirm whether it is due to the low cellularity of the tumors and poor SNV calling in these samples. The reported mutational signatures as well as subtype-specific mutational frequency may also be affected by these technical problems.

As mentioned earlier, we have carefully re-analyzed all our samples, and the new data should be reliable. Our mutational signature analysis revealed that C>T base substitution was the predominant signature in NPC (Fig. 1F), and this part is consistent with previous studies. The second frequent signature in our NPC cohort was the C>A transition, and consistent conclusion was also demonstrated by previous report (Chung et al., 2019). We speculated that increased C>A transition frequency in our cohort was associated with smoking exposure (Supplementary Fig. 2B). It was well established that smokers had more C>A base substitution than non-smokers in pan-cancer levels (Alexandrov et al., 2016), which primarily caused by COSMIC signature 4 and 29 (https://cancer.sanger.ac.uk/cosmic/signatures_v2.tt). In our NPC cohort, significant higher frequency of C>A transition was observed in smoker than in non-smoking patients (Supplementary Fig. 2D).

(10) For the CNV analysis, it is interesting that the *MST1R* homozygous deletion was found in 48.4% of tumors. Nevertheless, this important finding should be carefully validated. The authors should perform FISH analysis to confirm the *MST1R* homozygous deletion in all these tumors. It is also noted that the *BAP1* gene on the same chromosomal region also shows high frequency of homozygous deletion. It is possible that *MST1R* is not the driver cancer genes, only a bystander in this region. The *BAP1* tumor suppressor function has been demonstrated in previous NPC WES study.

Thank you very much for the suggestions. In order to confirm the prevalence of *MST1R* loss in NPC. We performed a serious of investigations listed as following.

1) To make sure the mutation calling provided true positive results, we applied two independent callers, Sequenza and CNVkit, and only considered overlapped ones as real mutations (Fig. 2C; Supplementary Fig. 4A, B). Thus, the frequency of double-confirmed *MST1R* loss was 55.7%, we didn't differ homozygous and heterozygous loss in this time. High frequency of *MST1R* loss in NPC had been also reported by a previous study published in PNAS (Dai et al., 2016b). In their results, the frequency of *MST1R* loss was 65%, which is very close to ours.

2) *MST1R* located in 3p21.31 and *BAP1* located in 3p21.1, the two genes were neighboring genes. Chromosomal 3p is the most frequently deleted region in NPC (Supplementary Table 3). It is possible that the deletion of both genes is largely because of their close distance. Since *BAP1* is a well-known tumor suppressor, it is possible that deletion of *MST1R* is not the driver cancer event, and only as a bystander in this region.

3). Of note, *MST1R* was reported as a C-Met-related tyrosine kinase, and normally harbored activation/gain mutations and/or overexpression in other cancer types and could play an oncogenic role as a tyrosine kinase to enhance the activation of Ras/MAPK and other signaling cascades (Cheng et al., 2005; Maggiora et al., 2003; Wang et al., 2006; Yang et al., 2017). Study also showed that *MST1R* played a vital function in host defense against viral infection, including Epstein–Barr virus (EBV) and human immunodeficiency virus (HIV) (Cary et al., 2013; Dai et al., 2016b; Feng et al., 2016). Thus, loss of *MST1R* might increase oncogenic susceptibility associated to EBV infection in NPC patients. To explore if *MST1R* could affect cell growth, we applied CRISPR-Cas9 to knockout *MST1R* in wildtype PDOs, our results showed that loss of *MST1R* didn't increase the growth of PDOs (Supplementary Fig. 5F), indicating the loss of *MST1R* alone could not affect the growth of organoid.

4) To confirm the deletion of *MST1R*, we first conducted FISH analysis, however, due to a technical problem, we failed. We then asked a commercial company to conduct the FISH for us, and the Company tried two times but still did not get it work. To bypass this difficulty, we used IHC assay instead, and detected strong expression of *MST1R* in wildtype samples but only weak or no signals in *MST1R* deleted samples (Supplementary Fig. 5A, B)

(11) Furthermore, Supplementary Figure 3A, C and D indicates a high background noise signals. It is also noted that the frequencies of amplification/homozygous deletions identified in this study is much higher than reported previously. The CNV calling should be carefully reviewed and selected amplification and deletion regions in the tumors are needed to be validated by FISH analysis. The potential subtype-associated cancer driver pathways and networks should be reanalyzed while the SNV/indel/CNV data is confirmed.

Thank you very much for the suggestions. After improving our bioinformatics, background noise signals were no longer observed (Supplementary Fig. 3A).

As mentioned in answer to question 8, we compared our results with several previous NPC studies (Supplementary Table 3, 4). Our alteration frequencies of key driver genes were consistent to theirs. In addition, our results were overlapped from two independent callers, they should be of good reliability to be true mutations.

Furthermore, we applied IHC assay to validate a key gene alteration, *MST1R* loss,

mentioned in answer to question 10.

(12) The establishment of multiple patient-derived organoids from NPC tumor is a breakthrough in the field. Importantly, the EBV latency in these PDOs should be determined.

Furthermore, STR profiling should be performed in the established PDOs and corresponding tumor samples for authentication. The authors should be clearly mentioned the PDOs are derived from the tumors involved in the WES study. WES should also be performed in these PDOs. The somatic alterations identified in PDOs and corresponding tumor samples should be compared. A number of SNPs/Indels/CNV missing in WES study may be identified from these homogeneous tumor samples. It may allow us to identify the potential link between somatic changes and this EC and SC subtypes.

Thank you very much for the suggestions. We have performed IHC staining for Epstein–Barr virus latent membrane protein 1 (LMP1) on paired tumors and PDOs. As demonstrated in Fig. 3F, PDOs retained EBV latency consistently.

We agree with you that to evaluate genomic consistency from parental tumors to PDOs is important. We felt that WES might provide more information than STR profiling, and therefore we conducted WES for 15 pair of tumors and PDOs. Genome-wide CNV analysis demonstrated that chromosomal gains and losses of parental tumors were well retained in PDOs for most pairs (Fig. 3G). Similarly, paired tumors and PDOs maintained consistent mutational signature profiles (Supplementary Fig. 7A, B). Furthermore, the cancer related gene mutations in parental tumors, thought to be tumor-specific drivers, were preserved in most paired PDOs (Fig. 3H).

Taken together, our results demonstrated that PDOs could well preserved the EBV latency and genomic alterations of parental tumors.

(13) It is possible that the EC and SC features may be influenced by the EBV latent gene expression, e.g. the heterogeneous LMP2A or LMP1 expression. The authors should examine the EBV transcriptome from the RNA-seq data of these two types of PDO, as well as T55E and T55S. It would be important to check whether there are different acquired mutations in these two PDO sublines. The correlation of somatic genetic alterations and the transcriptome profiles of the two/three subtypes PDOs should be investigated to check whether the transcriptome and phenotypes of these PDOs are driven by somatic alterations or EBV latent gene expression. In addition to the signaling pathways and cellular mechanisms described in Figure 5, the stem cell-related expression profile in these organoids should be explored. The unique transcriptome profile and phenotype of SC or MSEC PDOs may be due to the different passage method by which the stem-cell population was enriched.

Thank you very much for the suggestions. As an EBV latent oncoprotein, LMP1 plays vital functions on NPC development (Tsao et al., 2002), therefore, we examined LMP1 expression levels among NPC subtypes by IHC staining. 16 NPC tumor samples, including 10 EC, 4 MSEC and 2 SC, were examined. The results demonstrated that 13 samples (8 EC, 3 MSEC and 2 SC,) were strongly positive and 3 samples were low positive (2 EC and 1 MSEC) (Data shown below). These data suggest that LMP1 may not have obvious subtype-specific influences on NPC.

Response Figure 1. LMP staining on NPC tumors (n=16).

Loss of *MST1R* was demonstrated to be an EBV infection susceptibility events (Cary et al., 2013; Dai et al., 2016b; Feng et al., 2016), we therefore examined the its deletion frequency among subtypes. As shown in Supplementary Fig. 5C, *MST1R* was deleted in 34/57 EC, 13/26 MSEC, and 9/20 SC. Although EC had a slightly higher percentage than other 2 subtypes, there was no remarkable difference on *MST1R* deletion frequencies among subtypes.

We have conducted RNA-seq on 14 PDOs, including 7 EC, 5 MSEC and 2 SC. The EBV transcriptome mapping reads is low and sporadic in our PDO samples, thus we believe the expression of EBV latent gene has very little effect on the growth of the PDOs.

We also calculated stemness score for 14 PDOs, including 7 EC, 5 MSEC and 2 SC, using transcriptome data. The results indicated that EC subtype PDOs maintained high stemness properties, whereas the opposite phenotype was shown in SC subtype PDOs (Supplementary Fig.12K). Distinctly to EC and SC, a diverse stemness distribution was observed in MSEC PDOs (Supplementary Fig.12K). Since PDOs from MSEC and SC were cultured under same culture condition, but still significant difference was observed among MSEC PDOs. Considering that the intrinsic mixed genomic background of MSEC tumors should contribute influence on derived organoids, we think the observed stemness difference among subtypes might due to their genomics variations of parental tumors rather than later culture method difference on generating PDOs.

(14) As the MSEC type organoids showed morphology changes to SC type organoids,

it suggests that the EC type NPC cells were lost or undergone ENT transition to SC subtype during continuous passage. Thus, it is reasonable the transcriptome profile of MSEC type is similar to that of SC type. The author should discuss this point in their manuscript. It is likely that the epithelial- and spindle-types of NPC cells occur in all EC, MSEC and SC cells, but with different proportions. For the MSEC type NPC, the treatment of MTi or EGFRi may only result in the selection/enrichment of either EC or SC subclones of the tumors in the patients. It may also be happened in the EC and SC subtype of NPC while the tumors contain a small subpopulation of EC and SC cells. A poor response of the NPC patients was reported in a previous clinical trial of EGFRi. The PDO drug screening provides a novel approach for NPC precision treatment. However, it is important to identify a drug that is able to efficiently inhibit the growth of both epithelial- and spindle-type tumor cells. It is noted that the antitumor effect of Cisplatin, a common first line chemo-agent for NPC treatment has not been included in the drug screening panel. It is important to compare the new potential drugs to the standard chemotherapy for NPC. In Figure 6, the treatment of RT+ Cisplatin should be conducted on the PDOs for comparison.

We agree with your comments. Our data indicated the sarcomatoid tumor cells grow faster than epithelial tumor cells among MSEC subtype PDOs, and sarcomatoid tumor cells exhibited obvious EMT phenotype, while epithelial tumor cells were not (Fig. 3D, 3E, 6D). We could observe two distinct populations of cell types among MSEC PDOs in the beginning several passages, but gradually all cells underwent EMT and maintain as SC-type PDOs. The two distinct cell populations were presented with different drug responses for a few drugs, such as MTi and EGFRi (Fig. 4A). However, when combined with RT, both EC and SC type PDOs were demonstrated with good responses (Fig. 5C).

Among EC subtype samples, the majority of tumor regions consist of epithelial cells, although the percentage may not be 100%. Similar results occurred in SC subtype tumors (Fig. 1A-C, S1A). Followed by these observations, our proposed subtype-specific chemotherapy regimens should be effective to eliminate most tumor cells in targeted tumors. Considering CRT is a popular treatment strategy used in NPC, and has been demonstrated to be more effective than either RT alone (He et al., 2017; Xu et al., 2017) or chemotherapy only (You et al., 2020), and RT was effective to both EC and SC subtype PDOs. Our results regarding subtype-specific MTi and EGFRi responses were valuable for pursuing the subtype-guided precision CRT treatment in clinical. Inspired by this scenario, CRT by combining MTi, EGFRi and RT may represent the effective treatment for MSEC subtype samples, by eliminating both epithelial and sarcomatoid tumor cells.

According to our CRT screening results, CRT demonstrated good sensitivity on all EC, MSEC and SC subtype PDOs (Fig. 5C), including two commonly clinically used regimens, cisplatin + RT and fluorouracil + RT. Based on these results, we believed cisplatin + RT and fluorouracil + RT were good CRT options against NPC, irrelevant to subtypes.

Furthermore, when comparing our new proposed subtype-guided chemotherapy and CRT treatment regimens, including MTi/ MTi + RT for SC subtype tumors and EGFRi/ EGFRi + RT for EC subtype tumors, with standard cisplatin/ fluorouracil-based chemotherapy and CRT treatment. The potential subtype-guided treatment strategy achieved better effectiveness than standard regimens in tumors belong to corresponding subtypes (Fig. 5C).

(15) In Figure 7, the somatic changes in PDOs should be further validated. Since there is no hotspot mutations or functional activations of TRKs and other pathways by the CNVs in Figure 7A and 7B, there are no sufficient evidences to support authors' conclusion in the last section. In supplementary Figure 11, the data shows that the copy ratio of MET and KRAS is less than 1.5 and 1. It means that the two genes are with copy number gains, but not amplification. The low level copy number gain may not be able to activate these oncogenes.

Thank you very much for the comments and we agree that the copy ratio of MET and KRAS is about 1.4 and 1.3, respectively, which is indeed with CNV gain (Supplementary Fig. 15B). Multiple studies have demonstrated that *MET* and/or *KRAS* CNV gain were associated with EGFRi resistance (Cappuzzo et al., 2009; Dahabreh et al., 2010; Mekenkamp et al., 2012; Nanjo et al., 2017). In general, high-level amplification gave the best association to active oncogenic activity, however, in many cases, general-level CNV gain also worked. As further evidenced by combinational drug treatment results, we speculate the EGFRi resistance found in this sample was likely due to CNV gains of *MET* and *KRAS*, and after we blocked these two oncogenes with drugs, the EGFRi resistance was overcome.

We have modified our description accordingly in the manuscript (Page 15-16).

Reviewer #2 (Remarks to the Author): Expertise in organoids

NPC is a leading cause of cancer-related deaths in Southeast Asia and North Africa. NPC is currently classified into 4 morphological subtypes that are linked to prognosis. However, the major driving events (genomic and transcriptomic features) for the 4 histological subtypes and the druggable targets among the 4 subtypes are currently unknown. Only few drugs are approved for NPC and many NPCs of the SC type show relapse to current therapies. Better understanding of the driving events of NPC subtypes is needed that can be used to find new subtype-related strategies.

Ding et al. use transcriptomic and genomics on a cohort of 93 nasopharyngeal cancer (NPC) samples to explore subtype-specific molecular and genomic features.

With respect to point mutations, authors classify significantly mutated genes in 9 signaling pathways. Predominant mutational signature across all NPCs was Signature4, a result of smoking.

Next, copy-number alteration analysis revealed that specific genomic deletions and amplifications are consistent features for all NPCs.

Furthermore, they made a patient-derived organoid (PDOs) biobank of 28 NPC samples containing 3 different subtypes (out of 4). PDOs were used to integrate transcriptomics/genomics information underlying subtype classification with functional assays like drug response. Interestingly, NPC PDOs from EC subtype showed sensitivity towards treatment with EGFRi and resistance to treatment with microtubule targeting agents. Opposing sensitivities were found for SC and MSECs.

Overall, the strength of the manuscript lies in its extensive transcriptomic and genomic analyses of NPCs. In addition, new NPC PDOs methods were developed, including the establishment of a PDO biobank. Interestingly, drug screens on the PDOs revealed differentiating drug responses that correlate with morphology-based subtype classification. Despite the impressive amount of resource-like data, the manuscript remains rather descriptive concerning the drug phenotypes and might benefit from supporting underlying mechanistic insights.

Comments:

1. The results regarding the mutational landscape of NPCs is largely descriptive and overinterpreted by clustering mutations in 9 signaling pathways. What I found striking when studying the data is that not a single driver gene seems to be mutated in more than few cases. Even if you subclassify per tumor type, few mutations make it barely over 10%. Is it possible that no clear-cut driver mutations underly NPC development?

In contrast, based on CNA profiles, few genes were consistently deleted (up to 50% of the cases) in NPCs across subtypes. Most of them seem to co-occur in same patients. Are these genes positioned at similar chromosome locations (MST1R, BAP1, TRAF3, NFKB1A etc)? Are these tumor suppressor genes? Rescue experiments by ectopic expression of some of these genes might provide interesting answers. In light of the overinterpretation of the genomic data, I don't think that the systems biology interpretation of the genomic data (Fig. 2D and E) adds much value to the manuscript.

Thank you very much for the comments. We have carefully reanalyzed genomic data using additional 2 mutation callers to identify SNVs, only mutations detected by at least 2 callers were kept as true positive ones and were assigned for further analysis. (Supplementary Fig. 2A). We also added in 13 new cancer samples to increase the total sample number to 106. For SNVs analysis, we excluded 11 unpaired samples, which contributed to low mutational frequencies in previous version. So, our results are of better reliability now. Despite these efforts, we would like to indicate that compared with some other cancer types (Zack et al., 2013), NPC is a cancer type with relatively low mutational rate with averagely 29 somatic SNVs per sample in our cohort and ranging from 22 to 32 per sample in other studies (Chow et al., 2017; Lin et al., 2014; Zhang et al., 2017; Zheng et al., 2016). The driver SNV frequencies identified in our cohort were comparable with previous studies (Li et al., 2017b; Lin et al., 2014; Zhang et al., 2017; Zheng et al., 2016). For example, among somatic SNVs, frequencies of

TP53 were 9.0% (Li et al., 2017b), 10.6% (Lin et al., 2014), 7.3% (Zheng et al., 2016) and 5.9% (Zhang et al., 2017) in previous studies, which is consistent to our result (7.4%). More comparisons on NPC driver SNVs and CNVs frequencies between ours with several previous studies are summarized in Supplementary Table 2, 3, 4, which indicated our current results were of good confidence.

Supplementary Table 2. Comparison of driver SNV frequencies in NPC.

Gene (SNV)	Our study (n=95)	(Li et al., 2017a) (n=111)	(Lin et al., 2014) (n=113)	(Zheng et al., 2016) (n=124)	(Zhang et al., 2017) (n=51)
TP53	7.4%	9.0%	10.6%	7.3%	5.9%
CYLD	9.5%	9.9%	-	3.2%	5.9%
KMT2C	5.3%	5.4%	-	3.2%	2.0%
FBXW7	4.2%	3.6%	-	-	3.9%
NOTCH2	6.3%	2.7%	0.9%	-	-
NFKBIA	4.2%	6.3%	-	4.8%	-
TRAF3	3.2%	8.1%	-	-	5.9%
PTEN	2.1%	3.6%	-	2.4%	-
MTOR	2.1%	1.8%	1.8%	-	-
PIK3C2B	2.1%	-	0.9%	-	-
PIK3CG	2.1%	-	1.8%	-	-
KMT2B	3.2%	1.8%	-	2.4%	-
ARID1A	2.1%	-	4.4%	1.6%	-
NF1	4.2%	1.8%	-	-	-
ATM	3.2%	-	-	1.6%	-
BAP1	2.1%	-	3.5%	3.2%	-
SYNE1	2.1%	-	8.9%	-	2.0%
FAT2	3.2%	-	2.7%	-	-

“-” refers to unmentioned.

CNV profiles provided more insights on exploring subtype-specific drivers, because the mutational frequencies of CNVs were much higher than that of SNVs. After applying strict filtering conditions by overlapping results from two independent callers, sequenza (v3.0.0) (Favero et al., 2015) and CNVkit (v0.9.0) (Talevich et al., 2016), the deletion frequencies of several top frequent genes, including *MST1R*, *BAP1*, *TRAF3* and *NFKBIA*, were identified with 55.6%, 55.6%, 57.5% and 44.3% in our cohort. These CNVs were also demonstrated as NPC drivers by previous genomics studies (Dai et al., 2016b; Li et al., 2017b; Lin et al., 2014; Zhang et al., 2017). *MST1R* and *BAP1* were neighboring genes and located in 3p21.31 and 3p21.1, respectively. Similarly, *TRAF3* and *NFKBIA* located in 14q32.32 and 14q13.2, respectively. Chromosome 3p and 14q were demonstrated to be most recurrently deleted regions in NPC with prevalence ranging from approximately 20% to 75% in different studies (Supplementary Table 3). Considering the extreme high mutational frequencies, losses

of chromosome 3p and 14q were believed to be early events contributing to NPC progression (Dai et al., 2016a), which would explain why *MST1R*, *BAP1*, *TRAF3* and *NFKB1A* often co-occurred in NPC tumors. The concurrent rate of copy number losses regarding these four genes was 36.8% (39/106) and 68 patients harbored at least one of these mutations in our cohort (Fig. 2C).

Inactivation of these four genes previously have been demonstrated as oncogenic drivers for NPC (Dai et al., 2016b; Li et al., 2017b; Lin et al., 2014; Zhang et al., 2017). Among them, *MST1R* was considered with double-side functions, including serving as an oncogenic c-MET related RTK and contributing to cancer susceptibility related to EBV infection. To explore if *MST1R* could affect NPC cell growth, we applied CRISPR-Cas9 to knockout *MST1R* in wildtype PDOs, our results showed that loss of *MST1R* didn't increase the growth of PDOs (Supplementary Fig. 5F), indicating that loss of *MST1R* alone could not affect the growth of NPC organoids.

We agree with your suggestion. For previous Fig. 2D (Protein-protein interaction network analysis), we moved it to supplemental figures (Supplementary Fig. 5G) in this version. For the diagram summarizing subtype-specific mutations and pathways (Fig. 2E), we believed it is useful to illustrate the conclusion of the finding clearly and concisely. So, we prefer to maintain its place as the main figure.

2. The authors demonstrate convincing differences in tumor phenotypes that are recapitulated during organoid growths. In fig 5 gene expression is shown that correlates to MT inhibitor response. Transcriptional profiling of tumors is of interest, but this data should be presented from an unbiased angle.

Thank you very much for the suggestions. We reanalyzed the transcriptional data by firstly performing unbiased gene set enrichment analysis (GSEA) to identify top differentiated pathways between EC- and SC-type PDOs (p value ≤ 0.01 , absolute NES ≥ 1) (Fig. 6B), then differentially expressed genes were identified by DESeq2 (v1.26.0) and limma (v3.42.2) (fold change ≥ 2 , p value ≤ 0.05) (Fig. 6C), and finally were presented in heatmap (Fig. 6D). In consistent with the conclusions in previous version, the microtubule-targeting drugs associated mechanisms, including microtubule signaling, mitotic cell cycle, apoptosis, NF-kB signaling, and androgen response, were among the top differentiated pathways.

3. Using all the genomic/ transcriptomic information, can the authors demonstrate or speculate about the possibility that the tumor subtypes originate from different cell types as origin, rather than accumulation of different driver mutations/ CNA?

Thank you very much for the comments. Our speculation was that different tumor subtypes could be originated from same type of cells, possibly the epithelial cells. In general, NPC is the cancer type arising from the nasopharynx epithelium. As shown in Fig. 2A, D, different subtypes shared a variety of common chromosomal variations with

very high frequencies, such as chromosomal losses of Chr. 3p and 14q, which indicates that the early transformation from epithelium to NPC were probably due to these common genomic alterations across subtypes. During tumor evolution, early NPC continues to acquire more genomic mutations, which might subsequently drive the tumor to evolve into different subtypes.

Further insights came from genomics analysis on MSEC subtype PDOs. We separated EC- and SC-type PDOs from one MSEC tumor and assigned them for WES. The data revealed that EC- and SC-type PDOs share a dozen of common genomic alterations (Fig. 3G; Supplementary Fig. 7C), even though they exhibited distinct histological phenotypes. The distinct two populations derived from MSEC also harbored a few unique mutations with amount less than common mutations, indicating they might be acquired later than common ones and contribute to the differentiation to heterogenous subtypes.

Because cancer origin is quite complex, we would like to point out that although this finding provides evidence that EC, MSEC and SC subtypes could originate from the same clone during tumor evolution, it does not exclude a possibility that they could also derive from different types of cells. The nasopharynx epithelium is very heterogenous and might contain some subtypes of epithelial cells, which could develop into cancers with distinct morphology.

4. The authors show that ECs show higher mutational frequencies in pathways affecting MT polymerization, mitotic spindle checkpoint, DNA repair and ABC transporters, whereas RTK genes show more CNV amplifications in SCs. However, from figure 2C it seems that RTK signaling is deregulated in both subtypes.

RTK signaling is indeed deregulated in both EC and SC subtypes. After reanalyzing, we didn't observe strong subtype specificity on mutational frequencies regarding these three RTK genes, so we withdraw the claiming on RTKs.

5. The authors show that MSECs contain organoids with EC phenotypes as well as organoids with SC phenotypes that show distinct expression of markers for EMT and epithelium.

First, are these MSECs a mixed population of subtypes that originate from the same original tumor cell or are MSECs a polyclonal tumor? Overlap in mutational landscapes between EC and SC organoids from 'same' MSEC tumor might be informative.

Thank you very much for the comments. Our speculation was that both EC-and SC-type populations among MSEC originated from same clone because they shared common genomic alterations (Fig. 3G; Supplementary Fig. 7C) as explained in the answer to question 3).

6. Drug screening in organoids reveals that ECs are sensitive to EGFRi treatment,

whereas SCs and MSECs are sensitive to MTi treatment. The differences in drug responses per subtype is very striking. However, not only MTi show differential effect, same holds true for the topoisomerase inhibitors.

a) Can it be that SCs and MSECs are sensitive to all classic chemotherapies? The authors should expand the list with chemo's with different mechanism of action (cisplatin/ carboplatin/ 5FU) for a number of organoids.

SCs and MSECs are not sensitive to all classic chemotherapies. They are more resistant to EGFR inhibitors, although they more sensitive to Topoisomerase inhibitors and microtubule inhibitors. We tried to understand this by differential gene expression, and copy number variations, etc, which partially explain the results. More studies will certainly be required for a better understanding in future.

We have expanded the drug/CRT screening panel by including in more commonly used chemotherapeutic drugs for NPC (Fig. 5C). Cisplatin and fluorouracil didn't demonstrate obvious subtype specificity when use alone, however they showed better sensitivities on EC subtype PDOs than SCs and MSECs when combined with RT.

b) Drug response was tested with ATP assay, which cannot make the distinction between cell death or cell cycle arrest. Can the authors confirm by microscopy or FACS that organoids are dying or arrested in cell cycle?

Thank you very much for the suggestions. We compared multiple organoid viability detection assays in the beginning, including ATP, WST-1, Alamar Blue, and microscopy observation (Figure attached below). Our evaluation results demonstrated that ATP assay provided the best sensitivity with broad linearity, especially on measuring organoid viability with small amount cells. More importantly, the results of ATP assay reflected the microscopy observation on viable organoids.

Response Figure 2. Comparison among different organoid viability evaluation assays.

c) Drug screening and western blotting was done on 1-day old organoids, which could affect therapy outcomes due to stress. Do authors observe same results on 5-day old organoids?

We treated organoids with drugs starting at 24h after plating cells. For drug screening, the organoids were maintained for growing for additional 4 days together with drug incubation. For western blot, we checked targeted protein responses 1-2 days after drug treatment for studying their immediate response. As suggested, we compared the organoid responses to gefitinib between 1-day old (seeding 3000 cells per well, treat drug after 1 day) and 5-day old organoids (seeding 750 cells per well, treat drug after 5 days). Basically, they exhibited similar drug response pattern (Figure attached below), although 1-day treated organoid showed slightly better response, possibly due to lower cell density per well at the time starting drug intervene.

Response Figure 3. Comparison on gefitinib responses between 1-day old and 5-day old PDOs.

d) The relative IC₅₀ concentrations of EGFRi are very high and are likely not be reached in patients. Can similar results (pERK inhibition) be obtained with concentration ~0,5µM?

Thank you very much for the comments. The IC₅₀ values of gefitinib on sensitive EC-type organoids were approximately 2-4 µM in our study, which showed significant difference with resistant PDOs. Considering gefitinib is approved drug for non-small-cell lung cancer (NSCLC), we compared our results with that of NSCLC cohort in GDSC database (<https://www.cancerrxgene.org/>). Among 66 NSCLC cell lines, those with IC₅₀<4 were among top 20% sensitive cell lines, indicating our results of EGFRi should have feasibly clinical implication.

To investigate whether combining with inhibitors of EGFR downstream could increase EGFRi sensitivity, we employed ERKi (GDC0994, 5µM), AKTi (MK2206, 0.5µM) and STAT3i (C188-9, 10µM) to co-treat PDO63S with gefitinib. The results demonstrated that AKTi significantly increased EGFRi response to PDO63S, while ERKi and STAT3i could also slightly enhanced EGFRi sensitivity. When combining three inhibitors

together, EGFRi resistance in PDO86 could be overcome (Supplementary Fig. 9C-F).

e) EGFRi-resistant SCs show very low ERK activity during unperturbed conditions compared to ECs, suggesting that this pathway is not essential (hence the resistance upon inhibiting something that is not active). Can the authors confirm that limited baseline ERK levels correlate with no response to EGFRi in SC PDOs?

We fully agree that p-ERK levels were relatively low in the majority of EGFRi resistant SC and MSEC PDOs, whereas all 6 tested sensitive EC PDOs showed active p-ERK during unperturbed conditions (Fig. 4F; Supplementary Fig. 9A). We tested PDO63S and PDO86 on combinational treatment of EGFRi+ERKi, the result indicated that increased response to EGFRi was only observed in PDO63S (Supplementary Fig. 9C), but not in PDO86. So, we believed inhibition of ERKi might not be helpful to overcome EGFRi resistance for some of SCs.

f) In relation to point 4: low basal pERK activity as detected in SC PDOs (Fig 4F) is at odds with the statement of deregulated RTK signaling in this subtype.

Thank you very much for the comments. In the new version of manuscript, we have withdrawn our previous claims on RTK as a subtype-specific feature, because we didn't observe strong subtype specificity on mutational frequencies regarding RTK genes after reanalyzing the data with more strict mutation calling strategies.

In addition, RTKs have various downstream signaling and activation of MAPK/ERK is just one of them, it is possible that tumor have activation on upstreaming RTKs but bypassed MAPK/ERK signaling to support proliferation and survival.

7. The authors attribute the MTi resistant phenotype of ECs to CNVs in various signaling pathways. However, most significantly different CNVs between ECs and SCs affect ABC transporters. Are ABC transporters involved in drug resistance? Can authors block ABC transporters with inhibitors to see if enhanced sensitivity to MTi are established in ECs? Furthermore, in relation to point 6a, do ECs show a general resistant phenotype to most classical chemo's or only MTi? Perhaps ABC transporters underly most drug phenotypes?

Similar to RTKs, the ABC transporter genes didn't show with strongly subtype-specificity in updated version, so we withdraw the previous claims on ABC transporters and didn't go further to investigate their functions on overcoming MTi resistance in NPC.

8. The authors showed that ECs were more sensitive to irradiation than SC and MSECs. In addition, they show that EGFRi + irradiation show synergistic effects in ECs. Is MTi + irradiation relevant in SCs?

Thank very much for the comments. MTi alone were very potent for SC-type PDOs. After combining IR with MTi, we further observed better killing effect with averagely 2.6- and 2.9-fold change increase on IC50 value for docetaxel and paclitaxel on SC-type PDOs (Fig. 5C; Supplementary Data 4), but without synergistic effect for the majority of cases.

9. The authors show that treatment of SCs with EGFRi, HDACi and METi induces a synergistic effect. However, effect of individual drug responses should be shown, as well dual combinations in order to evaluate synergism of three combined.

Thank you very much for the comments. We showed these drug response curves as suggested in Supplementary Fig. 15C-E.

Reviewer #3 (Remarks to the Author): Expertise in organoids drug screens

In this manuscript, Ding Reng-bo et al acquired tumors from 93 NSC patients and classified them into four different subtypes EC, SC, MSEC and SCC based on histology and molecular markers. In addition, they used genomic analysis to begin to investigate the molecular basis for the different subtypes.

The authors also report generation of a library of ~28 organoids and identify feature that relate them to some of the NPC subtypes. They used the organoids test for sensitivities to drugs and found subtype differences in drug response, where EC-subtype PDOs are sensitive to EGFRi, while SC-subtype are sensitive towards MTs. Interestingly the EC subtype organoids were more sensitive when treated with EGFRi in combination with radiotherapy (RT). Overall this is an interesting study that requires further development before it can be considered for publication in Nature Communication.

Major concerns:

1) This study expands previous genomic analysis of NPC. Although the authors attempted to understand subtype specific differences in genetic alternations, the analysis did not provide an compelling argument for a relationship between genomic signatures and histopathological subtype classification perhaps due to low sample number representing some of the subtypes and hence statistical rigor. Most of the analysis presented in Figures 1 and 2 are weak and may need to be revisited and reduced in scope.

Thank you very much for the comments. Yes, we tried to understand subtype specific differences from multiple aspects, including genomics, gene expression, organoid growth patterns, and drug responses. We agree that there is lack on strong correlation between genomic signatures and histopathological subtype classification. We have very carefully reanalyzed our genomic data by using more software/programs and also added

13 new samples. We believed that this is perhaps mainly due to the fact that NPC intrinsically has low mutation rate as revealed by us and others (Supplementary Table 2). We believe our revised data are reliable which are shown in Figures 1 and 2. Because our sample number is comparable with others, therefore, we suspect that this mutation rate might not change significantly even if some more samples are added. On the other hand, our study revealed that different NPC subtypes exhibited stronger pattern of differential gene expression. For example, the EC subtype, besides expressing epithelial markers, such as AE1/3, CK5/6 and P63, it also displayed more obviously microtubule polymerization activation, defective mitotic spindle checkpoint regulation and DNA repair. SC was associated with EMT/invasion promoting signaling, whereas the MSEC was a more heterogeneous subtype with molecular features of both EC and SC.

Supplementary Table 2. Comparison of driver SNV frequencies in NPC.

Gene (SNV)	Our study (n=95)	(Li et al., 2017a) (n=111)	(Lin et al., 2014) (n=113)	(Zheng et al., 2016) (n=124)	(Zhang et al., 2017) (n=51)
TP53	7.4%	9.0%	10.6%	7.3%	5.9%
CYLD	9.5%	9.9%	-	3.2%	5.9%
KMT2C	5.3%	5.4%	-	3.2%	2.0%
FBXW7	4.2%	3.6%	-	-	3.9%
NOTCH2	6.3%	2.7%	0.9%	-	-
NFKBIA	4.2%	6.3%	-	4.8%	-
TRAF3	3.2%	8.1%	-	-	5.9%
PTEN	2.1%	3.6%	-	2.4%	-
MTOR	2.1%	1.8%	1.8%	-	-
PIK3C2B	2.1%	-	0.9%	-	-
PIK3CG	2.1%	-	1.8%	-	-
KMT2B	3.2%	1.8%	-	2.4%	-
ARID1A	2.1%	-	4.4%	1.6%	-
NF1	4.2%	1.8%	-	-	-
ATM	3.2%	-	-	1.6%	-
BAP1	2.1%	-	3.5%	3.2%	-
SYNE1	2.1%	-	8.9%	-	2.0%
FAT2	3.2%	-	2.7%	-	-

“-” refers to unmentioned.

2) The development and use organoid models is interesting and an important new contribution to the field. However, it is not clear how well the expression profile performed in the 14 organoid models relate to the gene expression profile of matched primary patient tumors. Furthermore, it would be important for the authors to investigate how the drug response observed in organoid models respond the treatment received by the patient to understand how well the organoid models replicate the patient's clinical response.

Thank you very much for the comments. Although RNA sequencing was only performed on PDOs rather than primary tumors, we believed our transcriptional results could well reflect the gene expression profile of matched primary tumors. Except that PDOs could well recapitulate the histological features (Fig. 3A), the expression of molecular markers (Fig. 3D, 3E), and the EBV latency (Fig. 3F), we also demonstrated that PDOs preserved genomic features of parental tumors as evidenced by WES on paired tumors and PDOs (Fig. 3G, H; Supplementary Fig.7A, B). Furthermore, the top differential pathways uncovered by transcriptional analysis on PDOs were consistent to subtype-specific aberrant genomics signaling revealed by WES on primary tumors. Taken together, we believed PDOs could serve as reliable proxies of parental tumors on both transcriptional study and patient's clinical response indication.

3) The comparison between the PDO sensitivity and GDSC database is of unknown significance because there are many confounding factors that differentiate the two sets of observations, which include culture conditions, tumor type, cell lines used etc.,

Thank you very much for the comments. We agree that the ideal dataset for drug prediction evaluation should come from NPC patients. However, after searching all available datasets from literatures, GEO and TCGA databases, we couldn't find any perfect matched source containing both transcriptome and MTi treatment response in NPC patients. Alternatively, we used datasets of breast cancer patients, breast cancer PDXs and pan-cancer cell lines to examine whether the gene signature derived from our PDOs could predict MTi response. Theoretically, the MTi sensitivity should be determined by specific gene profiles rather than cancer types, thus, our validation on breast cancer patients should reflect the situation on NPC patients with good confidence.

4) The relationship between EC subtype and sensitivity to CRT is interesting and can be significant clinical benefit. However, a pharmacogenomic analysis to tailor such a treatment may not be needed as a combination of histopathological and IHC based identification of EC subtype may be sufficient. It would be informative and important for field, if the authors can demonstrate a correlation between EC subtype and clinical response to CRT in a retrospective analysis using a classification system that uses IHC and histopathology.

Thank you very much for the suggestions. We agree that retrospective analysis is very important to confirm that the proposed subtype-guided treatment indeed works in clinical. We analyzed a retrospective cohort containing 3893 NPC patients on the RT and CRT responses among subtypes (Supplementary Fig. 10A, B) (Wang et al., 2016). In consistent with our finding based on PDOs that ECs were more sensitive to RT and CRT (cisplatin +RT and fluorouracil +RT) than SCs, the Kaplan-Meier survival analysis demonstrated that EC patients live significantly longer than SC patients (Supplementary Fig. 10A, B).

5) A mechanistic insight into how do MT and EGFR inhibitors affect organoids growth

will be necessary. Perhaps this relates to inhibition of one or more pathways identified by genomic analysis.

We have attempted to investigate this by analyzing responses of PDOs to MTi and EGFRi. Our data revealed that EGFRi (gefitinib) significantly reduced both p-AKT and p-ERK in all sensitive PDOs, but not in resistant PDOs. This observation suggests that the MAPK and PI3K-AKT signaling pathways might mediate efficiency of EGFRi in NPC (Fig. 4F; Supplementary Fig. 9B). Combinational treatment with AKT inhibitor MK2206, ERK inhibitor GDC0994, STAT3 inhibitor C188-9 and gefitinib could overcome the EGFRi resistance in PDO63S (Supplementary Fig. 9C-F). On the other hand, MTi (docetaxel) treatment increased the expression of mitotic checkpoint proteins (cyclin A, and cyclin B), degraded the anti-apoptotic protein BCL2 through phosphorylation, and activated pro-apoptotic proteins such as BAX and cleaved caspase-3 (CASP3) in sensitive PDOs, while no significant changes in these proteins were observed in resistant PDOs (Fig. 4G; Supplementary Fig. 9B).

Our genomic analysis identified *TPPP*, *MAP2* and *SKP2* as EC-specific copy number gains (Fig. 2C, E) and their activation might contribute to MTi resistance (Fig. 7A). Thus, we applied CRISPR-Cas9 system to knockout microtubule associated *TPPP* and *MAP2*, and mitosis related *SKP2* in MTi resistant wildtype PDOs. Our results indicated that *TPPP*, *MAP2* and *SKP2* knockout organoids showed sensitive responses to docetaxel treatment when comparing with parental organoids (Fig. 7C, D; Supplementary Fig. 14D).

6) The manuscript needs significant editing for English usage as there are major grammatical errors which makes it hard for the reader to comprehend and follow.

Thank you very much for the suggestion. We have sent our manuscript to American Journal Experts for English editing.

Minor

1. Fig1A, B. Images of better quality and resolution should be provided. The

Histological features claimed by the authors are not clear.

Thank you very much for the suggestion. We believe it might be due to the technique problem on file compression step during submission, which caused the low-resolution of our images. We will pay more attention this time, try to upload the file as bigger as possible.

2. Fig 3 – how were the 28 organoids chosen for PDO analysis? Provide better phase images for Fig 3A.

There was no selection for deriving organoids from NPC. We basically cultured PDOs from all fresh NPCs provided by the Hospital, except a few samples, which did not have enough cells for culture. Because the biopsy sample size is relatively small, in many cases, the hospitals didn't have enough extra samples for us to culture, so they only provided us with the FFPE samples/slides for WES. In this revision, our number of PDO lines increased to 42, which were derived from 40 patients (2 MSECs have both EC- and SC-type PDOs).

We will pay more attention on image compression issue this time.

3. Fig 3D – Provide images showing that the spindle type organoids generated using the aggregate method and stained for pan-epithelial markers.

Thank you very much for the suggestion. The representative image showing SC-type organoid generated using aggregated method were shown in Fig. 3C at the lower right panel, it was taken 24h after organoid embedment in Matrigel. In the beginning of embedment, the surfaces of organoids were smooth, and then the invasive protrusions gradually come out and invade in Matrigel.

We stained the PDOs with AE1/3 (Fig. 3D), the results demonstrated that EC-type organoids were positive to AE1/3, whereas SC-type organoid show negative.

4. Do the tumor cells restart growth after drug withdrawal?

Thank you very much for the comments, we did comparison experiment between directly examining organoid viability after 4-day gefitinib incubation and following the procedure that treating organoids with gefitinib for 4 days, then withdrawing drugs by refreshing medium and culturing for another 4 days, finally examining the viability at day 9. Our results demonstrated that the gefitinib treated organoids in drug withdrawing group exhibited even lower survival rate than the group examined right after 4-day drug incubation, indicating tumor cell may not be able to restart growth at regular rate after drug withdrawing, instead they might undergo continuously apoptosis induced by previous drug treatment.

Response Figure 4. Comparison of organoid survival between with and without gefitinib withdrawal on PDOs.

5. Can the authors provide insight into the genomics of cells remaining in PDO cultures that resist treatments?

Thank you very much for the comments. We believe those drug-associated genomic alterations were well preserved in corresponding PDOs. 1) uncovered by WES comparison on 15 pairs of tumors and PDOs, chromosomal gains and losses of parental tumors were well retained in PDOs as well as a number of tumor-specific mutations (Fig. 3G, H); 2) genomics analysis on tumors revealed that microtubule-targeting drugs associated mechanisms, including microtubule polymerization, mitotic cell cycle, apoptosis, NF- κ B signaling, were subtype-specific driver pathways (Fig. 2C,E). In line with this, transcriptomic analysis on PDOs also come out with identical subtype-specific pathways (Fig.6B). 3) we applied CRISPR-Cas9 system to knockout microtubule associated *TPPP* and *MAP2*, and mitosis related *SKP2* in MTi resistant wildtype PDOs. These gene were identified as EC-specific CNVs (Fig. 2C, E) and their activation might contribute to MTi resistance (Fig. 7A). Our results indicated that *TPPP*, *MAP2* and *SKP2* knockout organoids both showed more sensitive responses to docetaxel treatment when comparing to parental organoids (Fig. 7C, D; Supplementary Fig. 14D)

References cited:

Alexandrov, L. B., Ju, Y. S., Haase, K., Van Loo, P., Martincorena, I., Nik-Zainal, S., Totoki, Y., Fujimoto, A., Nakagawa, H., Shibata, T., *et al.* (2016). Mutational signatures associated with tobacco smoking in human cancer. *Science (New York, NY)* 354, 618-622.

Cappuzzo, F., Jänne, P. A., Skokan, M., Finocchiaro, G., Rossi, E., Ligorio, C., Zucali, P. A., Terracciano, L., Toschi, L., Roncalli, M., *et al.* (2009). MET increased gene copy number and primary resistance to gefitinib therapy in non-small-cell lung cancer patients. *Annals of oncology : official journal of the European Society for Medical Oncology* 20, 298-304.

Cary, D. C., Clements, J. E., and Henderson, A. J. (2013). RON receptor tyrosine kinase, a negative regulator of inflammation, is decreased during simian immunodeficiency virus-associated central nervous system disease. *J Immunol* *191*, 4280-4287.

Chen, C., Yi, W., Gao, J., Li, X. H., Shen, L. J., Li, B. F., Tu, Z. W., Tao, Y. L., Jiang, C. B., and Xia, Y. F. (2014). Alternative endpoints to the 5-year overall survival and locoregional control for nasopharyngeal carcinoma: A retrospective analysis of 2,450 patients. *Mol Clin Oncol* *2*, 385-392.

Cheng, H. L., Liu, H. S., Lin, Y. J., Chen, H. H., Hsu, P. Y., Chang, T. Y., Ho, C. L., Tzai, T. S., and Chow, N. H. (2005). Co-expression of RON and MET is a prognostic indicator for patients with transitional-cell carcinoma of the bladder. *Br J Cancer* *92*, 1906-1914.

Chow, Y. P., Tan, L. P., Chai, S. J., Abdul Aziz, N., Choo, S. W., Lim, P. V., Pathmanathan, R., Mohd Kornain, N. K., Lum, C. L., Pua, K. C., *et al.* (2017). Exome Sequencing Identifies Potentially Druggable Mutations in Nasopharyngeal Carcinoma. *Sci Rep* *7*, 42980.

Chua, D. T., Sham, J. S., Kwong, D. L., and Au, G. K. (2003). Treatment outcome after radiotherapy alone for patients with Stage I-II nasopharyngeal carcinoma. *Cancer* *98*, 74-80.

Chung, A. K., OuYang, C. N., Liu, H., Chao, M., Luo, J. D., Lee, C. Y., Lu, Y. J., Chung, I. C., Chen, L. C., Wu, S. M., *et al.* (2019). Targeted sequencing of cancer-related genes in nasopharyngeal carcinoma identifies mutations in the TGF- β pathway. *Cancer medicine* *8*, 5116-5127.

Dahabreh, I. J., Linardou, H., Siannis, F., Kosmidis, P., Bafaloukos, D., and Murray, S. (2010). Somatic EGFR mutation and gene copy gain as predictive biomarkers for response to tyrosine kinase inhibitors in non-small cell lung cancer. *Clinical cancer research : an official journal of the American Association for Cancer Research* *16*, 291-303.

Dai, W., Zheng, H., Cheung, A. K., and Lung, M. L. (2016a). Genetic and epigenetic landscape of nasopharyngeal carcinoma. *Chinese clinical oncology* *5*, 16.

Dai, W., Zheng, H., Cheung, A. K., Tang, C. S., Ko, J. M., Wong, B. W., Leong, M. M., Sham, P. C., Cheung, F., Kwong, D. L., *et al.* (2016b). Whole-exome sequencing identifies MST1R as a genetic susceptibility gene in nasopharyngeal carcinoma. *Proc Natl Acad Sci U S A* *113*, 3317-3322.

Favero, F., Joshi, T., Marquard, A. M., Birkbak, N. J., Krzystanek, M., Li, Q., Szallasi, Z., and Eklund, A. C. (2015). Sequenza: allele-specific copy number and mutation profiles from tumor sequencing data. *Ann Oncol* *26*, 64-70.

Feng, T., Gan, J., Qin, A., Huang, X., Wu, N., Hu, H., and Yao, H. (2016). HIV1 downregulates the expression and phosphorylation of receptor tyrosine kinase by targeting the NFkappaB pathway. *Mol Med Rep* *14*, 1947-1952.

He, J., Wu, P., Tang, Y., Liu, S., Xie, C., Luo, S., Zeng, J., Xu, J., and Zhao, S. (2017). Chemoradiotherapy enhanced the efficacy of radiotherapy in nasopharyngeal carcinoma patients: a network meta-analysis. *Oncotarget* *8*, 39782-39794.

Kim, S., Scheffler, K., Halpern, A. L., Bekritsky, M. A., Noh, E., Kallberg, M., Chen, X., Kim, Y., Beyter, D., Krusche, P., and Saunders, C. T. (2018). Strelka2: fast and

accurate calling of germline and somatic variants. *Nat Methods* 15, 591-594.

Lee, A. W., Sze, W. M., Au, J. S., Leung, S. F., Leung, T. W., Chua, D. T., Zee, B. C., Law, S. C., Teo, P. M., Tung, S. Y., *et al.* (2005). Treatment results for nasopharyngeal carcinoma in the modern era: the Hong Kong experience. *Int J Radiat Oncol Biol Phys* 61, 1107-1116.

Li, Y. Y., Chung, G. T., Lui, V. W., To, K. F., Ma, B. B., Chow, C., Woo, J. K., Yip, K. Y., Seo, J., Hui, E. P., *et al.* (2017a). Exome and genome sequencing of nasopharynx cancer identifies NF-kappaB pathway activating mutations. *Nat Commun* 8, 14121.

Li, Y. Y., Chung, G. T., Lui, V. W., To, K. F., Ma, B. B., Chow, C., Woo, J. K., Yip, K. Y., Seo, J., Hui, E. P., *et al.* (2017b). Exome and genome sequencing of nasopharynx cancer identifies NF-κB pathway activating mutations. *Nat Commun* 8, 14121.

Lin, D. C., Meng, X., Hazawa, M., Nagata, Y., Varela, A. M., Xu, L., Sato, Y., Liu, L. Z., Ding, L. W., Sharma, A., *et al.* (2014). The genomic landscape of nasopharyngeal carcinoma. *Nat Genet* 46, 866-871.

Lo, Y. M., Chan, L. Y., Lo, K. W., Leung, S. F., Zhang, J., Chan, A. T., Lee, J. C., Hjelm, N. M., Johnson, P. J., and Huang, D. P. (1999). Quantitative analysis of cell-free Epstein-Barr virus DNA in plasma of patients with nasopharyngeal carcinoma. *Cancer Res* 59, 1188-1191.

Ma, S. X., Zhou, T., Huang, Y., Yang, Y. P., Zhan, J. H., Zhang, Y. X., Zhang, Z. H., Zhao, Y. Y., Fang, W. F., Ma, Y. X., *et al.* (2018). The efficacy of first-line chemotherapy in recurrent or metastatic nasopharyngeal carcinoma: a systematic review and meta-analysis. *Ann Transl Med* 6, 201.

Maggiara, P., Lorenzato, A., Fracchioli, S., Costa, B., Castagnaro, M., Arisio, R., Katsaros, D., Massobrio, M., Comoglio, P. M., and Flavia Di Renzo, M. (2003). The RON and MET oncogenes are co-expressed in human ovarian carcinomas and cooperate in activating invasiveness. *Exp Cell Res* 288, 382-389.

Mekenkamp, L. J., Tol, J., Dijkstra, J. R., de Krijger, I., Vink-Börger, M. E., van Vliet, S., Teerenstra, S., Kamping, E., Verwiel, E., Koopman, M., *et al.* (2012). Beyond KRAS mutation status: influence of KRAS copy number status and microRNAs on clinical outcome to cetuximab in metastatic colorectal cancer patients. *BMC cancer* 12, 292.

Mermel, C. H., Schumacher, S. E., Hill, B., Meyerson, M. L., Beroukhim, R., and Getz, G. (2011). GISTIC2.0 facilitates sensitive and confident localization of the targets of focal somatic copy-number alteration in human cancers. *Genome Biol* 12, R41.

Nanjo, S., Arai, S., Wang, W., Takeuchi, S., Yamada, T., Hata, A., Katakami, N., Okada, Y., and Yano, S. (2017). MET Copy Number Gain Is Associated with Gefitinib Resistance in Leptomeningeal Carcinomatosis of EGFR-mutant Lung Cancer. *Mol Cancer Ther* 16, 506-515.

Narzisi, G., Corvelo, A., Arora, K., Bergmann, E. A., Shah, M., Musunuri, R., Emde, A. K., Robine, N., Vacic, V., and Zody, M. C. (2018). Genome-wide somatic variant calling using localized colored de Bruijn graphs. *Commun Biol* 1.

Perri, F., Della Vittoria Scarpato, G., Caponigro, F., Ionna, F., Longo, F., Buonopane, S., Muto, P., Di Marzo, M., Pisconti, S., and Solla, R. (2019). Management of recurrent nasopharyngeal carcinoma: current perspectives. *Onco Targets Ther* 12, 1583-1591.

Stelow, E. B., and Wenig, B. M. (2017). Update From The 4th Edition of the World

Health Organization Classification of Head and Neck Tumours: Nasopharynx. *Head Neck Pathol* 11, 16-22.

Talevich, E., Shain, A. H., Botton, T., and Bastian, B. C. (2016). CNVkit: Genome-Wide Copy Number Detection and Visualization from Targeted DNA Sequencing. *PLoS Comput Biol* 12, e1004873.

Tsao, S. W., Tramoutanis, G., Dawson, C. W., Lo, A. K., and Huang, D. P. (2002). The significance of LMP1 expression in nasopharyngeal carcinoma. *Seminars in cancer biology* 12, 473-487.

Wang, H. Y., Chang, Y. L., To, K. F., Hwang, J. S., Mai, H. Q., Feng, Y. F., Chang, E. T., Wang, C. P., Kam, M. K., Cheah, S. L., *et al.* (2016). A new prognostic histopathologic classification of nasopharyngeal carcinoma. *Chinese journal of cancer* 35, 41.

Wang, M. H., Yao, H. P., and Zhou, Y. Q. (2006). Oncogenesis of RON receptor tyrosine kinase: a molecular target for malignant epithelial cancers. *Acta Pharmacol Sin* 27, 641-650.

Xu, C., Zhang, L. H., Chen, Y. P., Liu, X., Zhou, G. Q., Lin, A. H., Sun, Y., and Ma, J. (2017). Chemoradiotherapy Versus Radiotherapy Alone in Stage II Nasopharyngeal Carcinoma: A Systemic Review and Meta-analysis of 2138 Patients. *J Cancer* 8, 287-297.

Yang, S. Y., Nguyen, T. T., Ung, T. T., and Jung, Y. D. (2017). Role of Recepteur D'origine Nantais on Gastric Cancer Development and Progression. *Chonnam Med J* 53, 178-186.

Yi, J. L., Gao, L., Huang, X. D., Li, S. Y., Luo, J. W., Cai, W. M., Xiao, J. P., and Xu, G. Z. (2006). Nasopharyngeal carcinoma treated by radical radiotherapy alone: Ten-year experience of a single institution. *Int J Radiat Oncol Biol Phys* 65, 161-168.

You, R., Liu, Y. P., Huang, P. Y., Zou, X., Sun, R., He, Y. X., Wu, Y. S., Shen, G. P., Zhang, H. D., Duan, C. Y., *et al.* (2020). Efficacy and Safety of Locoregional Radiotherapy With Chemotherapy vs Chemotherapy Alone in De Novo Metastatic Nasopharyngeal Carcinoma: A Multicenter Phase 3 Randomized Clinical Trial. *JAMA oncology*.

Zack, T. I., Schumacher, S. E., Carter, S. L., Cherniack, A. D., Saksena, G., Tabak, B., Lawrence, M. S., Zhsng, C. Z., Wala, J., Mermel, C. H., *et al.* (2013). Pan-cancer patterns of somatic copy number alteration. *Nat Genet* 45, 1134-1140.

Zhang, L., MacIsaac, K. D., Zhou, T., Huang, P. Y., Xin, C., Dobson, J. R., Yu, K., Chiang, D. Y., Fan, Y., Pelletier, M., *et al.* (2017). Genomic Analysis of Nasopharyngeal Carcinoma Reveals TME-Based Subtypes. *Mol Cancer Res* 15, 1722-1732.

Zheng, H., Dai, W., Cheung, A. K., Ko, J. M., Kan, R., Wong, B. W., Leong, M. M., Deng, M., Kwok, T. C., Chan, J. Y., *et al.* (2016). Whole-exome sequencing identifies multiple loss-of-function mutations of NF- κ B pathway regulators in nasopharyngeal carcinoma. *Proc Natl Acad Sci U S A* 113, 11283-11288.

Reviewers' comments:

Reviewer #1 (Remarks to the Author):

In general, the poor quality of the genome data and lack of precise genomic profile supporting the NPC subtypes are my major concerns of this paper. As I indicated in last reviewer's report, the tumor samples in their study may have low cellularity and problems in informatics pipelines while orthogonal validation of the mutations should be performed in these tumors. In this revised version, the authors have not properly addressed my comments, evaluated their data and interpret the findings. The poor sequencing data quality were still observed in significant number of samples (supplementary table S1, S2 and S10). The CNV data is wrongly interpreted to fit their hypothesis of NPC-subtypes. In page 8 (line 229-239), the authors misinterpret their data by mixing up the concept of functional mutations and copy number gains/losses with unknown significant of a large list of genes involved in multiple pathways, further misleading us their roles in driving various features among NPC subtypes. The "Pharmacogenomics-based precision medicine" strategy shows multiple fundamental concept problems. The proposed treatments will only cause the selection of resistant cell population and rapid disease recurrence in these heterozygous tumors.

Reviewer #2 (Remarks to the Author):

Ren-Bo Ding's original manuscript provide an impressive amount of genomic and transcriptomic data of NPC subtypes, as well as new organoid derivation methods on which extensive drug responses have been performed and analyzed. The interested finding is that the drug screens revealed different sensitivities per morphology-based subtype classification.

The resource-like datasets will be of interest to the field, as well as the new methodology to grow PDOs. Although details or mechanistic underpinning remains challenging, the authors convince that PDOs shows differential drug sensitivity per NPC subtype. The manuscript will be helpful for the ongoing debate about NPC subtypes and treatments.

Minor comments: the figure legends are very concise and do not explain all features that can be seen in the panels (e.g. Fig5D-G, what is the additive?)

Reviewer #3 (Remarks to the Author):

In this manuscript the authors analyzed a large cohort of NPC cancers and generated more than 40 patient derived organoids models to develop pharmacogenomic models for drug response. This is well done study.

My only concern is that the authors do not rule out the possibility the response to chemotherapies and CRT treatment (Figure 4 and Figure 5) relates to difference in the proliferative potential of the PDO models. In other words are the most responsive models more proliferative than those that do not respond well? The authors should determine doubling time for a panel of responders and non-responders to chemotherapies to determine the correlation, if any, between proliferative potential and response to chemotherapies.

Response to Reviewers

Response to Comments of Reviewer 1

In general, the poor quality of the genome data and lack of precise genomic profile supporting the NPC subtypes are my major concerns of this paper. As I indicated in last reviewer's report, the tumor samples in their study may have low cellularity and problems in informatics pipelines while orthogonal validation of the mutations should be performed in these tumors. In this revised version, the authors have not properly addressed my comments, evaluated their data and interpret the findings. The poor sequencing data quality were still observed in significant number of samples (supplementary table S1, S2 and S10). The CNV data is wrongly interpreted to fit their hypothesis of NPC-subtypes. In page 8 (line 229-239), the authors misinterpret their data by mixing up the concept of functional mutations and copy number gains/losses with unknown significant of a large list of genes involved in multiple pathways, further misleading us their roles in driving various features among NPC subtypes. The "Pharmacogenomics-based precision medicine" strategy shows multiple fundamental concept problems. The proposed treatments will only cause the selection of resistant cell population and rapid disease recurrence in these heterozygous tumors.

We thank the reviewer for the constructive comments. To facilitate our responses to all comments in details, we have divided them into the following 4 points. Meanwhile, we feel we must express, respectfully, our different opinions on some of the comments.

1. In general, the poor quality of the genome data and lack of precise genomic profile supporting the NPC subtypes are my major concerns of this paper. As I indicated in last reviewer's report, the tumor samples in their study may have low cellularity and problems in informatics pipelines while orthogonal validation of the mutations should be performed in these tumors. In this revised version, the authors have not properly addressed my comments, evaluated their data and interpret the findings. The poor sequencing data quality were still observed in significant number of samples (supplementary table S1, S2 and S10).

We would like to thank reviewer very much for raising these comments, which prompt us to reexamine our data carefully. After analyzing our data together with quality control, which are outlined below, we believe our data are reliable and of solid quality.

1). For addressing on "low cellularity" concern, we assessed tumor purity using Sequenza and Absolute methods as described in "Materials and Methods" at page 24. The methods are available from the previously published Annals of Oncology (Favero et al., 2015) and Nature Biotechnology papers (Carter et al., 2012). We also prepared H&E staining for the tumors (Supplementary Data 2). In general, the tumor purity assessment by Sequenza and Absolute reflected the tumor cellularity status of tumor samples assigned for bulk sequencing (Supplementary Data 1, 2). Our results

demonstrated that the average tumor purity of bulk sequencing samples were 0.58 by Sequenza and 0.55 by Absolute (Supplementary Data 1), which should be able to provide reliable mutation calling.

2). For addressing on “poor genome data quality due to low cellularity” concern, we did several quality control analyses for association between tumor purity and mutation calling.

A). We found that the number of SNVs didn't directly correlate with the tumor purity in our NPC cohort, as revealed by Sequenza and Absolute (Response Figure 1). In other words, sample with higher tumor purity was not equivalent to sample detected with more SNVs.

Response Figure 1

Response Figure 1. The association between SNV amount and tumor purity. No obvious correlation was observed. The tumor purity was assessed by Sequenza (A) and Absolute (B).

B). We compared mutational signatures and top driver SNVs between total cohort and lower purity cohort (tumor purity < median purity value, n=44). The results showed that these two groups exhibited highly similar mutational signatures (Response Figure 2) and consistent driver mutation frequencies (Response Table 1).

Response Figure 2

Response Figure 2. Comparison of mutational signatures between lower purity cohort and total cohort. (A) The mutational signature of lower purity cohort (tumor purity < median purity value, n=44). (B) The mutational signature of total cohort (n=88). There was no significant difference on mutational signatures between lower purity cohort and total cohort.

Response Table 1. Comparison of driver SNV frequencies in NPC.

Gene (SNV)	Total cohort (n=88)	Lower purity cohort (n=44)	Micro-dissected cohort (n=44)	(Li et al., 2017) (n=111)	(Lin et al., 2014) (n=113)	(Zheng et al., 2016) (n=124)	(Zhang et al., 2017) (n=51)
TP53	8.0%	6.8%	9.1%	9.0%	10.6%	7.3%	5.9%
CYLD	10.2%	4.5%	11.4%	9.9%	-	3.2%	5.9%
KMT2C	5.7%	6.8%	9.1%	5.4%	-	3.2%	2.0%
FBXW7	4.5%	6.8%	6.8%	3.6%	-	-	3.9%
NOTCH2	6.8%	6.8%	11.4%	2.7%	0.9%	-	-
NFKBIA	4.5%	6.8%	2.3%	6.3%	-	4.8%	-
TRAF3	3.4%	2.3%	4.5%	8.1%	-	-	5.9%
KMT2B	3.4%	4.5%	2.3%	1.8%	-	2.4%	-
NF1	4.5%	6.8%	4.5%	1.8%	-	-	-
FAT2	3.4%	2.3%	4.5%	-	2.7%	-	-

“-” refers to unmentioned.

C). NPC intrinsically has less SNVs when comparing with some other solid cancers (such as breast cancer, colon cancer, etc). We have compared our SNVs results with previous NPC publications. The average SNV amount per sample in our cohort is 30.3, which is consistent to previous NPC studies ranging from 22 to 32 (Chow et al., 2017;

Lin et al., 2014; Zhang et al., 2017; Zheng et al., 2016). In these published studies, it is commonly found that a very few SNVs were detected in some of NPC tumors. For example, in the Nature Genetics paper, 17.9% tumors were identified with small number of SNVs ranging from 0 to 5 by WES, and the average SNV amount per tumor was 28.2 (Lin et al., 2014). Thus, by comparing with previous studies, we believed that the SNV frequencies in our study were equivalent to other NPC studies (Response Table 1).

D). We have also specifically compared the top recurrent chromosomal loss (Response Table 2), and the top recurrent chromosomal gain (Response Table 3) detected in the lower purity cohort of our samples with previous published studies, the frequencies of these driver mutations were comparable, suggesting the mutation calling results of our study is of good reliability.

Response Table 2. Comparison of recurrent chromosomal loss frequencies in NPC.

		Total cohort (n=99)	Lower purity cohort (n=44)	(Li et al., 2017) (n=111)	(Chen et al., 1999) (n=51)	(Hui et al., 1999) (n=20)	(Fang et al., 2001) (n=47)	(Chien et al., 2001) (n=30)	(Yan et al., 2001) (n=17)	(Rodriguez et al., 2005) (n=10)	(Yan et al., 2005) (n=32)
Loss	Gene	Copy number loss frequencies (WES)			Chromosomal alteration frequencies (comparative genomic hybridization)						
3p	MST1R (3p21.31)	55.6%	54.6%	-	53%	75%	43%	20%	53%	40%	34%
	BAP1 (3p21.1)	55.6%	50.0%	-							
9p	CDKN2A (9p21.3)	32.3%	22.7%	42.3%	41%	25%	-	-	-	20%	25%
	CDKN2B (9p21.3)	31.3%	20.5%	42.3%							
11q	BIRC2 (11q22.2)	30.3%	18.2%	-	29%	45%	36%	23%	47%	40%	31%
	BIRC3 (11q22.2)	30.3%	18.2%	-							
14q	NFKBIA (14q13.2)	43.4%	52.3%	21.6%	35%	65%	21%	13%	47%	40%	47%
	TRAF3 (14q32.32)	57.6%	59.1%	22.5%							
16q	CYLD (16q12.1)	16.2%	9.1%	28.3%	-	50%	55%	16%	29%	30%	34%
	NLRC5 (16q13)	16.2%	9.1%	21.6%							

17p	TP53 (17p13.1)	20.2%	18.2%	16.2%	-	-	-	-	-	-	31%
-----	-------------------	-------	-------	-------	---	---	---	---	---	---	-----

“-” refers to unmentioned.

Response Table 3. Comparison of recurrent chromosomal gain frequencies in NPC

		Total cohort (n=99)	Lower purity cohort (n=44)	(Li et al., 2017) (n=111)	(Chen et al., 1999) (n=51)	(Hui et al., 1999) (n=20)	(Fang et al., 2001) (n=47)	(Chien et al., 2001) (n=30)	(Yan et al., 2001) (n=17)	(Rodriguez et al., 2005) (n=10)	(Yan et al., 2005) (n=32)
Gain	Gene	Copy number gain frequencies (WES)			Chromosomal gain frequencies (comparative genomic hybridization)						
3q	PIK3CA (3q26.32)	8.1%	4.6%	10.8%	-	30%	34%	20%	29%	30%	34%
7q	MET (7q31)	11.1%	13.6%	-	-	20%	-	13%	-	10%	-
	BRAF (7q34)	12.1%	15.9%	-							
	KMT2C (7q36.1)	12.1%	15.9%	11.7%							
8q	RIPK2 (8q21.3)	12.1%	9.1%	-	-	30%	-	27%	29%	10%	47%
	NIBP (8q24.3)	17.2%	11.4%	-							
11q	CCND1 (11q13.3)	15.2%	15.9%	44.1%	41%	-	-	-	35%	20%	-
12p	KRAS (12p12.1)	31.3%	25.0%	24.3%	59%	60%	-	36%	-	40%	41%
	TNFRSF1A (12p13.31)	28.3%	20.5%	-							
	LTBR (12p13.31)	27.3%	20.5%	-							
12q	ERBB3 (12q13.2)	23.2%	15.9%	-	35%	60%	51%	33%	41%	10%	53%
	KMT2D (12q13.12)	21.2%	15.9%	9.0%							

“-” refers to unmentioned.

3). For addressing on “poor genome data quality due to low sequencing depth” concern, we conducted several quality control analyses for association between sequencing coverage and mutation calling.

A). We found that there was no obvious correlation between sequencing coverage and the detected SNV amount per sample in our cohort under the coverage range (Response Figure 3).

Response Figure 3

Response Figure 3. The association between SNV amount and tumor sequencing coverage. No obvious correlation was observed ($R=0.091$, $p=0.39$).

B). We have also paid careful attention when we designed the pipeline for mutation calling to maintain high standards on data reliability. In our pipeline, besides applying three callers (MuTect2, Strelka2 and LANCET) for mutation identification and filtering mutations with two-two overlapping, we also set up a criterion that only if the mutated locus was covered by $\geq 20X$ depth, the mutation was kept as candidate true mutations for further analysis. We examined the mutational signature patterns among groups of different on-target coverage cutoffs, they were almost identical (Response Figure 4). With above strict filtering conditions, all 2662 identified SNVs were with at least 20X on-target coverage and were detected by at least two callers, they should be of good reliability.

Response Figure 4

Response Figure 4. Comparison on mutational signatures among groups of different on-target coverage cutoffs. There was no significant difference on mutational signatures among groups.

C). We excluded 7 pairs of samples with average sequencing depth lower than 10X in either tumor or normal control in current version, now totally there were 88 pairs of samples used for SNV identification. Compared with previous data, the updated results still maintained consistent findings on identified driver mutations (Figure 1D) after excluding these potentially low-quality data.

D). Since the microdissected (Luzhou) samples generally have low sequencing coverages. To examine the result reliability of the microdissected cohort, we performed several quality control analyses. We firstly compared mutational signatures between microdissected and total cohort, the results were closely similar (Response Figure 5). Secondly, we compared the top frequent driver SNVs identified from microdissected, total cohort and previous published studies. The NPC driver mutations showed comparable results (Response Table 1), suggesting the good reliability of mutation calling results in microdissected cohort.

Response Figure 5

Response Figure 5. Comparison on mutational signatures between microdissected and total cohort. (A) The mutational signature of microdissected cohort (n=44). (B) The mutational signature of total cohort (n=88). There was no significant difference on mutational signatures between two cohorts.

4). For addressing on “problems in informatics pipelines” concern, we outlined our pipeline in Supplementary Figure 18. Our bioinformatic pipelines are similar with others and the top frequent mutations/CNVs are consistent to previous studies with similar prevalence (Response Table 1-3, Supplementary Table 2-4). In order to get reliable results, to overlap mutations identified by multiple callers and exclude mutations of low coverage are effective practice, it is well accepted by the field. Although there are some samples with low cellularity, which is very commonly found in NPC tumors, we have used microdissection and increased sequencing depth to minimize the influence caused by the tumor purity issue, and the quality of our data is equivalent to previous sequencing studies.

5). For addressing on “orthogonal validation of the mutations” suggestion, we have made several efforts to confirm the reliability of key SNVs/CNVs identified in our study.

A). We performed Sanger sequencing validation for a few key mutations listed in Figure 1D. The confirming rate of true mutation is 100% (38/38) (Supplementary Data 4).

B). We compared the driver mutation prevalence with previous studies. The top driver SNVs/CNVs and their mutational frequencies in NPC are very consistent between our study and multiple previous studies (Supplementary Table 2-4; Response Table 1-3). The consistent finding should serve as another credible evidence to demonstrate the good reliability of identified mutations/CNVs presented in our manuscript.

C). Our data demonstrated frequent chromosomal deletions of Chr 3p (locus of *MST1R*), 9p, 11q, 14q and 16q, and amplifications of 3q, 8q, 12p and 12q in NPC, which is

consistent to previous study (Li et al., 2017) (Response Figure 6).

Response Figure 6

Response Figure 6. Global chromosomal gains (shown in red) and deletions (shown in blue) across NPC tumors.

(A) Data obtained from the published Nature Communications paper (Li et al., 2017).

(B) Data of our cohort showed frequent chromosomal deletions of Chr 3p, 9p, 11q, 14q and 16q, and amplifications of 3q, 8q, 12p and 12q in NPC (n=99).

D). As we indicated in our earlier responses to reviewer, we had made efforts to performed FISH validation for *MST1R*, however, due to the technical problem, we failed. We then asked a commercial company to conduct the FISH for us, and the

company tried two times but still did not get it work. Because of this technical difficulty, we used IHC assay to validate the absence of MST1R and detected strong expression of MST1R in wildtype and amplified samples (n=8) but only weak or no signals in *MST1R* deleted samples (n=14) (Supplementary Figure 5A, B).

In sum, according to our quality control analyses, either the hypothetical low tumor purity issue or low sequencing coverage concern in some of samples had a very minimal impact on our overall mutation calling results, primarily because of our several pre-designed improvement actions on performing microdissection, increasing sequencing depth for bulk samples and applying strict mutation calling pipeline. Our quality control analyses demonstrated there are no significant quality difference on mutation calling results among total cohort, lower purity cohort, microdissected cohort and other previously published data (Response Table 1-3). Thus, our genomic analysis and mutation calling provide reliable and solid data.

2. The CNV data is wrongly interpreted to fit their hypothesis of NPC-subtypes. In page 8 (line 229-239), the authors misinterpret their data by mixing up the concept of functional mutations and copy number gains/losses with unknown significant of a large list of genes involved in multiple pathways, further misleading us their roles in driving various features among NPC subtypes.

We thank the reviewer very much for providing these comments, but we would like to respectively express different opinions.

It is well recognized by cancer field that CNV are closely associated with corresponding gene functions. Amplification potentially contributes to higher expression and some example cases are the clinical applications of ERBB2/Her2 in breast cancer and MET in lung cancer. Vice versa, deletion may indicate loss of function on corresponding gene.

To be specifically in NPC, Li et al. identified activation of NF- κ B pathway as driver of NPC, primarily was based on the recurrent copy number losses of *CYLD*, *TRAF3*, *NFKBIA* and *NLRC5*, which were all considered as negative regulators of NF- κ B pathway in previous Nature Communication study (Li et al., 2017). Lin et al. and Li et al. identified aberrant G1-S transition was NPC driver pathway, primarily according to frequent copy number losses of *CDKN2A/B* and *TP53* and copy number gain of *CCND1* and *MYC* in previous Nature Genetics and Nature Communication studies (Li et al., 2017; Lin et al., 2014). Similar strategies were also applied on identifying the NPC oncogenic driver roles of RTK (*PI3KCA*, *KRAS* and *AKT2* gain; *PTEN* loss) and chromatin remodeling (*KMT2C* and *KMT2D* gain; *FBXW7* loss) and other important pathways and genes (Li et al., 2017; Lin et al., 2014). In multiple NPC genomics studies, CNVs were faithfully used for analyzing functional pathways.

In our manuscript, we carefully analyzed the subtype-specific CNV frequencies and identified the CNVs with particularly higher recurrence in one subtype but showed

lower frequencies in other subtypes. To further illustrate the functional pathways involved by these identified subtype-specific CNVs, in addition to routine functional annotation by pathway enrichment analysis, we have examined all individual genes listed in Figure 2C by literature review on their functional studies, which means all functional annotation were reference-based rather than “with unknown significance”. For example, we identify activation of microtubule polymerization was driver pathway in EC subtype and contributed to drug resistance to microtubule inhibitors. Copy number gains of *TPPP*, *MAP2*, *PHLDB2*, *DNAH5* and losses of *KIF2A*, *KIF3A*, *STK11* were recurrently mutated in EC subtype but not in SC subtype. By literature confirmation, *TPPP*, *MAP2*, *PHLDB2*, *DNAH5* were positive regulators for microtubule polymerization and *KIF2A*, *KIF3A*, *STK11* played negative roles. Activated microtubule polymerization would induce microtubule-targeting drug resistance. The microtubule polymerization associated drug resistance shown in EC subtype PDOs could be overcome by *TPPP* and *MAP2* knockout using CRISPR-Cas9 (Figure 7C; Supplementary Figure 14D), which further demonstrated our claiming on that activation of microtubule polymerization was driver pathway in EC subtype. Similar practice was used for identifying other important pathways listed in Figure 2C. Therefore, we politely consider that the comments of “misinterpret” and “misleading” here are not appropriate.

3. The “Pharmacogenomics-based precision medicine” strategy shows multiple fundamental concept problems.

The “Pharmacogenomics-based precision medicine (PBPM)” strategy referred in this study is a relatively new concept, specifically means the combination of genomic study and drug sensitive test to identify therapeutic options. It has been known that it is very difficult to identify real cancer drivers just based on DNA-sequencing. Our study is actually trying to explore this new concept by integrating drug response with transcriptomics and genomics, primarily CNVs. In transcriptomics integration part, we identified signature gene expressions for drug prediction and validated the predictive efficacy with independent clinical datasets (Figure 7E-I). In genomics integration part, we identified the association between microtubule inhibitor sensitivity with important CNVs involving in microtubule polymerization, mitosis regulation, DNA repair and NF-kB signaling (Figure 7A, B), which might be important druggable targets for NPC. We also performed gene manipulation validation (Figure 7C, D; Supplementary Figure 14D).

As we estimated, the “fundamental concept problems” comment comes from that the reviewer didn’t acknowledge the significance of CNVs on playing functional oncogenic roles and being associated with drug response. We have explained detailly in the response to point #2.

We agree that the proposed PBPM strategy is still at pilot stages and it is not perfect, which needs continuing efforts to improve. However, we politely disagree with the

comment “shows multiple fundamental concept problems”.

4. “The proposed treatments will only cause the selection of resistant cell population and rapid disease recurrence in these heterozygous tumors”.

Cancers are extremely difficult to treat with high motility (9.6 million cancer patients died in 2018). Patient-derived organoid (PDO) based drug sensitivity test combined with genomic sequencing has been recently developed for achieving precision oncology. We believe the drug sensitivity test will, at least, provide more choices for doctor in case there are a few drugs to test from. Currently, this approach is still in the experimenting stage and very few successful cases were reported.

Our current study is aiming to introduce a few more treatment options guided by genomics and tumor subtypes, which theoretically would be better than uniform regimens without any additional indications. In addition to chemotherapy alone, we also demonstrated the better effectiveness of combining IR and chemotherapy. We didn't make the strong claiming that our proposed treatment could eliminate all tumor cells, instead we investigated if there is any possible way to improve current clinical outcomes by utilizing genomics and PDOs, and made conclusions based on whatever we had. We neither made any claim that our approach could completely kill cancers.

Indeed, drug resistance is a big problem in cancer therapy and numerous scientists have been trying to use various approach to investigate it even if their approaches could not achieve complete killing. Our approach represents a new direction for precision oncology. Although it is still in trial stage, it already exhibits some advantages than many approaches using 2D cell culture (Ooft et al., 2019; Vlachogiannis et al., 2018; Yao et al., 2020; Yin et al., 2020). We sincerely hope our efforts should be encouraged rather than discouraged.

Response to Comments of Reviewer 2

Ren-Bo Ding's original manuscript provide an impressive amount of genomic and transcriptomic data of NPC subtypes, as well as new organoid derivation methods on which extensive drug responses have been performed and analyzed. The interested finding is that the drug screens revealed different sensitivities per morphology-based subtype classification.

The resource-like datasets will be of interest to the field, as well as the new methodology to grow PDOs. Although details or mechanistic underpinning remains challenging, the authors convince that PDOs shows differential drug sensitivity per NPC subtype. The manuscript will be helpful for the ongoing debate about NPC subtypes and treatments.

Minor comments: the figure legends are very concise and do not explain all features that can be seen in the panels (e.g. Fig5D-G, what is the additive?)

We thank reviewer very much for providing this constructive suggestion, and we have revised all figure legends with more detailed feature description accordingly. For example, in Figure 5D, we revised the legend description as following, “Representative dose response curves of the PDO11 treated with cisplatin alone and cisplatin combined with IR at 4 Gy. The single chemotherapy curve was presented with blue color, the combinational treatment curve was labelled with red color, the additive curve was marked with green color. The additive curve represented the estimated combined effect of chemotherapy and IR being equal to the sum of their separate effects. If the combinational treatment curve (red) was lower than the additive curve (green), the synergistic effect was demonstrated, vice versa. Good synergy was displayed by combinational treatment of cisplatin and IR on PDO11.”

Response to Comments of Reviewer 3

In this manuscript the authors analyzed a large cohort of NPC cancers and generated more than 40 patient derived organoids models to develop pharmacogenomic models for drug response. This is well done study.

My only concern is that the authors do not rule out the possibility the response to chemotherapies and CRT treatment (Figure 4 and Figure 5) relates to difference in the proliferative potential of the PDO models. In other words are the most responsive models more proliferative than those that do not respond well? The authors should determine doubling time for a panel of responders and non-responders to chemotherapies to determine the correlation, if any, between proliferative potential and response to chemotherapies.

We thank reviewer very much for providing this valuable comment. To rule out the possibility that treatment response differences might be due to the intrinsic diversity of different PDOs on proliferative rate. We provided the doubling time of treatment-native PDOs (Supplementary Table 5) and did correlation analyses between doubling time and drug sensitivities on responder- and non-responder-PDOs (Response Figure 7). The results demonstrated that there was obvious correlation between proliferative potential and responses to chemotherapies on PDOs.

Response Figure 7

Response Figure 7. Evaluation on intrinsic proliferative potential of PDOs on treatment responses.

(A) The association between doubling time of PDOs and response to docetaxel. No obvious correlation was observed ($R=0.00083, p=0.88$).

(B) The association between doubling time of PDOs and response to erlotinib. No obvious correlation was observed ($R=0.0062, p=0.68$).

Reference cited

Carter, S. L., Cibulskis, K., Helman, E., McKenna, A., Shen, H., Zack, T., Laird, P. W., Onofrio, R. C., Winckler, W., Weir, B. A., *et al.* (2012). Absolute quantification of somatic DNA alterations in human cancer. *Nat Biotechnol* 30, 413-421.

Chen, Y. J., Ko, J. Y., Chen, P. J., Shu, C. H., Hsu, M. T., Tsai, S. F., and Lin, C. H. (1999). Chromosomal aberrations in nasopharyngeal carcinoma analyzed by comparative genomic hybridization. *Genes, chromosomes & cancer* 25, 169-175.

Chien, G., Yuen, P. W., Kwong, D., and Kwong, Y. L. (2001). Comparative genomic hybridization analysis of nasopharyngeal carcinoma: consistent patterns of genetic aberrations and clinicopathological correlations. *Cancer genetics and cytogenetics* 126, 63-67.

Chow, Y. P., Tan, L. P., Chai, S. J., Abdul Aziz, N., Choo, S. W., Lim, P. V., Pathmanathan, R., Mohd Kornain, N. K., Lum, C. L., Pua, K. C., *et al.* (2017). Exome Sequencing Identifies Potentially Druggable Mutations in Nasopharyngeal Carcinoma. *Sci Rep* 7, 42980.

Fang, Y., Guan, X., Guo, Y., Sham, J., Deng, M., Liang, Q., Li, H., Zhang, H., Zhou, H., and Trent, J. (2001). Analysis of genetic alterations in primary nasopharyngeal carcinoma by comparative genomic hybridization. *Genes, chromosomes & cancer* 30, 254-260.

Favero, F., Joshi, T., Marquard, A. M., Birkbak, N. J., Krzystanek, M., Li, Q., Szallasi, Z., and Eklund, A. C. (2015). Sequenza: allele-specific copy number and mutation profiles from tumor sequencing data. *Annals of oncology : official journal of the*

European Society for Medical Oncology 26, 64-70.

Hui, A. B., Lo, K. W., Leung, S. F., Teo, P., Fung, M. K., To, K. F., Wong, N., Choi, P. H., Lee, J. C., and Huang, D. P. (1999). Detection of recurrent chromosomal gains and losses in primary nasopharyngeal carcinoma by comparative genomic hybridisation. *International journal of cancer* 82, 498-503.

Kim, S., Scheffler, K., Halpern, A. L., Bekritsky, M. A., Noh, E., Kallberg, M., Chen, X., Kim, Y., Beyter, D., Krusche, P., and Saunders, C. T. (2018). Strelka2: fast and accurate calling of germline and somatic variants. *Nat Methods* 15, 591-594.

Li, H., and Durbin, R. (2010). Fast and accurate long-read alignment with Burrows-Wheeler transform. *Bioinformatics* 26, 589-595.

Li, Y. Y., Chung, G. T., Lui, V. W., To, K. F., Ma, B. B., Chow, C., Woo, J. K., Yip, K. Y., Seo, J., Hui, E. P., *et al.* (2017). Exome and genome sequencing of nasopharynx cancer identifies NF-kappaB pathway activating mutations. *Nat Commun* 8, 14121.

Lin, D. C., Meng, X., Hazawa, M., Nagata, Y., Varela, A. M., Xu, L., Sato, Y., Liu, L. Z., Ding, L. W., Sharma, A., *et al.* (2014). The genomic landscape of nasopharyngeal carcinoma. *Nat Genet* 46, 866-871.

McKenna, A., Hanna, M., Banks, E., Sivachenko, A., Cibulskis, K., Kernytsky, A., Garimella, K., Altshuler, D., Gabriel, S., Daly, M., and DePristo, M. A. (2010). The Genome Analysis Toolkit: a MapReduce framework for analyzing next-generation DNA sequencing data. *Genome Res* 20, 1297-1303.

McLaren, W., Gil, L., Hunt, S. E., Riat, H. S., Ritchie, G. R., Thormann, A., Flicek, P., and Cunningham, F. (2016). The Ensembl Variant Effect Predictor. *Genome Biol* 17, 122.

Mermel, C. H., Schumacher, S. E., Hill, B., Meyerson, M. L., Beroukhim, R., and Getz, G. (2011). GISTIC2.0 facilitates sensitive and confident localization of the targets of focal somatic copy-number alteration in human cancers. *Genome Biol* 12, R41.

Narzisi, G., Corvelo, A., Arora, K., Bergmann, E. A., Shah, M., Musunuri, R., Emde, A. K., Robine, N., Vacic, V., and Zody, M. C. (2018). Genome-wide somatic variant calling using localized colored de Bruijn graphs. *Commun Biol* 1.

Ooft, S. N., Weeber, F., Dijkstra, K. K., McLean, C. M., Kaing, S., van Werkhoven, E., Schipper, L., Hoes, L., Vis, D. J., van de Haar, J., *et al.* (2019). Patient-derived organoids can predict response to chemotherapy in metastatic colorectal cancer patients. *Sci Transl Med* 11.

Rodriguez, S., Khabir, A., Keryer, C., Perrot, C., Drira, M., Ghorbel, A., Jlidi, R., Bernheim, A., Valent, A., and Busson, P. (2005). Conventional and array-based comparative genomic hybridization analysis of nasopharyngeal carcinomas from the Mediterranean area. *Cancer genetics and cytogenetics* 157, 140-147.

Talevich, E., Shain, A. H., Botton, T., and Bastian, B. C. (2016). CNVkit: Genome-Wide Copy Number Detection and Visualization from Targeted DNA Sequencing. *PLoS Comput Biol* 12, e1004873.

Tarasov, A., Vilella, A. J., Cuppen, E., Nijman, I. J., and Prins, P. (2015). Sambamba: fast processing of NGS alignment formats. *Bioinformatics* 31, 2032-2034.

Vlachogiannis, G., Hedayat, S., Vatsiou, A., Jamin, Y., Fernández-Mateos, J., Khan, K., Lampis, A., Eason, K., Huntingford, I., Burke, R., *et al.* (2018). Patient-derived

organoids model treatment response of metastatic gastrointestinal cancers. *Science* (New York, NY) *359*, 920-926.

Yan, J., Fang, Y., Liang, Q., Huang, Y., and Zeng, Y. (2001). Novel chromosomal alterations detected in primary nasopharyngeal carcinoma by comparative genomic hybridization. *Chinese medical journal* *114*, 418-421.

Yan, W., Song, L., Wei, W., Li, A., Liu, J., and Fang, Y. (2005). Chromosomal abnormalities associated with neck nodal metastasis in nasopharyngeal carcinoma. *Tumour biology : the journal of the International Society for Oncodevelopmental Biology and Medicine* *26*, 306-312.

Yao, Y., Xu, X., Yang, L., Zhu, J., Wan, J., Shen, L., Xia, F., Fu, G., Deng, Y., Pan, M., *et al.* (2020). Patient-Derived Organoids Predict Chemoradiation Responses of Locally Advanced Rectal Cancer. *Cell Stem Cell* *26*, 17-26 e16.

Yin, S., Xi, R., Wu, A., Wang, S., Li, Y., Wang, C., Tang, L., Xia, Y., Yang, D., Li, J., *et al.* (2020). Patient-derived tumor-like cell clusters for drug testing in cancer therapy. *Sci Transl Med* *12*.

Zhang, L., MacIsaac, K. D., Zhou, T., Huang, P. Y., Xin, C., Dobson, J. R., Yu, K., Chiang, D. Y., Fan, Y., Pelletier, M., *et al.* (2017). Genomic Analysis of Nasopharyngeal Carcinoma Reveals TME-Based Subtypes. *Mol Cancer Res* *15*, 1722-1732.

Zheng, H., Dai, W., Cheung, A. K., Ko, J. M., Kan, R., Wong, B. W., Leong, M. M., Deng, M., Kwok, T. C., Chan, J. Y., *et al.* (2016). Whole-exome sequencing identifies multiple loss-of-function mutations of NF- κ B pathway regulators in nasopharyngeal carcinoma. *Proc Natl Acad Sci U S A* *113*, 11283-11288.

Reviewers' comments:

Reviewer #3 (Remarks to the Author):

The authors have addressed my concerns in a satisfactory manner

Reviewer #4 (Remarks to the Author): Expert in cancer genomics and analysis of tumour purity

The authors of this manuscript focus on nasopharyngeal carcinoma (NPC) is a malignant head and neck cancer type with high morbidity. Using integrative pharmacogenomics, they show that NPC subtypes maintain distinct molecular features, drug responsiveness, and graded radiation sensitivity. Furthermore, they leverage patient-derived organoid (PDO)-based drug test identifies potential subtype-specific treatment regimens, in that SC and MSEC subtypes are sensitive to microtubule inhibitors, whereas EC subtype is more responsive to EGFR inhibitors and show through combinational chemoradiotherapy (CRT) screening, effective CRT regimens are also suggested for patients showing less sensitivity to radiation. The manuscript is well written and easy to understand. This review focuses on the computational cancer biology aspect of the manuscript with a special focus on the concerns brought up by Reviewer 1.

While the sample quality and tumor purity concerns raised by Reviewer 1 are adequately addressed in my opinion, there are several points came to my attention and should be addressed:

From 106 samples collected for WES analysis, 43 were isolated from bulk fresh tumors, but 4 from bulk FFPE and 59 from micro-dissected FFPE samples. Would the sample heterogeneity provide a batch effect? How does this breakdown correlate with identified subtypes?

The WES analysis pipeline seems appropriate both for somatic mutation and CNV analysis.

For transcriptomics analysis, was the multiple hypothesis testing correction applied?

What was the significance cutoff for pathway analysis?

Minor Comments:

Line 207: There is a typo: Should be “one of the interesting findingS”

Response to Comments of Reviewer 4

The authors of this manuscript focus on nasopharyngeal carcinoma (NPC) is a malignant head and neck cancer type with high morbidity. Using integrative pharmacogenomics, they show that NPC subtypes maintain distinct molecular features, drug responsiveness, and graded radiation sensitivity. Furthermore, they leverage patient-derived organoid (PDO)-based drug test identifies potential subtype-specific treatment regimens, in that SC and MSEC subtypes are sensitive to microtubule inhibitors, whereas EC subtype is more responsive to EGFR inhibitors and show through combinational chemoradiotherapy (CRT) screening, effective CRT regimens are also suggested for patients showing less sensitivity to radiation. The manuscript is well written and easy to understand. This review focuses on the computational cancer biology aspect of the manuscript with a special focus on the concerns brought up by Reviewer 1.

While the sample quality and tumor purity concerns raised by Reviewer 1 are adequately addressed in my opinion, there are several points came to my attention and should be addressed:

From 106 samples collected for WES analysis, 43 were isolated from bulk fresh tumors, but 4 from bulk FFPE and 59 from micro-dissected FFPE samples. Would the sample heterogeneity provide a batch effect? How does this breakdown correlate with identified subtypes?

We thank the reviewer very much for providing this valuable comment. We have examined the data carefully again and believed that different sample types, regarding bulk and microdissected samples, do not generate a significant batch effect on our cohort. We observe similar distribution frequencies of NPC top driver mutations between bulk and microdissected cohort (Response Figure 1A). Furthermore, there is no particular subtype preference among bulk fresh, bulk FFPE and microdissected samples (Response Figure 1A, B). The proportions of EC, MSEC and SC subtypes in these three sample batches are similar to those in total samples (n=106) (Response Figure 1B).

Response Figure 1

B

	EC	MSEC	SC	SCC
Bulk fresh tumors (n=43)	55.8%	27.9%	16.3%	0.0%
Microdissected FFPE (n=59)	52.5%	22.0%	20.3%	5.1%
Bulk FFPE (n=4)	50.0%	25.0%	25.0%	0.0%
Total tumors (n=106)	53.7%	24.5%	18.7%	2.8%

Response Figure 1. Evaluation of batch effect on identified subtypes.

- A. Oncoplot demonstrating subtype proportions and top driver mutations between bulk and microdissected cohorts.
- B. Proportions of EC, MSEC, SC and SCC subtypes among bulk fresh, bulk FFPE, microdissected FFPE and total tumors.

The WES analysis pipeline seems appropriate both for somatic mutation and CNV analysis.

Thank reviewer very much for the comments and recognition of our efforts.

For transcriptomics analysis, was the multiple hypothesis testing correction applied? What was the significance cutoff for pathway analysis?

We thank the reviewer very much for providing this constructive comment. We have performed the multiple hypothesis testing and applied significance cutoff during our transcriptomics analysis.

To be specific, we used DESeq2 and limma-voom for downstream differential expression analysis respectively. Differentially expressed genes were filtered by absolute value of \log_2 fold change >1 , p value <0.05 , FDR (padj) value <0.05 and further overlapped by the two methods (DESeq2 and limma-voom). The multiple hypothesis testing was performed using Benjamin-Hochberg correction implemented in the DESeq2 and limma package.

The pathway enrichment and GSEA was performed using the R package clusterProfiler, results were filtered by absolute NES value >1 , p value <0.05 and FDR value <0.05 .

The multiple hypothesis testing was performed using Benjamin-Hochberg correction implemented in clusterProfiler package.

Minor Comments:

Line 207: There is a typo: Should be “one of the interesting findingS”

We thank reviewer very much and have corrected the typo accordingly.

Reviewers' comments:

Reviewer #4 (Remarks to the Author):

Thank you, the authors have adequately addressed my concerns.